# Transformer Learns Optimal Variable Selection in Group-Sparse Classification

**Chenyang Zhang**[†]**, Xuran Meng**[‡]**, Yuan Cao**[†]
[†] The University of Hong Kong     [‡] University of Michigan, Ann Arbor
`chyzhang@connect.hku.hk, xuranm@umich.edu, yuancao@hku.hk`

## Abstract

Transformers have demonstrated remarkable success across various applications. However, the success of transformers have not been understood in theory. In this work, we give a case study of how transformers can be trained to learn a classic statistical model with "group sparsity", where the input variables form multiple groups, and the label only depends on the variables from one of the groups. We theoretically demonstrate that, a one-layer transformer trained by gradient descent can correctly leverage the attention mechanism to select variables, disregarding irrelevant ones and focusing on those beneficial for classification. We also demonstrate that a well-pretrained one-layer transformer can be adapted to new downstream tasks to achieve good prediction accuracy with a limited number of samples. Our study sheds light on how transformers effectively learn structured data.

## 1 Introduction

The Transformer architecture (Vaswani et al., 2017) has emerged as one of the most popular models in the modern deep learning, demonstrating remarkable success across a wide range of real-world applications, including language processing and language modeling (Vaswani et al., 2017; Radford et al., 2019; OpenAI, 2023), computer vision (Dosovitskiy et al., 2020; Rao et al., 2021), and reinforcement learning (Jumper et al., 2021; Chen et al., 2021; Janner et al., 2021). Despite its empirical achievements, the underlying mechanisms of transformers remain poorly understood due to their complex architecture, especially the mechanism of the self-attention layers.

As the core of the Transformer architecture, the attention mechanism has consistently been the primary focus of research aimed at understanding how Transformers work. Some empirical research (Xu et al., 2019; Vig & Belinkov, 2019; Rao et al., 2021; Yao et al., 2021; Chen et al., 2022) demonstrated that the attention layer can effectively extract structure information by assigning different weights to different input tokens. To fully understand such property theoretically, researchers explore the expressive power of transformers across various aspects (Edelman et al., 2022; Bai et al., 2024). Edelman et al. (2022) study the capacity of single-head attention to approximate sparse function by presenting the covering-number of attention function class. Bai et al. (2024) demonstrate that the transformers can implement a broad class of standard machine learning algorithms in context with various data distributions.

In addition to examining the expressive power of Transformers with custom-designed parameters, some recent studies have focused on analyzing the training dynamics of Transformers to determine whether these favorable properties can be attained using popular optimization algorithms in the deep learning community. Zhang et al. (2024) provide a global convergence analysis of gradient flow on the linear regression in-context tasks, where they consider a linear layer for attention calculation. Huang et al. (2024); Siyu et al. (2024) extend this result to one-layer transformers with a softmax attention layer. Wang et al. (2024); Chen et al. (2024b) study the capacities of transformers to learn sparse linear regression tasks. Specifically, by exploring the optimization trajectory, Wang et al. (2024) show that one-layer transformers can effectively extract the positional information, and provide a rigorous convergence analysis. At the same time, fully-connected neural networks fail in the worst case. Chen et al. (2024b) study the mechanisms of multi-head attention, revealing that multi-head attention will exhibit a sparsity in multiple-layer transformers. All the preceding works focus

on the regression tasks, and for the classification task, Jelassi et al. (2022) theoretically demonstrate that the one-layer vision transformer trained by gradient descent can converge on spatially structured data and the attention map exhibit patch association. Tarzanagh et al. (2023a;b) prove that one-layer attention can converge in direction to the hard margin solution of SVM. Li et al. (2023a) provide a generalization error bound for vision transformers trained by stochastic gradient descent. Although these papers provide some insights into mechanisms of one-layer attention for classification tasks, their theoretical results rely on some specific initialization which is practically infeasible, or strong assumptions that input data follows some particular patterns or certain parameters of attention architecture remain fixed throughout the training.

To further explore how transformers extract structure information in classification tasks, we consider a classic statistical problem with "group sparsity". In this setting, input variables are generated from multiple groups, while the true label of this input is determined by variables from a single group. We investigate the properties of one-layer transformers trained by gradient descent on this data model. The major contributions of this paper can be summarized as follows:

1. We establish a tight global convergence analysis with a matching lower and upper bound for the population cross-entropy loss of a one-layer transformer trained by gradient descent (Theorem 2.2). All parameters in the one-layer transformer are jointly trained with the same learning rate from zero initialization, without imposing any prior knowledge of the desirable patterns of the parameters. Specifically, by precisely characterizing the global optimization trajectories of all trainable parameters, we decrypt the working mechanism of each component of the one-layer transformer for learning the "group sparse" data model. Our results theoretically verify that transformers can learn the optimal variable selections.

2. We demonstrate that the well pre-trained one-layer transformers on "group-sparse" inputs can be efficiently transferred to a downstream task sharing the same group sparsity pattern. Specifically, denote the number of variable groups as $D$, the dimension of variables in each group as $d$, and the downstream sample size as $n$. Then we show that the one-layer transformers fine-tuned by online-SGD on the downstream task can achieve $\widetilde{O}\left(\frac{d+D}{n}\right)$ generalization error bound (Theorem 3.2).

3. We conduct numerical experiments, empirically show that training loss will converge, and verify our conclusions regarding the optimization trajectories of trainable parameters. Specifically, the sparsity of the attention score matrix empirically demonstrates that one-layer transformers can effectively learn the optimal variable selection. Additionally, we transfer the pre-trained one-layer transformers to downstream tasks, and empirically show that it can achieve a good generalization performance with a small sample size. All these empirical observations back up our theoretical findings.

**Notation.** Given two sequences $\{x_n\}$ and $\{y_n\}$, we denote $x_n = O(y_n)$ if there exist some absolute constant $C_1 > 0$ and $N > 0$ such that $|x_n| \leq C_1|y_n|$ for all $n \geq N$. Similarly, we denote $x_n = \Omega(y_n)$ if there exist $C_2 > 0$ and $N > 0$ such that $|x_n| \geq C_2|y_n|$ for all $n > N$. We say $x_n = \Theta(y_n)$ if $x_n = O(y_n)$ and $x_n = \Omega(y_n)$ both holds. Besides, we denote $x_n = o(y_n)$ if, for any $\epsilon > 0$, there exists some $N(\epsilon) > 0$ such that $|x_n| \leq \epsilon|y_n|$ for all $n \geq N(\epsilon)$, and we denote $x_n = \omega(y_n)$ if $y_n = o(x_n)$. We use $\widetilde{O}(\cdot)$, $\widetilde{\Omega}(\cdot)$, and $\widetilde{\Theta}(\cdot)$ to hide logarithmic factors in these notations respectively. Moreover, we denote $x_n = \mathrm{poly}(y_n)$ if $x_n = O(y_n^D)$ for some positive constant $D$, and $x_n = \mathrm{polylog}(y_n)$ if $x_n = \mathrm{poly}(\log(y_n))$. For two scalars $a$ and $b$, we denote $a \vee b = \max\{a, b\}$ and $a \wedge b = \min\{a, b\}$. Finally, for any $n \in \mathbb{N}_+$, we use $[n]$ to denote the set $\{1, 2, \cdots, n\}$.

## 2 TRANSFORMERS LEARN "GROUP SPARSE" DATA

In this section, we present our theoretical findings that one layer transformer can learn "group sparse" data by implementing an optimal variable selection. We first introduce the definition of "group-sparse" data distributions and the one-layer transformer we study in this paper.

**Group sparse learning problem.** Assume the feature vector $\widehat{\mathbf{x}} \in \mathbb{R}^p$, the label $y$ of feature vector $\widehat{\mathbf{x}}$ is determined by a labeling function $\phi(\cdot) : \mathbb{R} \to \mathbb{R}$ as $y = \phi(\langle\widehat{\mathbf{x}}, \boldsymbol{\beta}^*\rangle)$, where $\boldsymbol{\beta}^* \in \mathbb{R}^p$ is a pre-defined ground truth. Furthermore, we assume there exists a predefined $D$ disjoint partitions of $[p]$, specifically, $[p] = \cup_{j=1}^D G_j$ and $G_i \cap G_j = \emptyset$ for any $i \neq j$. Then we refer to this learning

problem as "group sparse" if the ground truth vector $\beta^*$ satisfies that

$$\text{supp}(\boldsymbol{\beta}^*) := \{k : \boldsymbol{\beta}_k^* \neq 0; k \in [p]\} \subset G_j$$

for some $j \in [D]$. In particular, if the labeling function $\phi(x) = x$, we denote this learning problem as group sparse linear regression. If the labeling function $\phi(x) = \text{sign}(x)$, we denote this learning problem as group sparse linear classification.

The definition above regarding group learning problem is motivated by Huang & Zhang (2010); Li et al. (2023b). In this paper, we focus on a binary group sparse linear classification problem. For simplicity, We consider the setting where all $G_j$'s are of the same size $d$, implying $p = dD$. Besides, we convert the feature vector $\widehat{\mathbf{x}} \in \mathbb{R}^{dD}$ into a matrix $\mathbf{X} \in \mathbb{R}^{d \times D}$, where each column $\mathbf{x}_j$ is the collection of the variables from $G_j$. Let $j^*$ be the index of the label-relevant group, i.e. $\text{supp}(\boldsymbol{\beta}^*) \subset G_{j^*}$. Then we can denote $\mathbf{v}^*$ the $d$-dimensional vector obtained by restricting $\boldsymbol{\beta}^*$ on $G_{j^*}$, so that $\langle \boldsymbol{\beta}^*, \widehat{\mathbf{x}} \rangle = \langle \mathbf{v}^*, \mathbf{x}_{j^*} \rangle$. Lastly, we assume the features are generated from Gaussian distribution. We provide a formal definition of the data model as follows:

**Definition 2.1** (Group sparse inputs following Gaussian distribution). Let $\mathbf{v}^* \in \mathbb{R}^d$ be a fixed vector representing the parameters of the labeling function, and $j^* \in [D]$ be the index of the label-relevant group. The binary classification data input pair $(\mathbf{X}, y) \in \mathbb{R}^{d \times D} \times \{1, +1\}$ is generated from the following distribution $\mathcal{D}$:

1. The features are generated from Gaussian distribution, i.e., $\mathbf{X} = [\mathbf{x}_1, \mathbf{x}_2, \ldots, \mathbf{x}_D]$ while each column $\mathbf{x}_j, j \in [D]$ represents a group of variables and is independently generated from $N(\mathbf{0}, \sigma_x^2 \mathbf{I}_d)$.

2. The label of this input $\mathbf{X}$ is determined by the features from the label-relevant group indexed by $j^*$, defined as $y = \text{sign}(\langle \mathbf{x}_{j^*}, \mathbf{v}^* \rangle)$.

Notice that the label $y$ is determined solely by the direction of $\mathbf{v}^*$, while the norm of $\mathbf{v}^*$ does not affect the data distribution. For simplicity of expression, we assume that $\|\mathbf{v}^*\|_2 = 1$ in the following.

**One-layer self-attention transformer.** For each $\mathbf{x}_j$ with $j \in [D]$ from data distribution $\mathcal{D}$ defined in Definition 2.1, we define the corresponding positional encoding $\mathbf{p}_j$ as

$$\mathbf{p}_j = \left[ \sin\left( j \frac{\pi}{D+1} \right), \sin\left( 2j \frac{\pi}{D+1} \right), \cdots, \sin\left( Dj \frac{\pi}{D+1} \right) \right]^\top \tag{2.1}$$

The definition above is motivated by the fact that $\mathbf{p}_j, j \in [D]$ form an orthogonal basis. We generate our new training input by concatenating the Gaussian feature $\mathbf{X}$ with positional encoding matrix $\mathbf{P}$, which is defined as $\mathbf{P} = [\mathbf{p}_1, \mathbf{p}_2, \cdots, \mathbf{p}_D]$. Specifically, we define the new training input as $\mathbf{Z} = [\mathbf{z}_1, \mathbf{z}_2, \cdots, \mathbf{z}_D] \in \mathbb{R}^{(d+D) \times D}$ with $\mathbf{z}_j = [\mathbf{x}_j^\top, \mathbf{p}_j^\top]^\top$ for all $j \in [D]$. Now we introduce the one-layer self-attention architecture applied to the input $\mathbf{Z}$ as:

$$f(\mathbf{Z}, \mathbf{W}, \mathbf{v}) = \sum_{j=1}^{D} \mathbf{v}^\top \mathbf{Z} \mathcal{S}(\mathbf{Z}^\top \mathbf{W} \mathbf{z}_j) = \mathbf{v}^\top \mathbf{Z} \mathbf{S} \mathbf{1}_D. \tag{2.2}$$

Here $\mathcal{S}(\cdot) : \mathbb{R}^D \mapsto \mathbb{R}^D$ denote the softmax mapping as $\mathcal{S}(\mathbf{h})_j = \frac{e^{\mathbf{h}_j}}{\sum_{j'=1}^{D} e^{\mathbf{h}_{j'}}}$ and $\mathbf{S} = [\mathcal{S}(\mathbf{Z}^\top \mathbf{W} \mathbf{z}_1), \mathcal{S}(\mathbf{Z}^\top \mathbf{W} \mathbf{z}_2), \cdots, \mathcal{S}(\mathbf{Z}^\top \mathbf{W} \mathbf{z}_D)] \in \mathbb{R}^{D \times D}$. Besides, we denote the entry in the $j'$-th row and $j$-th column of $\mathbf{S}$ as $\mathbf{S}_{j',j}$. Compared to the classical single-head, one-layer self-attention structure in Vaswani et al. (2017); Dosovitskiy et al. (2020), we make some mild reparameterizations on the architecture:

1. We combine the query and key matrices into one trainable matrix $\mathbf{W} \in \mathbb{R}^{(d+D) \times (d+D)}$.

2. We replace the value matrix with one trainable value vector $\mathbf{v} \in \mathbb{R}^{d+D}$.

The consolidation of query matrix and key matrix is commonly considered in recent theoretical analyses of attention structure (Tian et al., 2023; Wang et al., 2024; Huang et al., 2024; Zhang et al., 2024). The simplification of the value matrix into a vector is because only one parameter vector $\mathbf{v}^*$ needs to be studied, and the value vector $\mathbf{v}$ could be regarded as a combination of the value matrix and a second layer with uniform weight. Similar simplification is also considered in some recent

studies about understanding the attention mechanisms in classification tasks. (Jelassi et al., 2022; Tarzanagh et al., 2023a).

**Loss function and training algorithm.** We consider minimizing the population cross-entropy loss to train the above one-layer attention transformer in (2.2). Specifically, the loss function is

$$\mathcal{L}(\mathbf{v}, \mathbf{W}) = \mathbb{E}_{(\mathbf{X}, y) \sim \mathcal{D}} \big[ \ell(y \cdot f(\mathbf{Z}, \mathbf{W}, \mathbf{v})) \big],$$

where $\ell(a) = \log(1+\exp(-a))$ is the cross-entropy loss function for binary classification. Although the choice of population loss implicitly assuming an infinite number training set is not feasible in practice. It can significantly simplify the training dynamics and enable us to focus on an analysis of the global optimization trajectories. This objective loss is considered in a large number of recent theoretical works (Jelassi et al., 2022; Zhang et al., 2024; Huang et al., 2024; Wang et al., 2024; Nichani et al., 2024; Chen et al., 2024b) focusing on the optimization analysis of transformers. Besides, as one of the most popular loss functions classification tasks, cross-entropy loss is widely considered in recent theoretical works concerning transformers on classification tasks (Jelassi et al., 2022; Tian et al., 2023; 2024). Compared to the hinge loss considered in Li et al. (2023a), it is more common and general in practice. We utilize the gradient descent algorithm to optimize the preceding loss function. We consider the joint training regime, where all trainable parameters $\mathbf{v}, \mathbf{W}$ are updated simultaneously with the same learning rate $\eta$, i.e.,

$$\mathbf{v}^{(t+1)} = \mathbf{v}^{(t)} - \eta \nabla_{\mathbf{v}} \mathcal{L}(\mathbf{W}^{(t)}, \mathbf{v}^{(t)}); \tag{2.3}$$

$$\mathbf{W}^{(t+1)} = \mathbf{W}^{(t)} - \eta \nabla_{\mathbf{W}} \mathcal{L}(\mathbf{W}^{(t)}, \mathbf{v}^{(t)}). \tag{2.4}$$

Besides, we consider zero initialization $\mathbf{v}^{(0)}, \mathbf{W}^{(0)} = \mathbf{0}$.

**Tight convergence bounds.** Following the preliminaries, the next theorem presents our main results regarding convergence bounds.

**Theorem 2.2.** For any $\epsilon > 0$, suppose that $D \geq \omega(\log^2(1/\epsilon))$, $d \leq O\big(\text{poly}(D)\big)$, $\sigma_x, \eta = \Theta(1)$ with $\sigma_x \leq 1/3$ and let $T^* = \Theta\big(D^3 \vee \frac{1}{D^3 \epsilon^3}\big)$. Under these conditions, it holds that

1. The self-attention extracts the variables from the label-relevant group: for any new sample $(\mathbf{X}, y) \sim \mathcal{D}$ with the corresponding softmax output matrix $\mathbf{S}^{(T^*)}$ at the $T^*$-th iteration, with probability at least $1 - \exp\big(-\Theta(\sqrt{D})\big)$, it holds that

   $$\mathbf{S}^{(T^*)}_{j^*, j} \geq 1 - \exp\big(-\Theta(D)\big)$$

   for all $j \in [D]$.

2. The first block of value vector aligns with the ground truth: $\mathbf{v}^{(T^*)} = [\mathbf{v}_1^{(T^*)}, \mathbf{0}_D^\top]^\top$ with $\mathbf{v}_1^{(T^*)} \in \mathbb{R}^d$, where

   $$\left\| \frac{\mathbf{v}_1^{(T^*)}}{\|\mathbf{v}_1^{(T^*)}\|_2} - \mathbf{v}^* \right\|_2 \leq \epsilon D \exp\big(-\Theta(\sqrt{D})\big).$$

3. The loss is sufficiently minimized:

   $$C_1\Big(\epsilon \wedge \frac{1}{D^2}\Big) \leq \mathcal{L}(\mathbf{v}^{(T^*)}, \mathbf{W}^{(T^*)}) \leq \epsilon \wedge \frac{1}{D^2},$$

   where $C_1$ is a positive constant solely depending on $\eta$ and $\sigma_x$ with $C_1 \leq 1$.

Theorem 2.2 implies that one-layer attention model (D.4) can learn the group-sparse data with an assumption of an infinite number of training data. It can achieve arbitrary small loss by imposing a mild assumption concerning the scale of sparsity in our data model $\mathcal{D}$. Specifically, for any given loss tolerance $\epsilon$, we need the number of variables group to be larger than the logarithm squared of the reciprocal for loss tolerance, i.e., $\log^2(1/\epsilon)$. Based on this assumption, Theorem 2.2 can establish a tight upper and lower bound on the convergence rate of population loss.

In addition to providing a tight convergence bound, Theorem 2.2 decrypts how each component of one-layer self-attention (2.2) takes effect when learning sparse group data, by precisely identifying the scale of the parameters at $T^*$-th iteration. Specially, for any new sample $(\mathbf{X}, y) \sim \mathcal{D}$,

the attention score regarding the label-relevant group at $T^*$-th iteration, i.e. $\mathbf{S}_{j,j}^{(T^*)}$ is approximately 1 with high probability. This observation indicates that the self-attention layer almost attends to the label-relevant group, effectively selecting the variables mostly beneficial for classification while disregarding all other irrelevant variables. Besides, the second block of $\mathbf{v}^{(T^*)}$ retains $\mathbf{0}$, implying positional encoding is only involved in the calculation of attention weight. Following the procedure of optimal variable selections, positional embedding is never presented in the final output of one-layer attention. This observation benefits our learning tasks as positional embedding can not provide additional beneficial information for classification. Finally, $\mathbf{v}_1^{(t)}$ is almost aligned with the ground truth direction $\mathbf{v}^*$, verifying that the value vector learns the optimal direction for the binary classification problem. These observations collectively guarantee that the one-layer transformer 2.2 can correctly implement the classification task on the "group sparse" data 2.1.

Notably, a recent work (Wang et al., 2024) studies the sparse token selection of transformers on a similar data model to ours. By characterizing the entire training dynamics, they demonstrate a global convergence of transformers on the sparse token selection task without further assumptions on initialization. Besides, they do not impose fixed positions on target tokens, which is more general than our settings. However, they assume the positional encoding of the next token an average of target tokens, which alleviates the technical challenges, is somewhat impractical. What matters most is that they consider the $l_2$ loss function in their settings, making their learning problem more akin to a regression task, which significantly distinguishes our optimization analysis from theirs.

## 3 DOWNSTREAM TASKS

The variable selection mechanism of one-layer self-attention enables it to be efficiently transferred to a downstream task sharing a similar structure. In this section, we provide a generalization error bound for the downstream task to theoretically verify this conclusion. We first introduce the definition of downstream task data distribution we study as follows:

**Definition 3.1.** Consider a downstream task with training data $(\mathbf{X}^{(i)}, y^{(i)}) \in \mathbb{R}^{d \times D} \times \{1, +1\}$, $i \in [n]$ generated from a new distribution $\widetilde{\mathcal{D}}$ satisfying:

1. Linear separability with a margin: $\max_{\mathbf{v}:\|\mathbf{v}\|_2 \leq 1} y^{(i)} \cdot \langle \mathbf{v}, \mathbf{x}_{j^*}^{(i)} \rangle \geq \gamma$ almost surely for all $i \in [n]$. Besides, the label-relevant group index $j^*$ is the same as the distribution $\mathcal{D}$ in Definition 2.1.

2. Each entry of $\mathbf{X}^{(i)}$ is independent sub-Gaussian, with $\left\|\mathbf{x}_{j,k}^{(i)}\right\|_{\psi_2} \leq \widetilde{\sigma}_x$ for all $i \in [n], k \in [d]$ and $j \in [D]$.

Compared to the data distribution for pre-training in Definition 2.1, we make downstream task distribution more general by relaxing the Gaussian data to sub-Gaussian data. Besides, the assumption of linear separability with a margin is commonly considered in the analysis concerning generalization risk bounds of classification tasks (Wu et al., 2024; Ji & Telgarsky, 2020; Cao & Gu, 2019).

**Fine-tuning with online stochastic gradient descent.** Similarly, we concatenate each $\mathbf{x}_j^{(i)}$ with positional encoding $\mathbf{p}_j$ as defined in (2.1), and generate the new training raining data $(\mathbf{Z}^{(i)}, y^{(i)})$, for $i \in [n]$. Then, we consider fine-tuning a one-layer attention model with the same structure defined in (2.2), and denote the new parameters as $\widetilde{\mathbf{v}}$ and $\widetilde{\mathbf{W}}$. We fine-tune the new parameters by online SGD, i.e.,

$$\widetilde{\mathbf{v}}^{(i+1)} = \widetilde{\mathbf{v}}^{(i)} - \widetilde{\eta} \nabla_{\mathbf{v}} \widetilde{\mathcal{L}}_i(\widetilde{\mathbf{W}}^{(i)}, \widetilde{\mathbf{v}}^{(i)}); \tag{3.1}$$

$$\widetilde{\mathbf{W}}^{(i+1)} = \widetilde{\mathbf{W}}^{(i)} - \widetilde{\eta} \nabla_{\mathbf{W}} \widetilde{\mathcal{L}}_i(\widetilde{\mathbf{W}}^{(i)}, \widetilde{\mathbf{v}}^{(i)}) \tag{3.2}$$

for all $i \in [n]$, where $\widetilde{\mathcal{L}}_i(\widetilde{\mathbf{W}}^{(i)}, \widetilde{\mathbf{v}}^{(i)}) = \ell(y^{(i)} \cdot f(\mathbf{Z}^{(i)}, \widetilde{\mathbf{W}}^{(i)}, \widetilde{\mathbf{v}}^{(i)}))$. The initialization are set as $\widetilde{\mathbf{v}}^{(0)} = \mathbf{0}$ and $\widetilde{\mathbf{W}}^{(0)} = \mathbf{W}^{(T^*)}$, which is obtained from the pre-trained model in Section 2. In the following theorem, we present a generalization error bound on this downstream task, which concerns an average of all iterates $\{(\widetilde{\mathbf{W}}^{(i)}, \widetilde{\mathbf{v}}^{(i)})\}_{i=1}^n$ of online SGD.

**Theorem 3.2.** Suppose that $D \geq \omega(\log^2(n))$, $d \leq O(\text{poly}(D))$, $\widetilde{\sigma}_x \leq O(1)$, and $\widetilde{\eta} = \Theta(\frac{1}{(d \vee D)D^2})$. Under these conditions and for any $\delta > 0$, it holds that

$$\frac{1}{n} \sum_{i=1}^{n} \mathbb{P}_{(\mathbf{X}, y) \sim \widetilde{\mathcal{D}}}\Big( y \cdot f(\mathbf{Z}, \widetilde{\mathbf{W}}^{(i)}, \widetilde{\mathbf{v}}^{(i)}) \leq 0 \Big) \leq O\Big( \frac{(d+D)\log^2 n}{\gamma^2 n} \Big) + O\Big( \frac{\log(1/\delta)}{n} \Big).$$

with probability at least $1 - \delta - n \exp(-\Theta(\sqrt{D}))$ over the randomness of $\{(\mathbf{X}^{(i)}, y^{(i)})\}_{i=1}^{n}$.

Theorem 3.2 establish a generalization error bound composed of two terms. The first term concerns the ratio between the practical dimension $d + D$ and sample size $n$. Here we consider $d + D$ as the practical dimension since the positional encoding is also involved in the learning process. The second term is a standard large-deviation error term. If we take $\gamma = \Theta(1)$ as most literature assumes, we can obtain a sample complexity on the order of $\widetilde{\Omega}(\frac{d+D}{\epsilon} + \frac{1}{\epsilon}\log(\frac{1}{\delta}))$. In comparison, the existing lower bound for the sample complexity of linear logistic regression on vectorized $\mathbf{X}$ is $\Omega(\frac{dD}{\epsilon} + \frac{1}{\epsilon}\log(\frac{1}{\delta}))$, according to the classical PAC learning (Long, 1995). This superiority in sample complexity theoretically demonstrates that one-layer transformers trained on "group data" can be effectively transferred to a similar downstream task.

## 4 PROOF SKETCH

In this section, we provide a comprehensive explanation of how a one-layer transformer implements variable selection on group sparse data. Furthermore, we share insights into the reasoning that led to the conclusions presented in Theorem 2.2, along with a proof sketch for the convergence bound.

For a clear illustration of each component in the one-layer self-attention, we first separate $\mathbf{v}^{(t)}$ and $\mathbf{W}^{(t)}$ into the following blocks,

$$\mathbf{v}^{(t)} = [(\mathbf{v}_1^{(t)})^\top, (\mathbf{v}_2^{(t)})^\top]^\top; \quad \mathbf{W}^{(t)} = \begin{bmatrix} \mathbf{W}_{1,1}^{(t)} & \mathbf{W}_{1,2}^{(t)} \\ \mathbf{W}_{2,1}^{(t)} & \mathbf{W}_{2,2}^{(t)} \end{bmatrix},$$

where $\mathbf{v}_1^{(t)} \in \mathbb{R}^d, \mathbf{v}_2^{(t)} \in \mathbb{R}^D, \mathbf{W}_{1,1}^{(t)} \in \mathbb{R}^{d \times d}, \mathbf{W}_{1,2}^{(t)} \in \mathbb{R}^{d \times D}, \mathbf{W}_{2,1}^{(t)} \in \mathbb{R}^{D \times d}$, and $\mathbf{W}_{2,2}^{(t)} \in \mathbb{R}^{D \times D}$. In the following, we will show that a precise characterization of all these blocks at $T^*$-th iteration plays a key role in our theoretical convergence analysis.

**A low-rank structure of $\mathbf{W}^{(T^*)}$ beneficial for attending to label-relevant group $j^*$.** The conclusion that the attention score $\mathbf{S}_{j^*,j}^{(T^*)} \approx 1$ with high probability for any new sample $(\mathbf{X}, y) \sim \mathcal{D}$ serves as a foundation for transformers to correctly implement classification on group-sparse data distribution $\mathcal{D}$. Without such a guarantee, the loss cannot converge regardless of the direction in which the value vector converges, since the product of the label and variables from irrelevant groups still follow a Gaussian distribution.

The following lemma accurately characterizes each block of $\mathbf{W}^{(T^*)}$, providing insights into the rational of $\mathbf{S}_{j^*,j}^{(T^*)} \approx 1$

**Lemma 4.1.** Under the same condition with Theorem 2.2, $\mathbf{W}^{(T^*)}$ satisfies that

$$\mathbf{W}_{1,2}^{(T^*)}, \mathbf{W}_{2,1}^{(T^*)} = \mathbf{0};$$

$$\mathbf{W}_{1,1}^{(T^*)} = \beta_1 \mathbf{v}^* \mathbf{v}^{*\top} + \mathbf{W}_{1,1,\text{error}}^{(T^*)};$$

$$\mathbf{W}_{2,2}^{(T^*)} = \beta_2 \Big( \sum_{j \neq j^*} (\mathbf{p}_{j^*} - \mathbf{p}_j) \Big) \Big( \sum_{j=1}^{D} \mathbf{p}_j^\top \Big) + \mathbf{W}_{2,2,\text{error}}^{(T^*)},$$

where $|\beta_1| \leq O(\sqrt{D})$, $\beta_2 = \Theta(\frac{1}{D^2})$. The error terms $\mathbf{W}_{1,1,\text{error}}^{(T^*)}$ and $\mathbf{W}_{2,2,\text{error}}^{(T^*)}$ are small such that

$$\big\| \mathbf{W}_{1,1,\text{error}}^{(T^*)} \big\|_2, \big\| \mathbf{W}_{2,2,\text{error}}^{(T^*)} \big\|_2 \leq \exp\big( -\Theta(\sqrt{D}) \big).$$

Lemma 4.1 indicates that $\mathbf{W}^{(T^*)}$ exhibits a particular pattern: (i). the off-diagonal blocks $\mathbf{W}_{1,2}^{(T^*)}$ and $\mathbf{W}_{2,1}^{(T^*)}$ remain $\mathbf{0}$; (ii). the leading component of left-top block $\mathbf{W}_{1,1}^{(T^*)}$ aligns directionally

with the projection matrix $\mathbf{v}^*\mathbf{v}^{*\top}$; (iii). the leading component of the right bottom block aligns directionally with $\left(\sum_{j\neq j^*}(\mathbf{p}_{j^*}-\mathbf{p}_j)\right)\left(\sum_{j=1}^{D}\mathbf{p}_j\right)^{\top}$. In the following, we explain why this pattern implies that $\mathbf{S}_{j^*,j}\approx 1$ for all $j$ with high probability, which benefits the label-relevant variable selection.

First, $\mathbf{W}_{1,2}^{(T^*)},\mathbf{W}_{2,1}^{(T^*)}=\mathbf{0}$ suggests that the features $\mathbf{x}_j$'s, do not engage with the positional embedding $\mathbf{p}_j$'s when calculating the attention scores. Besides, due to the orthogonality among the positional embedding $\mathbf{p}_j$'s, the position-position interaction term $\mathbf{p}_{j'}^{\top}\mathbf{W}_{2,2}^{(T^*)}\mathbf{p}_j$ takes a large value only when $j'=j^*$. Furthermore, by standard concentration inequalities, it can be shown that $\mathbf{x}_{j'}^{\top}\mathbf{W}_{1,1}^{(T^*)}\mathbf{x}_j \ll \mathbf{p}_{j'}^{\top}\mathbf{W}_{2,2}^{(T^*)}\mathbf{p}_j$ for all $j,j'\in[D]$ with high probability. All these observations collectively conclude that $\mathbf{p}_{j^*}^{\top}\mathbf{W}_{2,2}^{(T^*)}\mathbf{p}_j$ will eventually dominate the softmax calculation, which finally implies that $\mathbf{S}_{j^*,j}^{T^*}\approx 1$.

**Accurate characterization of the alignment between $\mathbf{v}^{(T^*)}$ and ground truth $\mathbf{v}^*$.** Based on Lemma 4.1 and preceding discussions, we have shown that one-layer attention can effectively extract the features from the label-relevant group. The remaining challenge is to accurately characterize the alignment between $\mathbf{v}^{(T^*)}$ and ground truth $\mathbf{v}^*$ in terms of both direction and scale. The following lemma presents this result.

**Lemma 4.2.** Under the same condition with Theorem 2.2, it holds that,

$$\mathbf{v}_1^{(T^*)}=\alpha^{(T^*)}\mathbf{v}^*+\mathbf{v}_{1,\text{error}}^{(T^*)};\quad \mathbf{v}_2^{(T^*)}=\mathbf{0},$$

where the error term $\mathbf{v}_{1,\text{error}}^{(T^*)}$ satisfies that $\langle\mathbf{v}_{1,\text{error}}^{(T^*)},\mathbf{v}^*\rangle=0$ and $\left\|\mathbf{v}_{1,\text{error}}^{(T^*)}\right\|_2 \leq \exp\left(-\Theta(\sqrt{D})\right)$. Besides, the coefficient of the projection of $\mathbf{v}^{(T^*)}$ onto the direction of $\mathbf{v}^*$, which is denoted by $\alpha^{(t)}$, follows that

$$C_2\left((T^*)^{\frac{1}{3}}+D\right)\leq \alpha^{(T^*)}\leq C_3\left((T^*)^{\frac{1}{3}}+D\right),$$

where $C_2,C_3$ are both positive constants solely depending on $\sigma_x$ and $\eta$.

Lemma 4.2 reveals two important facts. Firstly, the fact that $\mathbf{v}_2^{(T^*)}=\mathbf{0}$ implies that the positional embedding part only contributes to attention weight calculation, and is not presented in the output. Furthermore, $\mathbf{v}_1^{(T^*)}$ is well aligned with the direction of $\mathbf{v}^*$, and the scale of its projection onto this direction is accurately characterized by a matching lower and upper bound. Besides, the second conclusion of Theorem 2.2 is a direction corollary of this lemma as $\left\|\frac{\mathbf{v}_1^{(T^*)}}{\|\mathbf{v}_1^{(T^*)}\|_2}-\mathbf{v}^*\right\|_2 \leq \frac{2\|\mathbf{v}_1^{(T^*)}\|_2}{\alpha^{(T^*)}}$.

**Upper and lower bounds of $y\cdot f(\mathbf{Z},\mathbf{W}^{(T^*)},\mathbf{v}^{(T^*)})$.** A tight upper and lower bound of $y\cdot f(\mathbf{Z},\mathbf{W}^{(T^*)},\mathbf{v}^{(T^*)})$ is undoubtedly of utmost significance for the analysis of global loss convergence. The accurate characterization of $\alpha^{(t)}$ in Lemma 4.2 enables us to get this matching bound. Before we present the result, we introduce a notation $E_{T^*}$ to denote the event that the conclusion in Theorem 2.2 holds, i.e., $\mathbf{S}_{j^*,j}^{(T^*)}\geq 1-\exp(\Theta(D))$. Then, we present our results regarding the matching bound of $y\cdot f(\mathbf{Z},\mathbf{W}^{(T^*)},\mathbf{v}^{(T^*)})$ in the following lemma.

**Lemma 4.3.** There exist two i.i.d. random variables $B_1,B_2$ with $B_1/\sigma_x^2,B_2/\sigma_x^2\sim\chi_D^2$, such that

$$yf(\mathbf{Z},\mathbf{W}^{(T^*)},\mathbf{v}^{(T^*)})\geq -D\alpha^{(T^*)}B_1^{\frac{1}{2}}-D\left\|\mathbf{v}_{1,\text{error}}^{(T^*)}\right\|_2 B_2^{\frac{1}{2}}.$$

Moreover, under the event $E_{T^*}$, it holds that

$$\frac{D\alpha^{(T^*)}}{2}y\langle\mathbf{v}^*,\mathbf{x}_{j^*}\rangle-1\leq yf(\mathbf{Z},\mathbf{W}^{(T^*)},\mathbf{v}^{(T^*)})\leq D\alpha^{(T^*)}y\langle\mathbf{v}^*,\mathbf{x}_{j^*}\rangle+1.$$

Based on these lemmas, we are ready to prove the conclusion of convergence bound in Theorem 2.2.

*Proof of Theorem 2.2.* We separate $\mathcal{L}(\mathbf{v}^{(T^*)},\mathbf{W}^{(T^*)})$ into two parts as

$$\mathcal{L}(\mathbf{v}^{(T^*)},\mathbf{W}^{(T^*)})=\underbrace{\mathbb{E}\left[\ell(yf(\mathbf{Z},\mathbf{W}^{(T^*)},\mathbf{v}^{(T^*)}))\mathbf{1}_{\{E_{T^*}\}}\right]}_{I_1}+\underbrace{\mathbb{E}\left[\ell(yf(\mathbf{Z},\mathbf{W}^{(T^*)},\mathbf{v}^{(T^*)}))\mathbf{1}_{\{E_{T^*}^c\}}\right]}_{I_2}.$$

For $I_1$, by the fact that $\exp(-x)/2 \le \ell(x) \le \exp(-x)$ for all $x > 0$, and utilizing the results of Lemma 4.3 for $E_{T^*}$ occurs, it holds that

$$I_1 \le \mathbb{E}\left[ e \exp\left( \frac{D\alpha^{(T^*)}}{2} y\langle \mathbf{v}^*, \mathbf{x}_{j^*}\rangle \right) \mathbf{1}_{E_{T^*}} \right] \le \sqrt{\frac{2}{\pi}} \frac{2e}{\sigma_x D\alpha^{(T^*)}};$$

$$I_1 \ge \mathbb{E}\left[ \frac{1}{2e} \exp\left( -D\alpha^{(T^*)} y\langle \mathbf{v}^*, \mathbf{x}_{j^*}\rangle \right) \mathbf{1}_{E_{T^*}} \right] \le \sqrt{\frac{2}{\pi}} \frac{1}{4e\sigma_x D\alpha^{(T^*)}}.$$

The last steps of both formulas are derived by basic integral since $y\langle \mathbf{v}^*, \mathbf{x}_{j^*}\rangle$ follows a folded normal distribution with mean 0 and variance $\sigma_x^2$ (See more details in Lemma E.5 and Lemma E.6). Furthermore, for $I_2$, we provide an upper-bound as

$$I_2 \le \sqrt{\mathbb{E}\left[ \log^2\left( 1 + \exp\left( D\alpha^{(T^*)} B_1^{\frac{1}{2}} + D\|\mathbf{v}_{1,\text{error}}^{(T^*)}\|_2 B_2^{\frac{1}{2}} \right) \right) \right]} \sqrt{\mathbb{P}(E_{T^*}^c)}$$

$$\le 2\sqrt{2D^2\left(\alpha^{(T^*)}\right)^2 \mathbb{E}[B_1] + 2D^2\|\mathbf{v}_{1,\text{error}}^{(T^*)}\|_2^2 \mathbb{E}[B_2]} \sqrt{\mathbb{P}(E_{T^*}^c)} \le \frac{2\sqrt{2}\sigma_x D^{\frac{3}{2}}\left( \alpha^{(T^*)} + \|\mathbf{v}_{1,\text{error}}^{(T^*)}\|_2 \right)}{e^{\Theta(\sqrt{D})}}.$$

The first inequality is from Cauchy-Schwarz inequality and Lemma 4.3. The second inequality holds by the fact that $\log(1 + a) \le 2\log a$ for large $a$, and $(a + b)^2 \le 2(a^2 + b^2)$. The third inequality is derived by applying the definition of $B_1$, $B_2$ in Lemma 4.3 and the fact $\mathbb{P}(E_{T^*}^c) \le e^{-\Theta(\sqrt{D})}$ from the first conclusion in Theorem 2.2. (We provide a rigorous detail in Lemma C.5.) By the upper-bound of $\alpha^{(T^*)}$ in Lemma 4.2, the definition of $T^*$ in Theorem 2.2 and our condition $D \ge \omega(\log^2(1/\epsilon))$, we know that $\alpha^{(T^*)} \ll e^{-\Theta(\sqrt{D})}$, indicating that $I_2 \ll I_1$. Let $T^* = (\sqrt{\frac{\pi}{2}} \frac{C_2 \sigma_x}{4e})^3 (D^3 \vee \frac{1}{D^3 \epsilon^3})$, by applying the lower and upper bounds of $\alpha^{(T^*)}$ in Lemma 4.2, we can finally obtain that

$$\mathcal{L}(\mathbf{v}^{(T^*)}, \mathbf{W}^{(T^*)}) \le 2I_1 \le \sqrt{\frac{2}{\pi}} \frac{4e}{C_2 \sigma_x} \frac{1}{D(T^*)^{1/3} + D^2} \le \epsilon;$$

$$\mathcal{L}(\mathbf{v}^{(T^*)}, \mathbf{W}^{(T^*)}) \ge I_1 \ge \sqrt{\frac{2}{\pi}} \frac{1}{4eC_3 \sigma_x} \frac{1}{D(T^*)^{1/3} + D^2} \ge \frac{C_2}{32e^2 C_3} \epsilon.$$

This completes the proof. □

## 5 NUMERICAL EXPERIMENTS

In this section, we conduct numerical experiments on synthetic data to verify our theoretical conclusions. Two types of synthetic data are generated following the group sparse data distribution defined in Definition 2.1 and downstream task data distribution defined in Definition 3.1 respectively. For each type of synthetic data, we generate two samples with different sample sizes $n$, variable group numbers $D$, and variable dimensions $d$. Besides, the variable group numbers $D$, and variable dimensions $d$ are matched across these two different types of synthetic data to enable the parameters of one-layer self-attention to be transferred to downstream tasks.

In the experiments where transformers learn variable selection, we generate data according to Definition 2.1, setting $\sigma_x = 0.25$ and $j^* = 2$, with two different pairs of $(n, d, D)$: $(500, 4, 6)$ and $(200, 2, 4)$. The vector $\mathbf{v}^*$ is randomly generated and then fixed. We set the learning rate $\eta = 0.5$ and train the models for 400 iterations. During the training process, we plot the training loss, the cosine similarity $\frac{\langle \mathbf{v}_1^{(t)}, \mathbf{v}^* \rangle}{\|\mathbf{v}_1^{(t)}\|\|\mathbf{v}^*\|}$ and the norm ratio $\|\mathbf{v}_1^{(t)}\|/\|\mathbf{v}_2^{(t)}\|$. After the training loss converges (at final iteration), we calculate the attention score matrix for each sample and display the heatmap of the average attention score matrix across all samples.

As shown in Figures 1, the training loss converges to 0 after sufficient iterations, and $\mathbf{v}_1^{(t)}$ rapidly aligns with the direction of $\mathbf{v}^*$ early in the training and continues in that direction for the rest of the iterations. When the training loss converges, we can also observe in Figure 1 that $\|\mathbf{v}_2^{(t)}\|$ is relatively small compared to $\|\mathbf{v}_1^{(t)}\|$. When we examine the attention matrix, as shown in Figure 2,

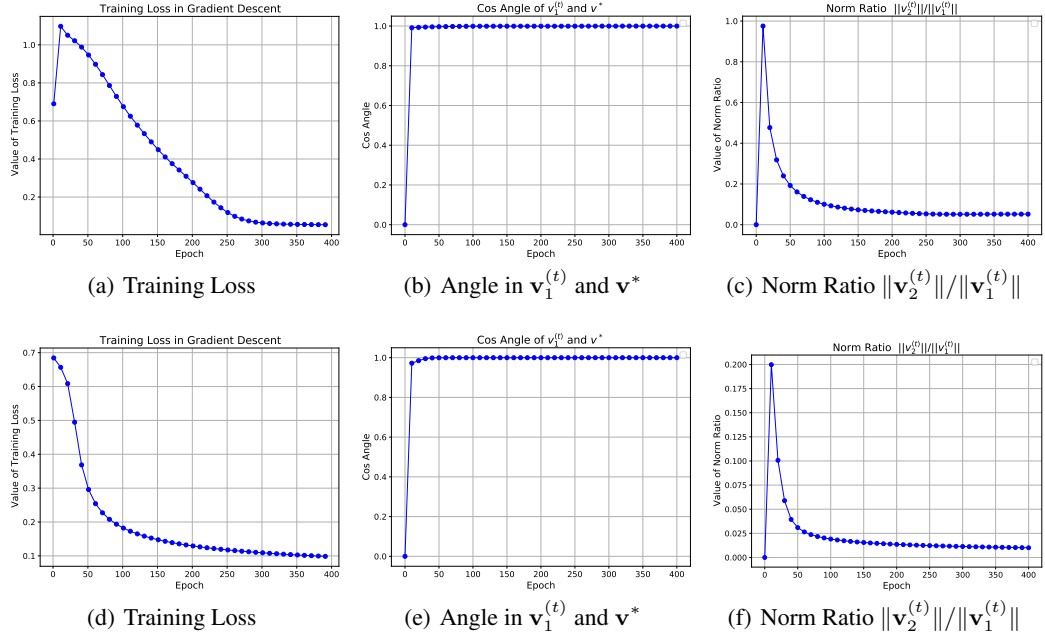

Figure 1: Figures on training loss, cosine similarity and norm ratio. The first line presents the training results with a sample size of 400, 6 variable groups, and a variable dimension of 4. The second line shows the training results for a sample size of 200, with 4 variable groups and a variable dimension of 2.

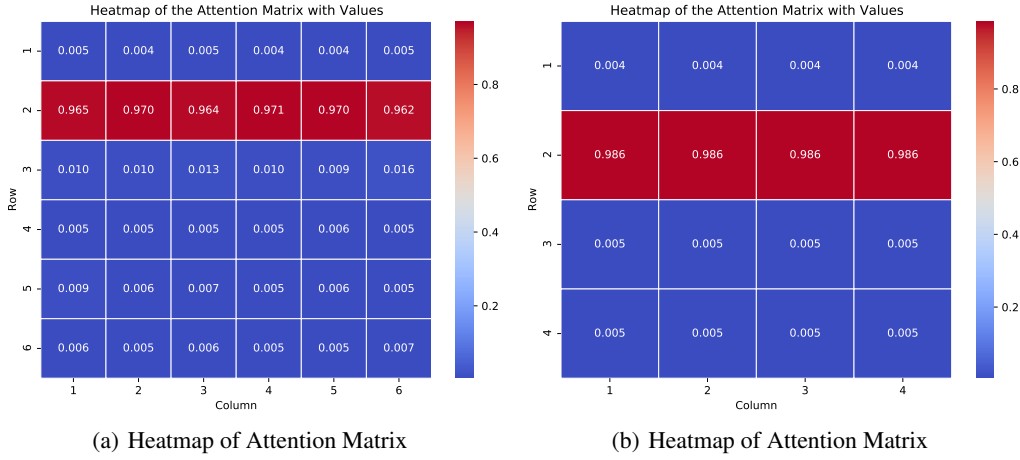

Figure 2: Heatmap of the average attention matrix. Figure 2(a) shows the heatmap of the attention matrix corresponding to the 6 variable groups, and Figure 2(b) shows the heatmap of the attention matrix corresponding to the 4 variable groups.

we observe that the second row is significantly larger than the others, indicating that the attention is predominantly focused on this row. This phenomenon well matches the prediction from our theory.

We conducted additional experiments on the downstream tasks by generating two new Gaussian samples, each with a different $\mathbf{v}^*$ from the previous setup. These samples follow Definition 3.1, with parameters set to $\widetilde{\sigma}_x = 1$ and $\gamma = 1$, while using the previous configurations for $d$ and $D$. To ensure linear separability with a margin, we applied a projection to adjust the data accordingly. The test accuracy is calculated after each iteration using 1000 test samples. Both experiments use a sample size of 400, and the learning rate is set to $10^{-3}$.

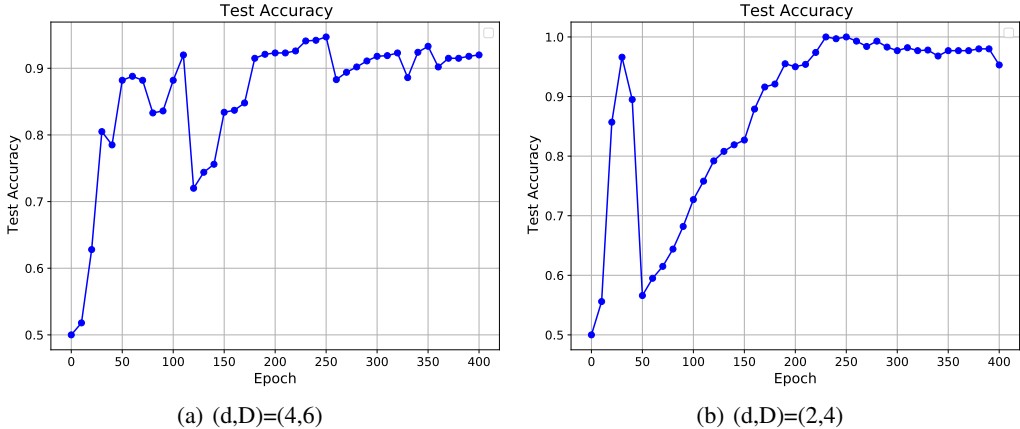

(a) (d,D)=(4,6)  (b) (d,D)=(2,4)

Figure 3: Test accuracy in the downstream task performance with different variable group numbers and variable dimensions.

As shown in Figure 3, the test accuracy initially oscillates during the early stages of training. However, after a longer training period, the test accuracy stabilizes and converges to 1 in both settings. This oscillation is primarily due to the transition in the direction of the vector $\mathbf{v}_1^{(t)}$ during training. According to our analysis, once $\mathbf{v}_1^{(t)}$ completes its direction transition, the test error decreases, leading to stable learning in the downstream task. These trends are consistent with the theory, confirming that the pre-trained transformers perform effectively on downstream tasks.

## 6 CONCLUSIONS AND LIMITATIONS

In this paper, we study how one-layer transformers trained by gradient descent learn a binary classification group sparse data, without imposing any prior knowledge on the initialization. We provide an accurate characterization of optimization trajectories. Based on this result, we theoretically demonstrate that the one-layer transformers can almost attend to the variables from the label-relevant group, and disregard other ones by leveraging the self-attention mechanism. Besides, we establish a tight convergence rate with a matching lower and upper bound for the population loss. Furthermore, we also propose that a pre-trained one layer transformers on a group sparse data can be effectively transferred to a downstream task with the same sparsity pattern. We theoretically demonstrate this conclusion by providing an improved generalization error bound for one-layer transformers, which surpasses that of linear logistic regression applied to vectorized features. Our numerical experiment observations support the theoretical findings. Although our theoretical findings provide insights into the mechanism of self-attention in variable selection, we focus exclusively on a one-layer self-attention model. Future research could be more intriguing if it explored deeper transformer architectures. Additionally, investigating the integration of self-attention with other modules, such as MLPs, ResNets, and normalization layers, presents a promising direction for further work.

ACKNOWLEDGMENTS

We thank the anonymous reviewers for their helpful comments. Yuan Cao is supported by NSFC 12301657 and HK RGC-ECS 27308624.

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

## A  ADDITIONAL RELATED WORKS

**Expressiveness of transformers.** There has been a line of works studying the expressiveness of transformers. Yun et al. (2020); Dehghani et al. (2019); Pérez et al. (2021); Wei et al. (2022) investigate the universal approximation results for transformers on a general class of functions. Likhosherstov et al. (2021); Bhattamishra et al. (2022) evaluate the expressive power of transformers on boolean concept classes by sample complexity. Bhattamishra et al. (2020); Liu et al. (2023) explore the the capacity of transformers to recognize formal languages. Sahiner et al. (2022) study the equivalent finite-dimensional convex problems of transformers by the lens of convex duality. Olsson et al. (2022); Dong et al. (2022); Garg et al. (2022); Ruis et al. (2020) investigate the capacity of transforms to learn in context and interpret the attention layers as gradient descent iterations. Sanford et al. (2024) discuss the strengths and limitations of the representation power of attention layers, by demonstrating the necessary intrinsic complexity of parameters on two different tasks.

**Optimization of transformers.** Several recent works study the optimizations of transformers. Zhang et al. (2020) compare the performance of transformers trained using SGD and adaptive methods, attributing the subpar performance of SGD in training language models to the heavy-tail distributed noise it introduces. In contrast, by adjusting the batch size in experiments, Kunstner et al. (2023) empirically demonstrates that the varying performance of Adam and SGD can be ascribed to the differences in their geometric trajectories rather than the noise introduced by stochasticity. Pan & Li (2023) compare the convergence rate of Adam and SGD on transformers, and argue that Adam has a better directional sharpness of the update steps than SGD. Through a two-stage training regime, Li et al. (2023c) investigate the optimal parameters of transformers applied to a masked topic structure model. Ildiz et al. (2024) explain the mechanism of attention from the perspective of Markov chains. The output of transformers is generated by a context-conditioned Markov chain, with weights determined by the attention structure. Nichani et al. (2024) demonstrate that two-layer transformers trained by gradient descent can encoder the latent causal graph in the first attention layer when solving in-context learning tasks with latent causal structure. Tian et al. (2023) analyze the training dynamic of transformers with one self-attention layer and one decoder layer trained by SGD, revealing how transformers combine the tokens and attend more to distinct tokens. Tian et al. (2024) further investigate the training dynamics jointly with a MLP layer, predicting a particular pattern of attention map and theoretically decrypting the hierarchies of tokens. Li et al. (2024) investigates the training dynamic of transformers when performing one-nearest neighbor selection. Gao et al. (2024) studies the global convergence of transformers under particular conditions. Chen et al. (2024a) investigates the in-context learning capacity of transformers by studying the training dynamics of a two-layer transformer on $n$-gram Markov chain data.

## B  DETAILED COMPARISON WITH JELASSI ET AL. (2022)

In Jelassi et al. (2022), the authors provide a theoretical guarantee that one-layer vision transformers can learn the inner structure of data, which is defined as "patch association" by the authors. This result offers novel insights into the training dynamics of one-layer transformers.

However, both the conclusions and proof techniques of Jelassi et al. (2022) can not be directly extended or applied to this work due to the distinctive settings among these two studies. Specifically, Jelassi et al. (2022) considers a one-layer vision transformer defined as

$$f(\mathbf{X}) = \sigma(\mathbf{v}^\top \mathbf{X} \mathcal{S}(\mathbf{A}))\mathbf{1}_D,$$

where $\mathbf{X}$ is the sequence of tokens/feature groups with each column $\mathbf{x}_j$ denoting a token/feature group, $\sigma(\cdot)$ represents the activation function, $\mathbf{v}$ indicates the value vector, $\mathcal{S}(\cdot)$ represents the softmax function, and $\mathbf{A}$ is input matrix of the softmax function. Unlike the common design of transformers, they directly treat the entries of the matrix $\mathbf{A}$ as trainable parameters and consider training $\mathbf{A}$ with gradient descent. In contrast, we consider softmax attention with the formulation $\mathcal{S}(\mathbf{Z}^\top \mathbf{W} \mathbf{Z})$, and treat the coefficient matrix $\mathbf{W}$ as the trainable parameters, which aligns with the general design of transformers. Besides, the initialization of the value vector $\mathbf{v}$ in Jelassi et al. (2022) is assumed to strictly align with the direction of ground truth $\mathbf{v}^*$, which is a strong and impractical assumption. In comparison, we consider general zero initializations.

In addition to the distinctions among the settings of problems, our theoretical results are more precise and refined. While Jelassi et al. (2022) provides an upper bound on the number of iterations

needed to achieve a population loss of $1/\text{poly}(d)$, we offer both matching upper and lower bounds for iterations to reach arbitrarily small population loss. Furthermore, we present a sample complexity analysis for transfer learning, which surpasses the conclusion of linear logistics regression on vectorized inputs from the PAC learning theory. In contrast, Jelassi et al. (2022) does not include such sample complexity analyses.

## C  PROOF IN SECTION 4

In this section, we provide detailed proof for lemmas in Section 4. Before we demonstrate the proof details, we first introduce some basic gradient calculations of $\mathcal{L}(\mathbf{v}, \mathbf{W})$ w.r.t $\mathbf{v}, \mathbf{W}$ as,

$$
\begin{aligned}
\nabla_{\mathbf{v}}\mathcal{L}(\mathbf{W}, \mathbf{v}) &= \mathbb{E}\left[\ell'(yf(\mathbf{Z}; \mathbf{W}, \mathbf{v})) \cdot y \cdot \frac{\partial f(\mathbf{Z}; \mathbf{W}, \mathbf{v})}{\partial \mathbf{v}}\right] \\
&= \mathbb{E}\left[\ell'(yf(\mathbf{Z}; \mathbf{W}, \mathbf{v})) \cdot y \cdot \mathbf{ZS1}_D\right] = \mathbb{E}\left[\ell'(yf(\mathbf{Z}; \mathbf{W}, \mathbf{v}))\sum_{j=1}^{D} y \cdot \mathbf{z}_j \sum_{j'=1}^{D} \mathbf{S}_{j,j'}\right];
\end{aligned}
$$
(C.1)

$$
\begin{aligned}
\nabla_{\mathbf{W}}\mathcal{L}(\mathbf{W}, \mathbf{v}) &= \mathbb{E}\left[\ell'(yf(\mathbf{Z}; \mathbf{W}, \mathbf{v})) \cdot y \cdot \frac{\partial f(\mathbf{Z}; \mathbf{W}, \mathbf{v})}{\partial \mathbf{W}}\right] \\
&= \mathbb{E}\left[\ell'(yf(\mathbf{Z}; \mathbf{W}, \mathbf{v})) \cdot y \cdot \sum_{j=1}^{D}\sum_{j'=1}^{D} \mathbf{v}^{\top}\mathbf{z}_{j'} \sum_{j''\neq j'} \frac{\exp(\mathbf{z}_{j''}^{\top}\mathbf{W}\mathbf{z}_j) \cdot \exp(\mathbf{z}_{j'}^{\top}\mathbf{W}\mathbf{z}_j)}{[\sum_{j''=0}^{D} \exp(\mathbf{z}_{j''}^{\top}\mathbf{W}\mathbf{z}_j)]^2}(\mathbf{z}_{j'} - \mathbf{z}_{j''})\mathbf{z}_j^{\top}\right] \\
&= \mathbb{E}\left[\ell'(yf(\mathbf{Z}; \mathbf{W}, \mathbf{v})) \cdot y \cdot \sum_{j=1}^{D}\sum_{j'=1}^{D}\sum_{j''\neq j'} \mathbf{S}_{j',j}\mathbf{S}_{j'',j}\mathbf{v}^{\top}\mathbf{z}_{j'}(\mathbf{z}_{j'} - \mathbf{z}_{j''})\mathbf{z}_j^{\top}\right]. 
\end{aligned}
$$
(C.2)

For the following derivation, we first introduce a shorthand notation that $\ell'^{(t)} = \ell'(yf(\mathbf{Z}, \mathbf{W}^{(t)}, \mathbf{v}^{(t)}))$. Besides, we also use $\mathbf{A}$ to denote an orthogonal matrix $\mathbf{A}$ with $\mathbf{v}^*$ being its first column. We denote $\boldsymbol{\xi}_2, \cdots, \boldsymbol{\xi}_d$ the rest columns in $\mathbf{A}$, i.e.,

$$
\mathbf{A} = [\mathbf{v}^*, \boldsymbol{\xi}_2, \cdots, \boldsymbol{\xi}_d] \in \mathbb{R}^{d \times d}, 
$$
(C.3)

where $\langle \mathbf{v}^*, \boldsymbol{\xi}_i \rangle = 0$, $\langle \boldsymbol{\xi}_i, \boldsymbol{\xi}_{i'} \rangle = 0$ and $\|\boldsymbol{\xi}_i\|_2 = 1$ for all $i, i' \in \{2, \cdots, d\}$.

### C.1  RESTATEMENT OF LEMMA 4.2 AND LEMMA 4.1

For the sake of conciseness and coherence in the presentation, we rearrange some content from Lemma 4.2 and Lemma 4.1. Specifically, we consolidate the conclusion that $\mathbf{v}_2^{(t)}, \mathbf{W}_{1,2}^{(t)}, \mathbf{W}_{2,1}^{(t)} = \mathbf{0}$ into Proposition C.1. The remaining conclusions of Lemma 4.2 are presented in Lemma C.2. We also include the remaining conclusions regarding $\mathbf{W}_{1,1}^{(t)}$ from Lemma 4.1 in Lemma C.3, and the conclusions about $\mathbf{W}_{2,2}^{(t)}$ from Lemma 4.1 in Lemma C.4. We present these new lemmas in this subsection and proof them respectively in the following subsections of this part.

We first introduce our new Proposition C.1.

**Proposition C.1.** For iterates $\mathbf{v}^{(t)}$ and $\mathbf{W}^{(t)}$ of gradient descent in (2.3) and (2.4), it holds that $\mathbf{v}_2^{(t)} = \mathbf{0}$, $\mathbf{W}_{1,2}^{(t)} = \mathbf{0}$ and $\mathbf{W}_{2,1}^{(t)} = \mathbf{0}$ for all $t \geq 0$.

The next Lemma C.2 is a restatement of conclusion of $\mathbf{v}_1^{(t)}$ in Lemma 4.2.

**Lemma C.2.** Under the same condition as Theorem 2.2, it holds that

$$
\mathbf{v}_1^{(t)} = \alpha^{(t)}\mathbf{v}^* + \mathbf{v}_{1,\text{error}}^{(t)}
$$

for all $3 \leq t \leq T^*$, where the error term satisfies that $\langle \mathbf{v}^*, \mathbf{v}_{1,\text{error}}^{(t)} \rangle = 0$ and

$$
\left\|\mathbf{v}_{1,\text{error}}^{(t)}\right\|_2 \leq e^{-C_4\sqrt{D}}
$$
(C.4)

for some positive constant $C_4$ solely depending on $\sigma_x, \eta$. Besides, the coefficient $\alpha^{(t)}$ satisfy that

$$\left( \sqrt{\frac{2}{\pi}} \frac{3\eta}{2\sigma_x D}(t-3) + \frac{2\eta^3 \sigma_x^3}{125\pi} \sqrt{\frac{2}{\pi}} D^3 \right)^{\frac{1}{3}} \leq \alpha^{(t)} \leq \sqrt{\frac{\pi}{2}} \frac{2}{\eta \sigma_x^3 D^3} + \left( \sqrt{\frac{2}{\pi}} \frac{6\eta}{\sigma_x D}(t-3) + \frac{2\eta^3 \sigma_x^3}{\pi} \sqrt{\frac{2}{\pi}} D^3 \right)^{\frac{1}{3}}. \tag{C.5}$$

In the following Lemma C.3 and Lemma C.4, we present the conclusion concerning $\mathbf{W}_{1,1}^{(t)}$ and $\mathbf{W}_{2,2}^{(t)}$ from Lemma 4.1 respectively.

**Lemma C.3.** Under the same condition as Theorem 2.2, it holds that

$$\mathbf{W}_{1,1}^{(t)} = \beta_1 \mathbf{v}^* \mathbf{v}^{*\top} + \mathbf{W}_{1,1,\text{error}}^{(t)} \tag{C.6}$$

for all $3 \leq t \leq T^*$, where $|\beta_1| \leq c_1 \sqrt{D}$ for some positive constant $c_1$ solely depending on $\eta, \sigma_x$. Besides, the error term satisfies that

$$\|\mathbf{W}_{1,1,\text{error}}^{(t)}\|_2 \leq e^{-C_7 \sqrt{D}}. \tag{C.7}$$

for some positive constant $C_7$ solely depending on $\sigma_x, \eta$.

**Lemma C.4.** Under the same condition as Theorem 2.2, it holds that

$$\mathbf{W}_{2,2}^{(t)} = \beta_2 \Big( \sum_{j \neq j^*} \big( \mathbf{p}_{j^*} - \mathbf{p}_j \big) \Big) \Big( \sum_{j=1}^{D} \mathbf{p}_j^\top \Big) + \mathbf{W}_{2,2,\text{error}}^{(t)}, \tag{C.8}$$

for all $3 \leq t \leq T^*$, where $\frac{c_2}{D^2} \leq |\beta_2| \geq \frac{c_3}{D^2}$ for some positive constants $c_2, c_3$ solely depending on $\eta, \sigma_x$, and the error term satisfies that

$$\|\mathbf{W}_{2,2,\text{error}}^{(t)}\|_2 \leq e^{-C_7 \sqrt{D}}. \tag{C.9}$$

for some positive constant $C_7$ solely depending on $\sigma_x, \eta$.

**Lemma C.5.** Under the same condition with Theorem 2.2, with probability at least $1 - \exp\big( - C_8 \sqrt{D} \big)$, it holds that

$$\mathbf{S}_{j^*,j}^{(t)} \geq 1 - D \exp(-C_9 D); \quad \mathbf{S}_{j',j}^{(t)} \leq \exp(-C_9 D)$$

for all $3 \leq t \leq T^*$, $j \in [D]$ and $j' \neq j^*$. The coefficients $C_8, C_9$ are all constants solely depending on $\sigma_x$ and $\eta$.

## C.2 Proof of Proposition C.1

We will prove Proposition C.1 by induction. And we first introduce several lemmas, which will be used for further proof of Proposition C.1.

**Lemma C.6.** If $\mathbf{W}_{1,2}^{(t)}, \mathbf{W}_{2,1}^{(t)} = \mathbf{0}$, then $\mathbf{S}_{j_1,j_2}^{(t)}$ can be expressed as a function of $\{y \cdot \mathbf{x}_1, \cdots, y \cdot \mathbf{x}_D\}$, and is independent with $y$ for all $j_1, j_2 \in [D]$.

*Proof of Lemma C.6.* When $\mathbf{W}_{1,2}^{(t)}, \mathbf{W}_{2,1}^{(t)} = \mathbf{0}$, we can re-write $\mathbf{S}_{j_1,j_2}^{(t)}$ as

$$\mathbf{S}_{j_1,j_2}^{(t)} = \frac{\exp\big( \mathbf{z}_{j_1}^\top \mathbf{W}^{(t)} \mathbf{z}_{j_2} \big)}{\sum_{j_3=1}^{D} \exp\big( \mathbf{z}_{j_3}^\top \mathbf{W}^{(t)} \mathbf{z}_{j_2} \big)} = \frac{\exp\big( \mathbf{x}_{j_1}^\top \mathbf{W}_{1,1}^{(t)} \mathbf{x}_{j_2} + \mathbf{p}_{j_1}^\top \mathbf{W}_{2,2}^{(t)} \mathbf{p}_{j_2} \big)}{\sum_{j_3=1}^{D} \exp\big( \mathbf{x}_{j_3}^\top \mathbf{W}_{1,1}^{(t)} \mathbf{x}_{j_2} + \mathbf{p}_{j_3}^\top \mathbf{W}_{2,2}^{(t)} \mathbf{p}_{j_2} \big)}$$

$$= \frac{\exp\big( y \cdot \mathbf{x}_{j_1}^\top \mathbf{W}_{1,1}^{(t)} y \cdot \mathbf{x}_{j_2} + \mathbf{p}_{j_1}^\top \mathbf{W}_{2,2}^{(t)} \mathbf{p}_{j_2} \big)}{\sum_{j_3=1}^{D} \exp\big( y \cdot \mathbf{x}_{j_3}^\top \mathbf{W}_{1,1}^{(t)} y \cdot \mathbf{x}_{j_2} + \mathbf{p}_{j_3}^\top \mathbf{W}_{2,2}^{(t)} \mathbf{p}_{j_2} \big)}.$$

If we omit all non-random components, we have $\mathbf{S}_{j_1,j_2}^{(t)} = \mathbf{S}_{j_1,j_2}^{(t)}(y \cdot \mathbf{x}_1, \cdots, y \cdot \mathbf{x}_D)$. Moreover, by Lemma E.2, we obtain that $\mathbf{S}_{j_1,j_2}^{(t)}$ is independent with $y$. $\square$

**Lemma C.7.** If $\mathbf{W}_{1,2}^{(t)}, \mathbf{W}_{2,1}^{(t)} = \mathbf{0}$ and $\mathbf{v}_2^{(t)} = \mathbf{0}$, then $\ell'^{(t)}$ can be expressed as a function of $\{y \cdot \mathbf{x}_1, \cdots, y \cdot \mathbf{x}_D\}$, and is independent with $y$.

*Proof of Lemma C.7.* When $\mathbf{v}_2^{(t)} = \mathbf{0}$, we express $\ell'^{(t)}$ as,

$$\ell'^{(t)} = \ell'(yf(\mathbf{Z}, \mathbf{W}^{(t)}, \mathbf{v}^{(t)})) = -\frac{1}{1 + \exp\left(yf(\mathbf{Z}, \mathbf{W}^{(t)}, \mathbf{v}^{(t)})\right)} = -\frac{1}{1 + \exp\left(\sum_{j=1}^{D}\langle \mathbf{v}_1^{(t)}, y\mathbf{x}_j\rangle \sum_{j'=1}^{D} \mathbf{S}_{j,j'}^{(t)}\right)}.$$

By Lemma C.6, $\mathbf{S}_{j,j'}^{(t)}$ is a function of $\{y \cdot \mathbf{x}_1, \cdots, y \cdot \mathbf{x}_D\}$ for all $j, j' \in [D]$. Therefore, $\ell'^{(t)}$ can also be expressed as a function of $\{y \cdot \mathbf{x}_1, \cdots, y \cdot \mathbf{x}_D\}$ when omitting all non-random components. By applying Lemma E.2, we have $\ell'^{(t)}$ is independent with $y$. $\square$

**Lemma C.8.** If $\mathbf{W}_{1,2}^{(t)}, \mathbf{W}_{2,1}^{(t)}, \mathbf{v}_2^{(t)} = \mathbf{0}$, $\mathbf{W}_{1,1}^{(t)} = a\mathbf{v}^*\mathbf{v}^{*\top}$ and $\mathbf{v}_1^{(t)} = b\mathbf{v}^*$ for some scalar $a, b$, then $\mathbf{S}_{j_1,j_2}^{(t)}$ and $\ell'^{(t)}$ can be expressed as functions of $\{y\langle\mathbf{v}^*, \mathbf{x}_1\rangle, \cdots, y\langle\mathbf{v}^*, \mathbf{x}_D\rangle\}$ for all $j_1, j_2 \in [D]$, and they are independent with $y\langle\boldsymbol{\xi}_i, \mathbf{x}_j\rangle$ for all $i \in \{2, \cdots, d\}$ and $j \in [D]$.

*Proof of Lemma C.8.* When $\mathbf{W}_{1,2}^{(t)}, \mathbf{W}_{2,1}^{(t)} = \mathbf{0}$ and $\mathbf{W}_{1,1}^{(t)} = a\mathbf{v}^*\mathbf{v}^{*\top}$, similar to the proof of Lemma C.6, we can re-write $\mathbf{S}_{j_1,j_2}$ as

$$\begin{aligned}
\mathbf{S}_{j_1,j_2}^{(t)} &= \frac{\exp\left(\mathbf{z}_{j_1}^\top \mathbf{W}^{(t)} \mathbf{z}_{j_2}\right)}{\sum_{j_3=1}^{D}\exp\left(\mathbf{z}_{j_3}^\top \mathbf{W}^{(t)} \mathbf{z}_{j_2}\right)} = \frac{\exp\left(a\mathbf{x}_{j_1}^\top \mathbf{v}^*\mathbf{v}^{*\top}\mathbf{x}_{j_2} + \mathbf{p}_{j_1}^\top \mathbf{W}_{2,2}^{(t)}\mathbf{p}_{j_2}\right)}{\sum_{j_3=1}^{D}\exp\left(a\mathbf{x}_{j_3}^\top \mathbf{v}^*\mathbf{v}^{*\top}\mathbf{x}_{j_2} + \mathbf{p}_{j_3}^\top \mathbf{W}_{2,2}^{(t)}\mathbf{p}_{j_2}\right)} \\
&= \frac{\exp\left(ay\langle\mathbf{v}^*, \mathbf{x}_{j_1}\rangle y\langle\mathbf{v}^*, \mathbf{x}_{j_2}\rangle + \mathbf{p}_{j_1}^\top \mathbf{W}_{2,2}^{(t)}\mathbf{p}_{j_2}\right)}{\sum_{j_3=1}^{D}\exp\left(ay\langle\mathbf{v}^*, \mathbf{x}_{j_3}\rangle y\langle\mathbf{v}^*, \mathbf{x}_{j_2}\rangle + \mathbf{p}_{j_3}^\top \mathbf{W}_{2,2}^{(t)}\mathbf{p}_{j_2}\right)}.
\end{aligned}$$

This is a function of $\{y\langle\mathbf{v}^*, \mathbf{x}_1\rangle, \cdots, y\langle\mathbf{v}^*, \mathbf{x}_D\rangle\}$ when omitting all non-random components. Besides, when $\mathbf{v}_2^{(t)} = \mathbf{0}$ and $\mathbf{v}_1 = b\mathbf{v}^*$, we express $\ell'^{(t)}$ as,

$$\ell'^{(t)} = \ell'(yf(\mathbf{Z}, \mathbf{W}^{(t)}, \mathbf{v}^{(t)})) = -\frac{1}{1 + \exp\left(yf(\mathbf{Z}, \mathbf{W}^{(t)}, \mathbf{v}^{(t)})\right)} = -\frac{1}{1 + \exp\left(b\sum_{j=1}^{D} y\langle\mathbf{v}^*, \mathbf{x}_j\rangle \sum_{j'=1}^{D} \mathbf{S}_{j,j'}^{(t)}\right)}.$$

Since we have demonstrated that $\mathbf{S}_{j_1,j_2}^{(t)}$ is a function of $\{y\langle\mathbf{v}^*, \mathbf{x}_1\rangle, \cdots, y\langle\mathbf{v}^*, \mathbf{x}_D\rangle\}$ for all $j, j' \in [D]$. Therefore, $\ell'^{(t)}$ can also be expressed as a function of $\{y\langle\mathbf{v}^*, \mathbf{x}_1\rangle, \cdots, y\langle\mathbf{v}^*, \mathbf{x}_D\rangle\}$ when omitting all non-random components. By Lemma E.1, we obtain that $y\langle\mathbf{v}^*, \mathbf{x}_j\rangle$ is independent with $y$ for all $j \in [D]$. Furthermore, by the orthogonality among $\mathbf{v}^*$ and $\boldsymbol{\xi}_2, \cdots, \boldsymbol{\xi}_d$, we can also obtain that $y\langle\mathbf{v}^*, \mathbf{x}_j\rangle$ is independent with $\langle\boldsymbol{\xi}_i, \mathbf{x}_j\rangle$ for all $i \in \{2, \cdots, d\}$. While for $j \neq j'$, $y\langle\mathbf{v}^*, \mathbf{x}_j\rangle$ is independent with $\langle\boldsymbol{\xi}_i, \mathbf{x}_{j'}\rangle$ since $\mathbf{x}_j$ is independent with $\mathbf{x}_{j'}$. Combining all the preceding results, we conclude that $y\langle\mathbf{v}^*, \mathbf{x}_j\rangle$ is independent with $y\langle\boldsymbol{\xi}_i, \mathbf{x}_{j'}\rangle$ for all $i \in \{2, \cdots, d\}$ and $j, j' \in [D]$. Since $\mathbf{S}_{j_1,j_2}^{(t)}$'s, $\ell'^{(t)}$ are functions of $\{y\langle\mathbf{v}^*, \mathbf{x}_1\rangle, \cdots, y\langle\mathbf{v}^*, \mathbf{x}_D\rangle\}$, we finally prove that they are independent with all $y\langle\boldsymbol{\xi}_i, \mathbf{x}_j\rangle$'s.

$\square$

Now, we are ready to prove Proposition C.1.

*Proof of Proposition C.1.* Since the iterates of $\mathbf{v}^{(t)}$ and $\mathbf{W}^{(t)}$ start with $\mathbf{v}^{(0)} = \mathbf{0}$ and $\mathbf{W}^{(0)} = \mathbf{0}$, it suffices to show that $\nabla_{\mathbf{v}_2}\mathcal{L}(\mathbf{v}^{(t)}, \mathbf{W}^{(t)}) = \mathbf{0}$, $\nabla_{\mathbf{W}_{1,2}}\mathcal{L}(\mathbf{v}^{(t)}, \mathbf{W}^{(t)}) = \mathbf{0}$ and $\nabla_{\mathbf{W}_{2,1}}\mathcal{L}(\mathbf{v}^{(t)}, \mathbf{W}^{(t)}) = \mathbf{0}$ given $\mathbf{v}_2^{(t)} = \mathbf{0}$, $\mathbf{W}_{1,2}^{(t)} = \mathbf{0}$ and $\mathbf{W}_{2,1}^{(t)} = \mathbf{0}$ by inductions. We first prove that $\nabla_{\mathbf{v}_2}\mathcal{L}(\mathbf{v}^{(t)}, \mathbf{W}^{(t)}) = \mathbf{0}$. By (C.1), we have

$$\nabla_{\mathbf{v}_2}\mathcal{L}(\mathbf{v}^{(t)}, \mathbf{W}^{(t)}) = \sum_{j=1}^{D}\sum_{j'=1}^{D}\mathbb{E}\left[y\ell'^{(t)}\mathbf{S}_{j',j}^{(t)}\right]\mathbf{p}_j = \sum_{j=1}^{D}\sum_{j'=1}^{D}\mathbb{E}[y]\mathbb{E}\left[\ell'^{(t)}\mathbf{S}_{j',j}^{(t)}\right]\mathbf{p}_j = \mathbf{0}.$$

The second equality holds because $\ell'^{(t)}\mathbf{S}_{j',j}^{(t)}$ is a function of $\{y \cdot \mathbf{x}_1, \cdots, y \cdot \mathbf{x}_D\}$ for all $j, j' \in [D]$ by Lemma C.6 and Lemma C.7 when $\mathbf{v}_2^{(t)} = \mathbf{0}$, $\mathbf{W}_{1,2}^{(t)} = \mathbf{0}$ and $\mathbf{W}_{2,1}^{(t)} = \mathbf{0}$. Then by Lemma E.2, we have $y$ is independent with $\ell'^{(t)}\mathbf{S}_{j',j}^{(t)}$. And the last equality holds since $\mathbb{E}[y] = 0$. Next, we prove

that $\nabla_{\mathbf{W}_{1,2}}\mathcal{L}(\mathbf{v}^{(t)}, \mathbf{W}^{(t)}) = \mathbf{0}$, and we skip the proof for $\nabla_{\mathbf{W}_{2,1}}\mathcal{L}(\mathbf{v}^{(t)}, \mathbf{W}^{(t)})$ since it's similar. By (C.2), we have

$$
\begin{aligned}
\nabla_{\mathbf{W}_{1,2}}\mathcal{L}(\mathbf{v}^{(t)}, \mathbf{W}^{(t)}) &= \sum_{j=1}^{D}\sum_{j'=1}^{D}\sum_{j''\neq j'} \mathbb{E}\Big[y\ell'^{(t)}\mathbf{S}_{j',j}^{(t)}\mathbf{S}_{j'',j}^{(t)}\langle\mathbf{v}^{(t)},\mathbf{z}_{j'}\rangle(\mathbf{x}_{j'}-\mathbf{x}_{j''})\Big]\mathbf{p}_j^\top \\
&= \sum_{j=1}^{D}\sum_{j'=1}^{D}\sum_{j''\neq j'} \mathbb{E}\Big[y\ell'^{(t)}\mathbf{S}_{j',j}^{(t)}\mathbf{S}_{j'',j}^{(t)}\langle\mathbf{v}_1^{(t)}, y\mathbf{x}_{j'}\rangle(y\mathbf{x}_{j'}-y\mathbf{x}_{j''})\Big]\mathbf{p}_j^\top \\
&= \sum_{j=1}^{D}\sum_{j'=1}^{D}\sum_{j''\neq j'} \mathbb{E}[y]\mathbb{E}\Big[\underbrace{\ell'^{(t)}\mathbf{S}_{j',j}^{(t)}\mathbf{S}_{j'',j}^{(t)}\langle\mathbf{v}_1^{(t)}, y\mathbf{x}_{j'}\rangle(y\mathbf{x}_{j'}-y\mathbf{x}_{j''})}_{I^{(t)}}\Big]\mathbf{p}_j^\top = \mathbf{0}.
\end{aligned}
$$

The second equality holds because $\langle\mathbf{v}^{(t)},\mathbf{z}_{j'}\rangle = \langle\mathbf{v}_1^{(t)},\mathbf{x}_{j'}\rangle$ when $\mathbf{v}_2^{(t)} = \mathbf{0}$, and $y^2 = 1$. The third equality holds since $I^{(t)}$ is a function of $\{y\cdot\mathbf{x}_1,\cdots,y\cdot\mathbf{x}_D\}$ for all $j, j', j'' \in [D]$ by Lemma C.6 and Lemma C.7 when $\mathbf{v}_2^{(t)} = \mathbf{0}$, $\mathbf{W}_{1,2}^{(t)} = \mathbf{0}$ and $\mathbf{W}_{2,1}^{(t)} = \mathbf{0}$. Similarly by Lemma E.2, we have $y$ is independent with $I^{(t)}$. The last equality holds since $\mathbb{E}[y] = 0$. This finishes the proof. $\square$

## C.3 CALCULATIONS FOR INITIAL ITERATIONS

Since we have demonstrated that $\mathbf{v}_2^{(t)}, \mathbf{W}_{1,2}^{(t)}, \mathbf{W}_{2,1}^{(t)} = \mathbf{0}$ throughout the training process in Proposition C.1, we will focus our analysis solely on the iterates $\mathbf{v}_1^{(t)}, \mathbf{W}_{1,1}^{(t)}$ and $\mathbf{W}_{2,2}^{(t)}$. Before we further present the global characterization of $\mathbf{v}_1^{(t)}, \mathbf{W}_{1,1}^{(t)}$ and $\mathbf{W}_{2,2}^{(t)}$, we first calculate several initial iterates of these parameters in the following lemmas.

**Lemma C.9.** Under the same condition with Theorem 2.2, the iterates $\mathbf{v}_1^{(t)}, \mathbf{W}^{(t)}$ of gradient descent defined in (2.3) and (2.4) satisfy that $\mathbf{W}^{(1)} = \mathbf{0}$ and $\mathbf{v}_1^{(1)} = \alpha^{(1)}\mathbf{v}^*$ with $\alpha^{(1)} = \frac{\eta\sigma_x}{\sqrt{2\pi}}$.

*Proof of Lemma C.9.* At $t = 0$, we have $\langle\mathbf{v}_1^{(0)},\mathbf{z}_{j'}\rangle$ for all $j' \in [D]$ since $\mathbf{v}_1^{(0)} = \mathbf{0}$, and correspondingly $\nabla_{\mathbf{W}}\mathcal{L}(\mathbf{v}^{(0)}, \mathbf{W}^{(0)}) = \mathbf{0}$. Therefore, we still have $\mathbf{W}^{(1)} = \mathbf{0}$. On the other hand, we calculate the gradient of $\mathcal{L}(\mathbf{v}^{(t)}, \mathbf{W}^{(t)})$ w.r.t $\mathbf{v}_1$ at $t = 0$ as

$$
\nabla_{\mathbf{v}_1}\mathcal{L}(\mathbf{v}^{(0)}, \mathbf{W}^{(0)}) = \sum_{j=1}^{D}\sum_{j'=1}^{D} \mathbb{E}\big[\ell'^{(0)}y\mathbf{S}_{j,j'}^{(0)}\mathbf{x}_j\big] = -\frac{1}{2}\sum_{j=1}^{D}\mathbb{E}\big[y\mathbf{x}_j\big] = -\frac{1}{2}\mathbb{E}\big[y\mathbf{x}_{j^*}\big] = -\frac{\sigma_x}{\sqrt{2\pi}}\mathbf{v}^*
$$

The second equality holds since $\ell'^{(0)} = \frac{1}{2}$ by the fact $f(\mathbf{Z}, \mathbf{v}^{(0)}, \mathbf{W}^{(0)}) = 0$, and $\mathbf{S}_{j',j}^{(0)} = \frac{1}{D}$ for all $j, j' \in [D]$ by the fact $\mathbf{W}^{(0)} = \mathbf{0}$. The third equality is by the independence between $y$ and $\mathbf{x}_j$ for $j \neq j^*$. The last equality is from E.4. By this result, we obtain that

$$
\mathbf{v}_1^{(1)} = \mathbf{v}_1^{(0)} - \eta\nabla_{\mathbf{v}_1}\mathcal{L}(\mathbf{v}^{(0)}, \mathbf{W}^{(0)}) = \frac{\eta\sigma_x}{\sqrt{2\pi}}\mathbf{v}^* = \alpha^{(1)}\mathbf{v}^*.
$$

which completes the proof. $\square$

Next, we provide the calculation for the second iteration.

**Lemma C.10.** Under the same condition with Theorem 2.2, the iterates $\mathbf{v}_1^{(t)}$ of gradient descent defined in (2.3) satisfies that $\mathbf{v}_1^{(2)} = \alpha^{(2)}\mathbf{v}^*$ with $\alpha^{(2)} = -\Theta(D)$.

*Proof of Lemma C.10.* By the fact $\mathbf{W}^{(1)} = \mathbf{0}$, we still have $\mathbf{S}_{j',j}^{(1)} = \frac{1}{D}$ for all $j, j' \in [D]$. Therefore we can calculate the gradient w.r.t $\mathbf{v}_1^{(t)}$ as

$$
\nabla_{\mathbf{v}_1}\mathcal{L}(\mathbf{v}^{(1)}, \mathbf{W}^{(1)}) = \sum_{j=1}^{D}\sum_{j'=1}^{D} \mathbb{E}\big[\ell'^{(1)}y\mathbf{S}_{j,j'}^{(1)}\mathbf{x}_j\big] = \mathbb{E}\big[\ell'^{(1)}y\mathbf{x}_{j^*}\big] + \sum_{j\neq j^*}\mathbb{E}\big[\ell'^{(1)}y\mathbf{x}_j\big]
$$

We analyze the value of $\mathbb{E}[\ell'^{(1)}y\mathbf{x}_{j^*}]$ and $\mathbb{E}[\ell'^{(1)}y\mathbf{x}_j]$ respectively. For $\mathbb{E}[\ell'^{(1)}y\mathbf{x}_{j^*}]$, we have

$$
\begin{aligned}
\mathbb{E}[\ell'^{(1)}y\mathbf{x}_{j^*}] &= \mathbf{A}\mathbf{A}^\top \mathbb{E}[\ell'^{(1)}y\mathbf{x}_{j^*}] = \mathbf{A}\mathbb{E}\left[\ell'^{(1)}y\big[\langle\mathbf{v}^*,\mathbf{x}_{j^*}\rangle,\langle\boldsymbol{\xi}_2,\mathbf{x}_{j^*}\rangle,\cdots,\langle\boldsymbol{\xi}_d,\mathbf{x}_{j^*}\rangle\big]^\top\right] \\
&= \mathbf{A}\mathbb{E}\left[\ell'^{(1)}y\big[\langle\mathbf{v}^*,\mathbf{x}_{j^*}\rangle,\langle\boldsymbol{\xi}_2,\mathbf{x}_{j^*}\rangle,\cdots,\langle\boldsymbol{\xi}_d,\mathbf{x}_{j^*}\rangle\big]^\top\right] \\
&= \mathbb{E}[\ell'^{(1)}y\langle\mathbf{v}^*,\mathbf{x}_{j^*}\rangle]\mathbf{v}^* + \sum_{i=2}^d \mathbb{E}[\ell'^{(1)}y\langle\boldsymbol{\xi}_i,\mathbf{x}_{j^*}\rangle]\boldsymbol{\xi}_i \\
&= \mathbb{E}[\ell'^{(1)}y\langle\mathbf{v}^*,\mathbf{x}_{j^*}\rangle]\mathbf{v}^* + \sum_{i=2}^d \mathbb{E}[\ell'^{(1)}y]\mathbb{E}[\langle\boldsymbol{\xi}_i,\mathbf{x}_{j^*}\rangle]\boldsymbol{\xi}_i = \mathbb{E}[\ell'^{(1)}y\langle\mathbf{v}^*,\mathbf{x}_{j^*}\rangle]\mathbf{v}^*,
\end{aligned}
$$

where $\mathbf{A}$ is the orthogonal matrix defined in (C.3). The penultimate equality holds since $\ell'^{(1)} = \frac{1}{1+\exp\left(\alpha^{(1)}\sum_{j=1}^D \langle\mathbf{v}^*,y\mathbf{x}_j\rangle\right)}$. By replacing $y$ with $\text{sign}\left(\langle\mathbf{v}^*,\mathbf{x}_{j^*}\rangle\right)$, we can notice that $y\ell'^{(1)}$ only contains the projection of $\mathbf{x}_{j^*}$ on the direction of $\mathbf{v}^*$, i.e., $\langle\mathbf{v}^*,\mathbf{x}_{j^*}\rangle$. Hence by the orthogonality among $\mathbf{v}^*$ and $\boldsymbol{\xi}_2,\cdots,\boldsymbol{\xi}_d$ and properties of Gaussian distribution, we have $\langle\boldsymbol{\xi}_i,\mathbf{x}_{j^*}\rangle$ is independent with $y\ell'^{(1)}$ for all $i \in \{2,\cdots,d\}$. The last equality is simply by $\mathbb{E}[\langle\boldsymbol{\xi}_i,\mathbf{x}_{j^*}\rangle] = 0$ for all $i \in \{2,\cdots,d\}$. Through a similar process, we can also derive that

$$
\mathbb{E}[\ell'^{(1)}y\mathbf{x}_j] = \mathbb{E}[\ell'^{(1)}y\langle\mathbf{v}^*,\mathbf{x}_j\rangle]\mathbf{v}^*
$$

for all $j \neq j^*$. Moreover, we could notice that $\ell'^{(1)}y\mathbf{x}_j$ have the same distribution for all $j \neq j^*$, and correspondingly $\mathbb{E}[\ell'^{(1)}y\langle\mathbf{v}^*,\mathbf{x}_j\rangle]$ take the same value for all $j \neq j^*$. By carefully checking the distribution of $\ell'^{(1)}y\langle\mathbf{v}^*,\mathbf{x}_{j^*}\rangle$ and $\ell'^{(1)}y\langle\mathbf{v}^*,\mathbf{x}_j\rangle$ with $j \neq j^*$, we notice that Lemma E.10 and Lemma E.11 apply to the calculation of $\mathbb{E}[\ell'^{(1)}y\langle\mathbf{v}^*,\mathbf{x}_{j^*}\rangle]$ and $\mathbb{E}[\ell'^{(1)}y\langle\mathbf{v}^*,\mathbf{x}_j\rangle]$ with $j \neq j^*$. We can derive that

$$
\nabla_{\mathbf{v}_1}\mathcal{L}(\mathbf{v}^{(1)},\mathbf{W}^{(1)}) = \left(\mathbb{E}[\ell'^{(1)}y\langle\mathbf{v}^*,\mathbf{x}_{j^*}\rangle] + \sum_{j\neq j^*}\mathbb{E}[\ell'^{(1)}y\langle\mathbf{v}^*,\mathbf{x}_j\rangle]\right)\mathbf{v}^*.
$$

And for the coefficients of $\mathbf{v}^*$, we have

$$
\mathbb{E}[\ell'^{(1)}y\langle\mathbf{v}^*,\mathbf{x}_{j^*}\rangle] + \sum_{j\neq j^*}\mathbb{E}[\ell'^{(1)}y\langle\mathbf{v}^*,\mathbf{x}_j\rangle] \geq -\sigma_x\sqrt{\frac{2}{\pi}} + \frac{1}{3}\sqrt{\frac{2D}{e\pi^3}}\exp\left(-\frac{\eta\sigma_x^2}{\sqrt{2\pi}} - \frac{2\pi}{\eta^2\sigma_x^4}\right) \geq \Theta(\sqrt{D}),
$$

and

$$
\mathbb{E}[\ell'^{(1)}y\langle\mathbf{v}^*,\mathbf{x}_{j^*}\rangle] + \sum_{j\neq j^*}\mathbb{E}[\ell'^{(1)}y\langle\mathbf{v}^*,\mathbf{x}_j\rangle] \leq -\frac{\sigma_x}{2\sqrt{2e\pi}}e^{-\frac{\eta\sigma_x^2}{2\pi}} + \frac{4(1+\eta\sigma_x^2)}{\eta\sigma_x}\sqrt{\frac{D}{\pi}}\exp\left(\frac{\eta^2\sigma_x^4}{2\pi}\right) \leq \Theta(\sqrt{D}).
$$

Applying these results into the gradient descent iteration of $\mathbf{v}_1^{(t)}$, we have

$$
\mathbf{v}_1^{(2)} = \mathbf{v}_1^{(1)} - \eta\left(\mathbb{E}[\ell'^{(1)}y\langle\mathbf{v}^*,\mathbf{x}_{j^*}\rangle] + \sum_{j\neq j^*}\mathbb{E}[\ell'^{(1)}y\langle\mathbf{v}^*,\mathbf{x}_j\rangle]\right)\mathbf{v}^* = \alpha^{(2)}\mathbf{v}^*,
$$

where $\alpha^{(2)} = -\Theta(\sqrt{D})$. This completes the proof. □

**Lemma C.11.** Under the same condition with Theorem 2.2, the iterates $\mathbf{W}_{1,1}^{(t)}$ of gradient descent defined in (2.4) satisfies that $\mathbf{W}_{1,1}^{(2)} = \beta_1\mathbf{v}^*\mathbf{v}^{*\top}$. The coefficient $\beta_1$ satisfy that $|\beta_1| \leq c_1\sqrt{D}$ for some non-negative constant $c_1$ solely depending on $\eta$ and $\sigma_x$.

*Proof of Lemma C.11.* We first demonstrate the calculation details of $\nabla_{\mathbf{W}_{1,1}}\mathcal{L}(\mathbf{v}^{(1)},\mathbf{W}^{(1)})$ Since $\mathbf{v}_1^{(1)} = \alpha^{(1)}\mathbf{v}^*$ and $\mathbf{v}_2^{(1)} = \mathbf{0}$, we always have $\langle\mathbf{v}^{(1)},\mathbf{z}_j\rangle = \alpha^{(1)}\langle\mathbf{v}^*,\mathbf{x}_j\rangle$ in the following calculations. For $\nabla_{\mathbf{W}_{1,1}}\mathcal{L}(\mathbf{v}^{(1)},\mathbf{W}^{(1)})$, we can obtain that

$$
\nabla_{\mathbf{W}_{1,1}}\mathcal{L}(\mathbf{v}^{(1)},\mathbf{W}^{(1)}) = \mathbb{E}\left[\ell'^{(1)}\cdot y\cdot\sum_{j=1}^D\sum_{j'=1}^D\sum_{j''\neq j'}\mathbf{S}_{j',j}^{(1)}\mathbf{S}_{j'',j}^{(1)}\langle\mathbf{v}^{(1)},\mathbf{z}_{j'}\rangle(\mathbf{x}_{j'}-\mathbf{x}_{j''})\mathbf{x}_j^\top\right]
$$

$$= \underbrace{\frac{\alpha^{(1)}(D-1)}{D^2} \sum_{j=1}^{D} \sum_{j'=1}^{D} \mathbb{E}\big[\ell'^{(1)} \cdot y \cdot \langle \mathbf{v}^*, \mathbf{x}_{j'}\rangle \mathbf{x}_{j'} \mathbf{x}_j^\top \big]}_{I_1} - \underbrace{\frac{\alpha^{(1)}}{D^2} \sum_{j=1}^{D} \sum_{j'=1}^{D} \sum_{j''\neq j'} \mathbb{E}\big[\ell'^{(1)} \cdot y \cdot \langle \mathbf{v}^*, \mathbf{x}_{j'}\rangle \mathbf{x}_{j''} \mathbf{x}_j^\top \big]}_{I_2}.$$

We analyze the value of $I_1$ and $I_2$ respectively in the following. For $I_1$, we can obtain that

$$I_1 = \frac{\alpha^{(1)}(D-1)}{D^2} \sum_{j=1}^{D} \sum_{j'=1}^{D} \mathbf{A} \underbrace{\mathbb{E}\left[\ell'^{(1)} y \langle \mathbf{v}^*, \mathbf{x}_{j'}\rangle \begin{bmatrix} \langle \mathbf{v}^*, \mathbf{x}_{j'}\rangle \\ \langle \boldsymbol{\xi}_2, \mathbf{x}_{j'}\rangle \\ \vdots \\ \langle \boldsymbol{\xi}_d, \mathbf{x}_{j'}\rangle \end{bmatrix} \big[\langle \mathbf{v}^*, \mathbf{x}_j\rangle, \langle \boldsymbol{\xi}_2, \mathbf{x}_j\rangle, \cdots, \langle \boldsymbol{\xi}_d, \mathbf{x}_j\rangle\big] \right]}_{\mathbf{B}^{j,j'} \in \mathbb{R}^{d \times d}} \mathbf{A}^\top$$

We denote the entry in the $i_1$-th row and $i_2$-th column of the expectation matrix $\mathbf{B}^{j,j'}$ as $\mathbf{B}^{j,j'}_{i_1,i_2}$. By utilizing Lemma C.8, we can examine the value of $\mathbf{B}^{j,j'}_{i_1,i_2}$ for two cases: $j = j'$ and $j \neq j'$.

**Case I:** $j = j'$.

1. $\mathbf{B}^{j,j}_{1,1} = \mathbb{E}\big[\ell'^{(1)} y \langle \mathbf{v}^*, \mathbf{x}_j\rangle^3\big]$.

2. $\mathbf{B}^{j,j}_{i_1,i_1} = \mathbb{E}\big[\ell'^{(1)} y \langle \mathbf{v}^*, \mathbf{x}_j\rangle \langle \boldsymbol{\xi}_{i_1}, \mathbf{x}_j\rangle^2\big] = \mathbb{E}\big[\ell'^{(1)} y \langle \mathbf{v}^*, \mathbf{x}_j\rangle\big] \mathbb{E}\big[\langle \boldsymbol{\xi}_{i_1}, \mathbf{x}_j\rangle^2\big] = \sigma_x^2 \mathbb{E}\big[\ell'^{(1)} y \langle \mathbf{v}^*, \mathbf{x}_j\rangle\big]$, for all $i_1 \neq 1$.

3. $\mathbf{B}^{j,j}_{1,i_2} = \mathbb{E}\big[\ell'^{(1)} y \langle \mathbf{v}^*, \mathbf{x}_j\rangle^2 \langle \boldsymbol{\xi}_{i_2}, \mathbf{x}_j\rangle\big] = \mathbb{E}\big[\ell'^{(1)} y \langle \mathbf{v}^*, \mathbf{x}_j\rangle^2\big] \mathbb{E}\big[\langle \boldsymbol{\xi}_{i_2}, \mathbf{x}_j\rangle\big] = 0$, for all $i_2 \neq 1$.

4. $\mathbf{B}^{j,j}_{i_1,1} = \mathbb{E}\big[\ell'^{(1)} y \langle \mathbf{v}^*, \mathbf{x}_j\rangle^2 \langle \boldsymbol{\xi}_{i_1}, \mathbf{x}_j\rangle\big] = \mathbb{E}\big[\ell'^{(1)} y \langle \mathbf{v}^*, \mathbf{x}_j\rangle^2\big] \mathbb{E}\big[\langle \boldsymbol{\xi}_{i_1}, \mathbf{x}_j\rangle\big] = 0$, for all $i_1 \neq 1$.

5. $\mathbf{B}^{j,j}_{i_1,i_2} = \mathbb{E}\big[\ell'^{(1)} y \langle \mathbf{v}^*, \mathbf{x}_j\rangle \langle \boldsymbol{\xi}_{i_1}, \mathbf{x}_j\rangle \langle \boldsymbol{\xi}_{i_2}, \mathbf{x}_j\rangle\big] = \mathbb{E}\big[\ell'^{(1)} y \langle \mathbf{v}^*, \mathbf{x}_j\rangle\big] \mathbb{E}\big[\langle \boldsymbol{\xi}_{i_1}, \mathbf{x}_j\rangle\big] \mathbb{E}\big[\langle \boldsymbol{\xi}_{i_2}, \mathbf{x}_j\rangle\big] = 0$, for all $i_1, i_2 \neq 1$ and $i_1 \neq i_2$.

**Case II:** $j \neq j'$.

1. $\mathbf{B}^{j,j'}_{1,1} = \mathbb{E}\big[\ell'^{(1)} y \langle \mathbf{v}^*, \mathbf{x}_{j'}\rangle^2 \langle \mathbf{v}^*, \mathbf{x}_j\rangle\big]$.

2. $\mathbf{B}^{j,j'}_{1,i_2} = \mathbb{E}\big[\ell'^{(1)} y \langle \mathbf{v}^*, \mathbf{x}_{j'}\rangle^2 \langle \boldsymbol{\xi}_{i_2}, \mathbf{x}_j\rangle\big] = \mathbb{E}\big[\ell'^{(1)} y \langle \mathbf{v}^*, \mathbf{x}_{j'}\rangle^2\big] \mathbb{E}\big[\langle \boldsymbol{\xi}_{i_2}, \mathbf{x}_j\rangle\big] = 0$, for all $i_2 \neq 1$.

3. $\mathbf{B}^{j,j'}_{i_1,1} = \mathbb{E}\big[\ell'^{(1)} y \langle \mathbf{v}^*, \mathbf{x}_{j'}\rangle \langle \boldsymbol{\xi}_{i_1}, \mathbf{x}_{j'}\rangle \langle \mathbf{v}^*, \mathbf{x}_j\rangle\big] = \mathbb{E}\big[\ell'^{(1)} y \langle \mathbf{v}^*, \mathbf{x}_{j'}\rangle \langle \mathbf{v}^*, \mathbf{x}_j\rangle\big] \mathbb{E}\big[\langle \boldsymbol{\xi}_{i_1}, \mathbf{x}_{j'}\rangle\big] = 0$, for all $i_1 \neq 1$.

4. $\mathbf{B}^{j,j'}_{i_1,i_2} = \mathbb{E}\big[\ell'^{(1)} y \langle \mathbf{v}^*, \mathbf{x}_{j'}\rangle \langle \boldsymbol{\xi}_{i_1}, \mathbf{x}_{j'}\rangle \langle \boldsymbol{\xi}_{i_2}, \mathbf{x}_j\rangle\big] = \mathbb{E}\big[\ell'^{(1)} y \langle \mathbf{v}^*, \mathbf{x}_{j'}\rangle\big] \mathbb{E}\big[\langle \boldsymbol{\xi}_{i_1}, \mathbf{x}_{j'}\rangle\big] \mathbb{E}\big[\langle \boldsymbol{\xi}_{i_2}, \mathbf{x}_j\rangle\big] = 0$, for all $i_1, i_2 \neq 1$.

By previous discussion of $\mathbf{B}^{j,j'}_{i_1,i_2}$, we derive that

$$I_1 = \frac{\alpha^{(1)}(D-1)}{D^2} \sum_{j=1}^{D} \sum_{j'=1}^{D} \mathbb{E}\big[\ell'^{(1)} y \langle \mathbf{v}^*, \mathbf{x}_{j'}\rangle^2 \langle \mathbf{v}^*, \mathbf{x}_j\rangle\big] \mathbf{v}^* \mathbf{v}^{*\top} + \frac{\alpha^{(1)}(D-1)\sigma_x^2}{D^2} \sum_{i=2}^{d} \sum_{j=1}^{D} \mathbb{E}\big[\ell'^{(1)} y \langle \mathbf{v}^*, \mathbf{x}_j\rangle\big] \boldsymbol{\xi}_i \boldsymbol{\xi}_i^\top.$$

$$(\text{C.10})$$

Similarly, for $I_2$, we have

$$I_2 = \frac{\alpha^{(1)}}{D^2} \sum_{j=1}^{D} \sum_{j'=1}^{D} \sum_{j''\neq j'} \mathbf{A} \underbrace{\mathbb{E}\left[\ell'^{(1)} y \langle \mathbf{v}^*, \mathbf{x}_{j'}\rangle \begin{bmatrix} \langle \mathbf{v}^*, \mathbf{x}_{j''}\rangle \\ \langle \boldsymbol{\xi}_2, \mathbf{x}_{j''}\rangle \\ \vdots \\ \langle \boldsymbol{\xi}_d, \mathbf{x}_{j''}\rangle \end{bmatrix} \big[\langle \mathbf{v}^*, \mathbf{x}_j\rangle, \langle \boldsymbol{\xi}_2, \mathbf{x}_j\rangle, \cdots, \langle \boldsymbol{\xi}_d, \mathbf{x}_j\rangle\big] \right]}_{\mathbf{C}^{j,j',j''} \in \mathbb{R}^{d \times d}} \mathbf{A}^\top$$

We denote the entry in the $i_1$-th row and $i_2$-th column of the expectation matrix $\mathbf{C}^{j,j',j''}$ as $\mathbf{C}^{j,j',j''}_{i_1,i_2}$. And we examine the value of $\mathbf{C}^{j,j',j''}_{i_1,i_2}$ for two cases: $j = j''$ and $j \neq j''$.

**Case I:** $j = j''$.

1. $\mathbf{C}^{j,j',j}_{1,1} = \mathbb{E}\big[\ell'^{(1)}y\langle\mathbf{v}^*,\mathbf{x}_j\rangle^2\langle\mathbf{v}^*,\mathbf{x}_{j'}\rangle\big]$.

2. $\mathbf{C}^{j,j',j}_{i_1,i_1} = \mathbb{E}\big[\ell'^{(1)}y\langle\mathbf{v}^*,\mathbf{x}_{j'}\rangle\langle\boldsymbol{\xi}_{i_1},\mathbf{x}_j\rangle^2\big] = \mathbb{E}\big[\ell'^{(1)}y\langle\mathbf{v}^*,\mathbf{x}_{j'}\rangle\big]\mathbb{E}\big[\langle\boldsymbol{\xi}_{i_1},\mathbf{x}_j\rangle^2\big] = \sigma_x^2\mathbb{E}\big[\ell'^{(1)}y\langle\mathbf{v}^*,\mathbf{x}_{j'}\rangle\big]$, for all $i_1 \neq 1$.

3. $\mathbf{C}^{j,j',j}_{1,i_2} = \mathbb{E}\big[\ell'^{(1)}y\langle\mathbf{v}^*,\mathbf{x}_{j'}\rangle\langle\mathbf{v}^*,\mathbf{x}_j\rangle\langle\boldsymbol{\xi}_{i_2},\mathbf{x}_j\rangle\big] = \mathbb{E}\big[\ell'^{(1)}y\langle\mathbf{v}^*,\mathbf{x}_{j'}\rangle\langle\mathbf{v}^*,\mathbf{x}_j\rangle\big]\mathbb{E}\big[\langle\boldsymbol{\xi}_{i_2},\mathbf{x}_j\rangle\big] = 0$, for all $i_2 \neq 1$.

4. $\mathbf{C}^{j,j',j}_{i_1,1} = \mathbb{E}\big[\ell'^{(1)}y\langle\mathbf{v}^*,\mathbf{x}_{j'}\rangle\langle\mathbf{v}^*,\mathbf{x}_j\rangle\langle\boldsymbol{\xi}_{i_1},\mathbf{x}_j\rangle\big] = \mathbb{E}\big[\ell'^{(1)}y\langle\mathbf{v}^*,\mathbf{x}_j\rangle\langle\mathbf{v}^*,\mathbf{x}_j\rangle\big]\mathbb{E}\big[\langle\boldsymbol{\xi}_{i_1},\mathbf{x}_j\rangle\big] = 0$, for all $i_1 \neq 1$.

5. $\mathbf{C}^{j,j',j}_{i_1,i_2} = \mathbb{E}\big[\ell'^{(1)}y\langle\mathbf{v}^*,\mathbf{x}_{j'}\rangle\langle\boldsymbol{\xi}_{i_1},\mathbf{x}_j\rangle\langle\boldsymbol{\xi}_{i_2},\mathbf{x}_j\rangle\big] = \mathbb{E}\big[\ell'^{(1)}y\langle\mathbf{v}^*,\mathbf{x}_{j'}\rangle\big]\mathbb{E}\big[\langle\boldsymbol{\xi}_{i_1},\mathbf{x}_j\rangle\big]\mathbb{E}\big[\langle\boldsymbol{\xi}_{i_2},\mathbf{x}_j\rangle\big] = 0$, for all $i_1, i_2 \neq 1$ and $i_1 \neq i_2$.

**Case II:** $j \neq j'$.

1. $\mathbf{C}^{j,j',j''}_{1,1} = \mathbb{E}\big[\ell'^{(1)}y\langle\mathbf{v}^*,\mathbf{x}_{j'}\rangle\langle\mathbf{v}^*,\mathbf{x}_{j''}\rangle\langle\mathbf{v}^*,\mathbf{x}_j\rangle\big]$.

2. $\mathbf{C}^{j,j',j''}_{1,i_2} = \mathbb{E}\big[\ell'^{(1)}y\langle\mathbf{v}^*,\mathbf{x}_{j'}\rangle\langle\mathbf{v}^*,\mathbf{x}_{j''}\rangle\langle\boldsymbol{\xi}_{i_2},\mathbf{x}_j\rangle\big] = \mathbb{E}\big[\ell'^{(1)}y\langle\mathbf{v}^*,\mathbf{x}_{j'}\rangle\langle\mathbf{v}^*,\mathbf{x}_{j''}\rangle\big]\mathbb{E}\big[\langle\boldsymbol{\xi}_{i_2},\mathbf{x}_j\rangle\big] = 0$, for all $i_2 \neq 1$.

3. $\mathbf{C}^{j,j',j''}_{i_1,1} = \mathbb{E}\big[\ell'^{(1)}y\langle\mathbf{v}^*,\mathbf{x}_{j'}\rangle\langle\boldsymbol{\xi}_{i_1},\mathbf{x}_{j''}\rangle\langle\mathbf{v}^*,\mathbf{x}_j\rangle\big] = \mathbb{E}\big[\ell'^{(1)}y\langle\mathbf{v}^*,\mathbf{x}_{j'}\rangle\langle\mathbf{v}^*,\mathbf{x}_j\rangle\big]\mathbb{E}\big[\langle\boldsymbol{\xi}_{i_1},\mathbf{x}_{j''}\rangle\big] = 0$, for all $i_1 \neq 1$.

4. $\mathbf{C}^{j,j',j''}_{i_1,i_2} = \mathbb{E}\big[\ell'^{(1)}y\langle\mathbf{v}^*,\mathbf{x}_{j'}\rangle\langle\boldsymbol{\xi}_{i_1},\mathbf{x}_{j''}\rangle\langle\boldsymbol{\xi}_{i_2},\mathbf{x}_j\rangle\big] = \mathbb{E}\big[\ell'^{(1)}y\langle\mathbf{v}^*,\mathbf{x}_{j'}\rangle\big]\mathbb{E}\big[\langle\boldsymbol{\xi}_{i_1},\mathbf{x}_{j''}\rangle\big]\mathbb{E}\big[\langle\boldsymbol{\xi}_{i_2},\mathbf{x}_j\rangle\big] = 0$, for all $i_1, i_2 \neq 1$.

By previous discussion of $\mathbf{C}^{j,j',j''}_{i_1,i_2}$, we derive that

$$I_2 = \frac{\alpha^{(1)}}{D^2}\sum_{j=1}^{D}\sum_{j'=1}^{D}\sum_{j''\neq j'}\mathbb{E}\big[\ell'^{(1)}y\langle\mathbf{v}^*,\mathbf{x}_{j'}\rangle\langle\mathbf{v}^*,\mathbf{x}_{j''}\langle\mathbf{v}^*,\mathbf{x}_j\rangle\big]\mathbf{v}^*\mathbf{v}^{*\top} + \frac{\alpha^{(1)}(D-1)\sigma_x^2}{D^2}\sum_{i=2}^{d}\sum_{j=1}^{D}\mathbb{E}\big[\ell'^{(1)}y\langle\mathbf{v}^*,\mathbf{x}_j\rangle\big]\boldsymbol{\xi}_i\boldsymbol{\xi}_i^\top.$$
(C.11)

Notice that the coefficients of $\boldsymbol{\xi}_i\boldsymbol{\xi}_i^\top$ are all equal in both $I_1$ and $I_2$. Besides, we define two sets: $J_1 = \{(j,j')|j,j' \in [D]; j,j' \neq j^*, j \neq j'\}$ and $J_2 = \{(j,j',j'')|j,j',j'' \in [D]; j,j',j'' \neq j^*, j \neq j' \neq j''\}$. Then by using (C.10) minus (C.11), we get

$$\nabla_{\mathbf{W}_{1,1}}\mathcal{L}(\mathbf{v}^{(1)}, \mathbf{W}^{(1)}) = I_1 - I_2$$

$$= \frac{\alpha^{(1)}}{D^2}\bigg(\underbrace{(D-1)\sum_{j=1}^{D}\sum_{j'=1}^{D}\mathbb{E}\big[\ell'^{(1)}y\langle\mathbf{v}^*,\mathbf{x}_{j'}\rangle^2\langle\mathbf{v}^*,\mathbf{x}_j\rangle\big] - \sum_{j=1}^{D}\sum_{j'=1}^{D}\sum_{j''\neq j'}\mathbb{E}\big[\ell'^{(1)}y\langle\mathbf{v}^*,\mathbf{x}_{j'}\rangle\langle\mathbf{v}^*,\mathbf{x}_{j''}\langle\mathbf{v}^*,\mathbf{x}_j\rangle\big]}_{I_3}\bigg)\mathbf{v}^*\mathbf{v}^{*\top}$$

$$= \frac{\alpha^{(1)}}{D^2}\bigg(\underbrace{(D-1)\sum_{(j,j')\in J_1}\mathbb{E}\big[\ell'^{(1)}y\langle\mathbf{v}^*,\mathbf{x}_{j'}\rangle^2\langle\mathbf{v}^*,\mathbf{x}_j\rangle\big]}_{I_{3,1}} - \underbrace{\sum_{(j,j',j'')\in J_2}\mathbb{E}\big[\ell'^{(1)}y\langle\mathbf{v}^*,\mathbf{x}_{j'}\rangle\langle\mathbf{v}^*,\mathbf{x}_{j''}\langle\mathbf{v}^*,\mathbf{x}_j\rangle\big]}_{I_{3,2}}$$

$$+ (D-1) \sum_{(j,j') \notin J_1} \mathbb{E}\big[\ell'^{(1)} y \langle \mathbf{v}^*, \mathbf{x}_{j'} \rangle^2 \langle \mathbf{v}^*, \mathbf{x}_j \rangle\big] - \underbrace{\sum_{(j,j',j'') \notin J_2} \mathbb{E}\big[\ell'^{(1)} y \langle \mathbf{v}^*, \mathbf{x}_{j'} \rangle \langle \mathbf{v}^*, \mathbf{x}_{j''} \langle \mathbf{v}^*, \mathbf{x}_j \rangle\big]}_{I_{3,3}} \Big) \mathbf{v}^* \mathbf{v}^{*\top}.$$

By carefully checking the terms inner expectation of $I_{3,1}$, we can utilize Lemma E.12 to obtain that

$$-\sqrt{\frac{2}{\pi}} \frac{4(\eta^2 \sigma_x^2 + 1)\eta \sigma_x^5}{\pi} e^{\frac{3\eta^2 \sigma_x^4}{4\pi}} D^{5/2} \le I_{3,1} \le -\frac{1}{48\pi \eta^4 \sigma_x^5} e^{-\frac{\eta \sigma_x^2}{\sqrt{2\pi}} - \frac{2\pi}{\eta^2 \sigma_x^4} - \frac{1}{2}} D^{5/2}.$$

Similarly, we obtain that

$$-\frac{16}{\eta \sigma_x^2} e^{\frac{\eta^2 \sigma_x^4}{4\pi}} \left( \frac{\sigma_x}{\sqrt{2\pi}} + \frac{\eta \sigma_x^3 e^{\frac{\eta^2 \sigma_x^4}{4\pi}}}{\sqrt{2\pi}} \right)^3 D^{5/2} \le I_{3,2} \le \frac{16}{\eta \sigma_x^2} e^{\frac{\eta^2 \sigma_x^4}{4\pi}} \left( \frac{\sigma_x}{\sqrt{2\pi}} + \frac{\eta \sigma_x^3 e^{\frac{\eta^2 \sigma_x^4}{4\pi}}}{\sqrt{2\pi}} \right)^3 D^{5/2}$$

by Lemma E.13, and

$$|I_{3,3}| \le 6\sqrt{\frac{2}{\pi}} \sigma_x^3 D^2$$

by Lemma E.14. Applying all the preceding results to the gradient descent iteration of $\mathbf{W}_{1,1}^{(t)}$, we finally obtain that

$$\mathbf{W}_{1,1}^{(2)} = \mathbf{W}_{1,1}^{(1)} - \eta \nabla_{\mathbf{w}_{1,1}} \mathcal{L}(\mathbf{v}^{(1)}, \mathbf{W}^{(1)}) = \beta_1 \mathbf{v}^* \mathbf{v}^{*\top}.$$

And the coefficient $\beta_1$ satisfy that $|\beta_1\| \le c_1 \sqrt{D}$ for some non-negative constant $c_1$ solely depending on $\eta$ and $\sigma_x$. This completes the proof. □

**Lemma C.12.** Under the same condition with Theorem 2.2, the iterates $\mathbf{W}_{2,2}^{(t)}$ of gradient descent defined in (2.4) satisfies that

$$\mathbf{W}_{2,2}^{(2)} = \beta_2 \Big( \sum_{j \neq j^*} (\mathbf{p}_{j^*} - \mathbf{p}_j) \Big) \Big( \sum_{j=1}^{D} \mathbf{p}_j^\top \Big).$$

The coefficient $\beta_2$ satisfy that $\frac{c_2}{D^2} \le \beta_2 \le \frac{c_3}{D^2}$ for some non-negative constants $c_2, c_3$ solely depending on $\eta$ and $\sigma_x$.

*Proof of Lemma C.12.* □

For $\nabla_{\mathbf{W}_{2,2}} \mathcal{L}(\mathbf{v}^{(1)}, \mathbf{W}^{(1)})$, we can derive that

$$\nabla_{\mathbf{W}_{2,2}} \mathcal{L}(\mathbf{v}^{(1)}, \mathbf{W}^{(1)}) = \mathbb{E}\left[ \ell'^{(1)} \cdot y \cdot \sum_{j=1}^{D} \sum_{j'=1}^{D} \sum_{j'' \neq j'} \mathbf{S}_{j',j}^{(1)} \mathbf{S}_{j'',j}^{(1)} \langle \mathbf{v}^{(1)}, \mathbf{z}_{j'} \rangle (\mathbf{p}_{j'} - \mathbf{p}_{j''}) \mathbf{p}_j^\top \right]$$

$$= \underbrace{\frac{\alpha^{(1)}(D-1)}{D^2} \sum_{j=1}^{D} \sum_{j'=1}^{D} \mathbb{E}\big[\ell'^{(1)} \cdot y \cdot \langle \mathbf{v}^*, \mathbf{x}_{j'} \rangle\big] \mathbf{p}_{j'} \mathbf{p}_j^\top}_{I_1} - \underbrace{\frac{\alpha^{(1)}}{D^2} \sum_{j=1}^{D} \sum_{j'=1}^{D} \sum_{j'' \neq j'} \mathbb{E}\big[\ell'^{(1)} \cdot y \cdot \langle \mathbf{v}^*, \mathbf{x}_{j'} \rangle\big] \mathbf{p}_{j''} \mathbf{p}_j^\top}_{I_2}.$$

We discuss the value of $I_1$ and $I_2$ respectively in the following. For $I_1$, we can obtain that

$$I_1 = \underbrace{\frac{\alpha^{(1)}(D-1)}{D^2} \mathbb{E}\big[\ell'^{(1)} y \langle \mathbf{v}^*, \mathbf{x}_{j^*} \rangle\big] \mathbf{p}_{j^*} \Big( \sum_{j=1}^{D} \mathbf{p}_j^\top \Big)}_{I_{1,1}} + \underbrace{\frac{\alpha^{(1)}(D-1)}{D^2} \sum_{j' \neq j^*} \mathbb{E}\big[\ell'^{(1)} y \langle \mathbf{v}^*, \mathbf{x}_{j'} \rangle\big] \mathbf{p}_{j'} \Big( \sum_{j=1}^{D} \mathbf{p}_j^\top \Big)}_{I_{1,2}}.$$

While for $I_2$, we can also obtain that

$$I_2 = \underbrace{\frac{\alpha^{(1)}}{D^2} \mathbb{E}\big[\ell'^{(1)} y \langle \mathbf{v}^*, \mathbf{x}_{j^*} \rangle\big] \sum_{j'' \neq j^*} \mathbf{p}_{j''} \Big( \sum_{j=1}^{D} \mathbf{p}_j^\top \Big)}_{I_{2,1}} + \underbrace{\frac{\alpha^{(1)}}{D^2} \sum_{j' \neq j^*} \mathbb{E}\big[\ell'^{(1)} y \langle \mathbf{v}^*, \mathbf{x}_{j'} \rangle\big] \mathbf{p}_{j^*} \Big( \sum_{j=1}^{D} \mathbf{p}_j^\top \Big)}_{I_{2,2}}$$

$$+ \frac{\alpha^{(1)}}{D^2} \underbrace{\sum_{j'' \neq j^*} \sum_{j' \neq j'', j^*} \mathbb{E}\big[\ell'^{(1)} y \langle \mathbf{v}^*, \mathbf{x}_{j'} \rangle\big] \mathbf{p}_{j''} \Big( \sum_{j=1}^{D} \mathbf{p}_j^\top \Big)}_{I_{2,3}}.$$

As we discussed earlier, $\mathbb{E}\big[\ell'^{(1)} y \langle \mathbf{v}^*, \mathbf{x}_j \rangle\big]$ takes the same value for all $j \neq j^*$. Therefore, we can obtain that

$$\nabla_{\mathbf{W}_{2,2}} \mathcal{L}(\mathbf{v}^{(1)}, \mathbf{W}^{(1)}) = I_1 - I_2 = \big(I_{1,1} - I_{2,1}\big) + \Big( \big(I_{1,2} - I_{2,3}\big) - I_{2,2} \Big)$$

$$= \frac{\alpha^{(1)}}{D^2} \Big( \mathbb{E}\big[\ell'^{(1)} y \langle \mathbf{v}^*, \mathbf{x}_{j^*} \rangle\big] - \mathbb{E}\big[\ell'^{(1)} y \langle \mathbf{v}^*, \mathbf{x}_{j'} \rangle\big] \Big) \Big( \sum_{j \neq j^*} \big(\mathbf{p}_{j^*} - \mathbf{p}_j\big) \Big) \Big( \sum_{j=1}^{D} \mathbf{p}_j^\top \Big).$$

Furthermore, we can utilize Lemma E.10 and Lemma E.11 to obtain that,

$$\mathbb{E}\big[\ell'^{(1)} y \langle \mathbf{v}^*, \mathbf{x}_{j^*} \rangle\big] - \mathbb{E}\big[\ell'^{(1)} y \langle \mathbf{v}^*, \mathbf{x}_j \rangle\big] \geq -\sigma_x \sqrt{\frac{2}{\pi}} - \frac{4(1 + \eta \sigma_x^2)}{\eta \sigma_x \sqrt{D\pi}} \exp\Big( \frac{\eta^2 \sigma_x^4}{2\pi} \Big),$$

and

$$\mathbb{E}\big[\ell'^{(1)} y \langle \mathbf{v}^*, \mathbf{x}_{j^*} \rangle\big] - \mathbb{E}\big[\ell'^{(1)} y \langle \mathbf{v}^*, \mathbf{x}_j \rangle\big] \leq -\frac{\sigma_x}{2\sqrt{2e\pi}} e^{-\frac{\eta \sigma_x^2}{2\pi}}.$$

Applying all the preceding results to the gradient descent iteration of $\mathbf{W}_{2,2}^{(t)}$, we finally obtain that

$$\mathbf{W}_{2,2}^{(2)} = \mathbf{W}_{2,2}^{(1)} - \eta \nabla_{\mathbf{W}_{2,2}} \mathcal{L}(\mathbf{v}^{(1)}, \mathbf{W}^{(1)})$$

$$= \frac{\eta \alpha^{(1)}}{D^2} \Big( - \mathbb{E}\big[\ell'^{(1)} y \langle \mathbf{v}^*, \mathbf{x}_{j^*} \rangle\big] + \mathbb{E}\big[\ell'^{(1)} y \langle \mathbf{v}^*, \mathbf{x}_{j'} \rangle\big] \Big) \Big( \sum_{j \neq j^*} \big(\mathbf{p}_{j^*} - \mathbf{p}_j\big) \Big) \Big( \sum_{j=1}^{D} \mathbf{p}_j^\top \Big)$$

$$= \beta_2 \Big( \sum_{j \neq j^*} \big(\mathbf{p}_{j^*} - \mathbf{p}_j\big) \Big) \Big( \sum_{j=1}^{D} \mathbf{p}_j^\top \Big),$$

where

$$\frac{1}{D^2} \frac{\eta^2 \sigma_x^2}{4\pi \sqrt{e}} e^{-\frac{\eta \sigma_x^2}{2\pi}} \leq \beta_2 \leq \frac{1}{D^2} \frac{\eta^2 \sigma_x}{\pi} \Big( \sigma_x + \frac{4(1 + \eta \sigma_x^2)}{\eta \sigma_x \sqrt{2D}} e^{\frac{\eta^2 \sigma_x^4}{2\pi}} \Big) \leq \frac{1}{D^2} \frac{2\eta^2 \sigma_x^2}{\pi}.$$

This completes the proof.

### C.4 PROOF OF LEMMA C.5, LEMMA C.2, LEMMA C.3 AND LEMMA C.4

In this subsection, we provide complete proof for Lemma C.5, Lemma C.2, Lemma C.3 and Lemma C.4 We first prove Lemma C.5, given that the result concerning $\mathbf{W}^{(t)}$ in Proposition C.1, Lemma C.3 and Lemma C.4 holds. Then we use Lemma C.5 to prove Lemma C.2, Lemma C.3 and Lemma C.4 by induction. We would like to clarify that this is not circular reasoning, since we are utilizing induction. It's reasonable to assume that all conclusions hold for each iteration and verify all conclusions still hold for the next iteration, as long as we can rigorously demonstrate these conclusions hold at the beginning.

*Proof of Lemma C.5.* By Lemma C.3 and Lemma C.4, there exists constants $c_1, c_2$ solely depending on $\eta, \sigma_x$ such that $|\beta_1| \leq c_1 \sqrt{D}$ and $\beta_2 \geq c_2 \frac{1}{D^2}$. Then further combining with Lemma E.15, Lemma E.16 and Lemma E.18, with probability at least $1 - \sqrt{\frac{26 c_1 \sigma_x^2}{c_2 \pi}} D e^{-\frac{c_2}{26 c_1 \sigma_x^2} \sqrt{D}} - D e^{-\frac{D}{2}}$, we can obtain that

$$\mathbf{z}_{j'}^\top \mathbf{W}^{(t)} \mathbf{z}_j = \mathbf{x}_{j'}^\top \mathbf{W}_{1,1}^{(t)} \mathbf{x}_j + \mathbf{p}_{j'}^\top \mathbf{W}_{2,2}^{(t)} \mathbf{p}_{j'}^\top$$

$$= \beta_1 \langle \mathbf{v}^*, \mathbf{x}_{j'} \rangle \langle \mathbf{v}^*, \mathbf{x}_j \rangle - \frac{(D+1)^2}{4} \beta_2 + \mathbf{x}_{j'}^\top \mathbf{W}_{1,1,\text{error}}^{(t)} \mathbf{x}_j + \mathbf{p}_{j'}^\top \mathbf{W}_{2,2,\text{error}}^{(t)} \mathbf{p}_j^\top$$

$$\leq \big| \beta_1 \langle \mathbf{v}^*, \mathbf{x}_{j'} \rangle \langle \mathbf{v}^*, \mathbf{x}_j \rangle \big| + \big\| \mathbf{W}_{1,1,\text{error}}^{(t)} \big\|_2 \|\mathbf{x}_{j'}\|_2 \|\mathbf{x}_j\|_2 + \big\| \mathbf{W}_{2,2,\text{error}}^{(t)} \big\|_2 \|\mathbf{p}_j\|_2 \|\mathbf{p}_{j'}\|_2$$

$$\leq c_1\sqrt{D}\left(\sqrt{\frac{c_2}{13c_1\sigma_x^2}}\sigma_x D^{\frac{1}{4}}\right)^2 + \frac{1}{e^{C_2\sqrt{D}}}\sigma_x^2(\sqrt{d}+\sqrt{D})^2 + \frac{1}{e^{C_2\sqrt{D}}}\left(\sqrt{\frac{D+1}{2}}\right)^2$$

$$\leq \frac{c_2}{12}D$$

holds for all $j' \neq j^*$ and $j \in [D]$, and

$$\mathbf{z}_{j^*}^\top \mathbf{W}^{(T)}\mathbf{z}_j = \mathbf{x}_{j^*}^\top \mathbf{W}_{1,1}^{(T)}\mathbf{x}_j + \mathbf{p}_{j^*}^\top \mathbf{W}_{2,2}^{(T)}\mathbf{p}_{j'}^\top$$

$$= \beta_1 \langle \mathbf{v}^*, \mathbf{x}_{j^*}\rangle\langle \mathbf{v}^*, \mathbf{x}_j\rangle + \frac{(D-1)(D+1)^2}{4}\beta_2 + \mathbf{x}_{j^*}^\top \mathbf{W}_{1,1,\text{error}}^{(T)}\mathbf{x}_j + \mathbf{p}_{j^*}^\top \mathbf{W}_{2,2,\text{error}}^{(T)}\mathbf{p}_j^\top$$

$$\geq \frac{c_2 D}{4} - \left|\beta_1\langle \mathbf{v}^*, \mathbf{x}_{j'}\rangle\langle\mathbf{v}^*, \mathbf{x}_j\rangle\right| - \left\|\mathbf{W}_{1,1,\text{error}}^{(T)}\right\|_2\|\mathbf{x}_j\|_2\|\mathbf{x}_{j'}\|_2 - \left\|\mathbf{W}_{2,2,\text{error}}^{(T)}\right\|_2\|\mathbf{p}_j\|_2\|\mathbf{p}_{j'}\|_2$$

$$\geq \frac{c_2 D}{4} - c_1\sqrt{D}\left(\sqrt{\frac{c_2}{13c_1\sigma_x^2}}\sigma_x D^{\frac{1}{4}}\right)^2 - \frac{1}{e^{C_2\sqrt{D}}}\sigma_x^2(\sqrt{d}+\sqrt{D})^2 - \frac{1}{e^{C_2\sqrt{D}}}\left(\sqrt{\frac{D+1}{2}}\right)^2$$

$$\geq \frac{c_2}{6}D$$

holds for all $j \in [D]$. Therefore, we can obtain that

$$\mathbf{S}_{j^*,j}^{(T)} \geq \frac{e^{\frac{c_2}{6}D}}{e^{\frac{c_2}{6}D}+(D-1)e^{\frac{c_2}{12}D}} \geq 1 - De^{-\frac{c_2}{12}D};$$

$$\mathbf{S}_{j',j}^{(T)} \leq \frac{e^{\frac{c_2}{12}D}}{e^{\frac{c_2}{6}D}+(D-1)e^{\frac{c_2}{12}D}} \leq e^{-\frac{c_2}{12}D},$$

holds for all $j' \neq j^*$ and $j \in [D]$ with probability at least $1 - \sqrt{\frac{26c_1\sigma_x^2}{c_2\pi}}D^{3/4}e^{-\frac{c_2}{26c_1\sigma_x^2}\sqrt{D}} - De^{-\frac{D}{2}} \geq 1 - e^{-C_8\sqrt{D}}$ for some non-negative constant $C_8$ solely depending on $\eta, \sigma_x$. □

Next, we prove Lemma C.2, Lemma C.3 and Lemma C.4 by induction. When we prove Lemma C.2, we will assume that Lemma C.3 and Lemma C.4 hold at current iteration. The same situation still holds for proof of Lemma C.3 and Lemma C.4. As we discussed earlier, this is not circular reasoning by the essence of induction. Rigorously, all the conclusions from these three lemmas could be composed into a big induction. However, for simplicity and consistency, we present them respectively. Besides, we denote $E_t$ the event that $|\langle \mathbf{v}^*, \mathbf{x}_j\rangle| \leq c_4 D^{1/4}$ and $\|\mathbf{x}_j\|_2 \leq \sigma_x(\sqrt{d}+\sqrt{D})$ for some constant $c_4$ solely depending on $\eta, \sigma_x$ and all $j \in [D]$, and denote $E_t^c$ the complement event. By Lemma C.5, the occurrence of $E_t$ can imply that $\mathbf{S}_{j^*,j}^{(t)} \geq 1 - D\exp(-C_9 D)$ and $\mathbf{S}_{j',j}^{(t)} \leq \exp(-C_9 D)$ for all $j' \neq j^*$ and $j \in [D]$. And the probability of $E_t$ follows that $\mathbb{P}(E_s^{(t)}) \geq 1 - \exp(-C_8\sqrt{D})$.

In the following, we present the proof for Lemma C.2.

*Proof of Lemma C.2.* By Lemma C.11 and Lemma C.12, we have $\mathbf{W}_{1,1}^{(2)} = \beta_1\mathbf{v}^*\mathbf{v}^{*\top}$ with $|\beta_1| \leq c_1\sqrt{D}$, and $\mathbf{W}_{2,2}^{(2)} = \beta_2\left(\sum_{j\neq j^*}(\mathbf{p}_{j^*}-\mathbf{p}_j)\right)\left(\sum_{j=1}^D \mathbf{p}_j^\top\right)$ with $c_2\frac{1}{D^2} \leq \beta_2 \leq c_3\frac{1}{D^2}$, aligning with the formulas of $\mathbf{W}_{1,1}^{(t)}, \mathbf{W}_{2,2}^{(t)}$ in Lemma C.3 and Lemma C.4. We assume the conclusions of Lemma C.3 and Lemma C.4 still hold for any $t < T^*$, then by Lemma C.5, we have $P(E_s^{(t)}) \geq 1 - \exp(-C_8\sqrt{D})$. Based on this result, we define a proxy gradient $\mathcal{G}_v^{(t)}$ which is calculated by assuming $\mathbf{S}_{j^*,j}^{(t)} = 1$, i.e.,

$$\mathcal{G}_v^{(t)} = D\mathbb{E}\left[\ell'\left(D\alpha^{(t)}y\langle\mathbf{v}^*,\mathbf{x}_{j^*}\rangle + Dy\langle\mathbf{v}_{1,\text{error}}^{(t)},\mathbf{x}_{j^*}\rangle\right)y\mathbf{x}_{j^*}\right].$$

Besides, since $\mathbf{v}_{1,\text{error}}^{(t)}$ is perpendicular to $\mathbf{v}^*$ (which is $\mathbf{0}$ at $t=2$), it is inner the linear subspace spanned by the $\{\boldsymbol{\xi}_2,\cdots,\boldsymbol{\xi}_d\}$, and we denote its decomposition as $\mathbf{v}_{1,\text{error}}^{(t)} = \sum_{i=2}^d a_i^{(t)}\boldsymbol{\xi}_i$. By the

orthogonality among $\boldsymbol{\xi}_2, \cdots, \boldsymbol{\xi}_d$, we have $\sum_{i=2}^{d} \left(a_i^{(t)}\right)^2 = \left\|\mathbf{v}_{1,\text{error}}^{(t)}\right\|_2^2$. We also denote $\mathcal{P}_{\boldsymbol{\xi}}$ the projection matrix of the linear subspace spanned by the $\{\boldsymbol{\xi}_2, \cdots, \boldsymbol{\xi}_d\}$. Then, we can decompose $\mathcal{G}_v^{(t)}$ by a similar process in Lemma C.10 as

$$\mathcal{G}_v^{(t)} = D\mathbb{E}\left[\ell'\left(D\alpha^{(t)}y\langle\mathbf{v}^*, \mathbf{x}_{j^*}\rangle + Dy\langle\mathbf{v}_{1,\text{error}}^{(t)}, \mathbf{x}_{j^*}\rangle\right)y\mathbf{x}_{j^*}\right]$$

$$= D\mathbf{A}\mathbf{A}^\top \mathbb{E}\left[\ell'\left(D\alpha^{(t)}y\langle\mathbf{v}^*, \mathbf{x}_{j^*}\rangle + Dy\sum_{i=2}^{d} a_i^{(t)}\langle\boldsymbol{\xi}_i, \mathbf{x}_{j^*}\rangle\right)y\mathbf{x}_{j^*}\right]$$

$$= D\mathbf{A}\mathbb{E}\left[\ell'\left(D\alpha^{(t)}y\langle\mathbf{v}^*, \mathbf{x}_{j^*}\rangle + Dy\sum_{i=2}^{d} a_i^{(t)}\langle\boldsymbol{\xi}_i, \mathbf{x}_{j^*}\rangle\right)y\left[\langle\mathbf{v}^*, \mathbf{x}_{j^*}\rangle, \langle\boldsymbol{\xi}_2, \mathbf{x}_{j^*}\rangle, \cdots, \langle\boldsymbol{\xi}_d, \mathbf{x}_{j^*}\rangle\right]^\top\right]$$

$$= D\mathbb{E}\left[\ell'\left(D\alpha^{(t)}y\langle\mathbf{v}^*, \mathbf{x}_{j^*}\rangle + D\sum_{i=2}^{d} a_i^{(t)}y\langle\boldsymbol{\xi}_i, \mathbf{x}_{j^*}\rangle\right)y\langle\mathbf{v}^*, \mathbf{x}_{j^*}\rangle\right]\mathbf{v}^*$$

$$+ \sum_{i=2}^{d} D\mathbb{E}\left[\ell'\left(D\alpha^{(t)}y\langle\mathbf{v}^*, \mathbf{x}_{j^*}\rangle + D\sum_{i=2}^{d} a_i^{(t)}y\langle\boldsymbol{\xi}_i, \mathbf{x}_{j^*}\rangle\right)y\langle\boldsymbol{\xi}_i, \mathbf{x}_{j^*}\rangle\right]\boldsymbol{\xi}_i.$$

And we can upper bound the difference term as

$$\left\|\nabla_{\mathbf{v}_1}\mathcal{L}(\mathbf{v}^{(t)}, \mathbf{W}^{(t)}) - \mathcal{G}_v^{(t)}\right\|_2 = \left\|\mathbb{E}\left[D\ell'\left(D\alpha^{(t)}y\langle\mathbf{v}^*, \mathbf{x}_{j^*}\rangle + Dy\langle\mathbf{v}_{1,\text{error}}^{(t)}, \mathbf{x}_{j^*}\rangle\right)y\mathbf{x}_{j^*} - \sum_{j'=1}^{D}\sum_{j=1}^{D}\ell'^{(t)}y\mathbf{x}_{j'}\mathbf{S}_{j',j}^{(t)}\right]\right\|_2$$

$$\leq \mathbb{E}\left[\left\|\left(D\ell'\left(D\alpha^{(t)}y\langle\mathbf{v}^*, \mathbf{x}_{j^*}\rangle + Dy\langle\mathbf{v}_{1,\text{error}}^{(t)}, \mathbf{x}_{j^*}\rangle\right)y\mathbf{x}_{j^*} - \sum_{j'=1}^{D}\sum_{j=1}^{D}\ell'^{(t)}y\mathbf{x}_{j'}\mathbf{S}_{j',j}^{(t)}\right)\mathbf{1}_{\{E_t\}}\right\|_2\right]$$

$$+ \mathbb{E}\left[\left\|\left(D\ell'\left(D\alpha^{(t)}y\langle\mathbf{v}^*, \mathbf{x}_{j^*}\rangle + Dy\langle\mathbf{v}_{1,\text{error}}^{(t)}, \mathbf{x}_{j^*}\rangle\right)y\mathbf{x}_{j^*} - \sum_{j'=1}^{D}\sum_{j=1}^{D}\ell'^{(t)}y\mathbf{x}_{j'}\mathbf{S}_{j',j}^{(t)}\right)\mathbf{1}_{\{E_t^c\}}\right\|_2\right]$$

$$\leq \underbrace{D\mathbb{E}\left[\left|\ell'\left(D\alpha^{(t)}y\langle\mathbf{v}^*, \mathbf{x}_{j^*}\rangle + Dy\langle\mathbf{v}_{1,\text{error}}^{(t)}, \mathbf{x}_{j^*}\rangle\right) - \ell'^{(t)}\right|\|\mathbf{x}_{j^*}\|_2\mathbf{1}_{\{E_t\}}\right]}_{I_1} + \underbrace{\mathbb{E}\left[\left(D - \sum_{j=1}^{D}\mathbf{S}_{j^*,j}^{(t)}\right)\|\mathbf{x}_{j^*}\|_2\mathbf{1}_{\{E_t\}}\right]}_{I_2}$$

$$+ \underbrace{\sum_{j'\neq j^*}\sum_{j=1}^{D}\mathbb{E}\left[\|\mathbf{x}_{j'}\|_2\mathbf{S}_{j',j}^{(t)}\mathbf{1}_{\{E_t\}}\right]}_{I_3} + \underbrace{D\mathbb{E}\left[\|\mathbf{x}_{j^*}\|_2\mathbf{1}_{\{E_t^c\}}\right]}_{I_4} + \underbrace{D\sum_{j'=1}^{D}\mathbb{E}\left[\|\mathbf{x}_{j'}\|_2\mathbf{1}_{\{E_t^c\}}\right]}_{I_5},$$

where the inequalities hold by triangle inequality and the fact that $|\ell'| \leq 1$ and $|\mathbf{S}_{j',j}| \leq 1$. Next, we demonstrate our analysis on two cases: $t = 2$ and $t \geq 3$.

When $t = 2$, $\mathbf{v}_1^{(2)} = \alpha^{(2)}\mathbf{v}^*$ implies that $\mathbf{v}_{1,\text{error}}^{(2)} = \mathbf{0}$. Therefore, we have

$$\mathcal{G}_v^{(2)} = D\mathbb{E}\left[\ell'\left(D\alpha^{(2)}y\langle\mathbf{v}^*, \mathbf{x}_{j^*}\rangle\right)y\langle\mathbf{v}^*, \mathbf{x}_{j^*}\rangle\right]\mathbf{v}^*.$$

And we provide the upper bounds for each term of $\left\|\nabla_{\mathbf{v}_1}\mathcal{L}(\mathbf{v}^{(2)}, \mathbf{W}^{(2)}) - \mathcal{G}_v^{(2)}\right\|_2$ respectively. Specially, for $I_1$, we can derive that

$$I_1 \leq D\mathbb{E}\left[\left|\ell'\left(D\alpha^{(t)}y\langle\mathbf{v}^*, \mathbf{x}_{j^*}\rangle + Dy\langle\mathbf{v}_{1,\text{error}}^{(t)}, \mathbf{x}_{j^*}\rangle\right) - \ell'^{(2)}\right|\|\mathbf{x}_{j^*}\|_2\mathbf{1}_{\{E_t\}}\right]$$

$$\leq \frac{D}{4}\mathbb{E}\left[\left|D\alpha^{(2)}y\langle\mathbf{v}^*, \mathbf{x}_{j^*}\rangle - \alpha^{(2)}\sum_{j'=1}^{D}\sum_{j=1}^{D}y\langle\mathbf{v}^*, \mathbf{x}_{j'}\rangle\mathbf{S}_{j',j}^{(t)}\right|\|\mathbf{x}_{j^*}\|_2\mathbf{1}_{\{E_t\}}\right]$$

$$\leq \frac{D}{4}|\alpha^{(2)}|\mathbb{E}\left[\left(D - \sum_{j=1}^{D}\mathbf{S}_{j^*,j}^{(t)}\right)|\langle\mathbf{v}^*, \mathbf{x}_{j^*}\rangle|\|\mathbf{x}_{j^*}\|_2\mathbf{1}_{\{E_t\}}\right] + \frac{D}{4}|\alpha^{(2)}|\sum_{j'\neq j^*}\sum_{j=1}^{D}\mathbb{E}\left[|\langle\mathbf{v}^*, \mathbf{x}_{j'}\rangle|\|\mathbf{x}_{j^*}\|_2\mathbf{S}_{j',j}^{(t)}\mathbf{1}_{\{E_t\}}\right]$$

$$\leq \frac{c_3 \sigma_x D^{\frac{13}{4}} |\alpha^{(2)}| (\sqrt{d} + \sqrt{D})}{2 e^{C_9 D}} \leq \frac{1}{5 \eta e^{C_4 \sqrt{D}}}.$$

The first inequality is because $\ell'$ is Lipschitz continuous with $\frac{1}{4}$. The penultimate inequality holds since $|\langle \mathbf{v}^*, \mathbf{x}_j \rangle| \leq c_3 D^{1/4}$, $\|\mathbf{x}_j\|_2 \leq \sigma_x (\sqrt{d} + \sqrt{D})$, $\mathbf{S}_{j^*,j}^{(t)} \geq 1 - D \exp(-C_9 D)$ and $\mathbf{S}_{j',j}^{(t)} \leq \exp(-C_9 D)$ when $\mathbf{1}_{\{E_t\}} = 1$. And the last inequality holds since $\alpha^{(2)} = -\Theta(\sqrt{D})$. Similarly, we also have

$$I_2 \leq \frac{\sigma_x D (\sqrt{d} + \sqrt{D})}{e^{C_9 D}} \leq \frac{1}{5 \eta e^{C_4 \sqrt{D}}}; \quad \text{and} \quad I_3 \leq \frac{\sigma_x D^2 (\sqrt{d} + \sqrt{D})}{e^{cD}} \leq \frac{1}{5 \eta e^{C_4 \sqrt{D}}}.$$

For $I_4$ and $I_5$, by applying Cauchy-Schwarz inequality, we have

$$I_4 \leq D \sqrt{\mathbb{E}\big[\|\mathbf{x}_{j^*}\|_2^2\big]} \sqrt{\mathbb{E}\big[\mathbf{1}_{\{E_t^c\}}^2\big]} \leq \frac{D \sqrt{d}}{e^{\frac{C_8}{2} \sqrt{D}}} \leq \frac{1}{5 \eta e^{C_4 \sqrt{D}}};$$

$$I_5 \leq D \sum_{j'=1}^{D} \sqrt{\mathbb{E}\big[\|\mathbf{x}_{j'}\|_2^2\big]} \sqrt{\mathbb{E}\big[\mathbf{1}_{\{E_t^c\}}^2\big]} \leq \frac{D^2 \sqrt{d}}{e^{\frac{C_8}{2} \sqrt{D}}} \leq \frac{1}{5 \eta e^{C_4 \sqrt{D}}}.$$

Combining all preceding results, we have $\big\|\nabla_{\mathbf{v}_1} \mathcal{L}(\mathbf{v}^{(2)}, \mathbf{W}^{(2)}) - \mathcal{G}_v^{(2)}\big\|_2 \leq \frac{1}{e^{C_4 \sqrt{D}}}$ for some constant $C_4$ only depending on $\sigma_x$ and $\eta$. Furthermore, we can derive that

$$\mathbf{v}_1^{(3)} = \mathbf{v}_1^{(2)} - \eta \nabla_{\mathbf{v}_1} \mathcal{L}(\mathbf{v}^{(2)}, \mathbf{W}^{(2)}) = \alpha^{(2)} \mathbf{v}^* - \eta \mathcal{G}_v^{(2)} - \eta \big(\nabla_{\mathbf{v}_1} \mathcal{L}(\mathbf{v}^{(2)}, \mathbf{W}^{(2)}) - \mathcal{G}_v^{(2)}\big)$$

$$= \Big(\alpha^{(2)} + D \eta \mathbb{E}\big[ -\ell'\big(D\alpha^{(2)} y \langle \mathbf{v}^*, \mathbf{x}_{j^*} \rangle\big) y \langle \mathbf{v}^*, \mathbf{x}_{j^*} \rangle\big] + \big\langle \mathbf{v}^*, \eta \mathcal{G}_v^{(2)} - \eta \nabla_{\mathbf{v}_1} \mathcal{L}(\mathbf{v}^{(2)}, \mathbf{W}^{(2)})\big\rangle\Big) \mathbf{v}^*$$

$$+ \eta \mathcal{P}_{\boldsymbol{\xi}}\big(\mathcal{G}_v^{(2)} - \nabla_{\mathbf{v}_1} \mathcal{L}(\mathbf{v}^{(2)}, \mathbf{W}^{(2)})\big)$$

$$= \alpha^{(3)} \mathbf{v}^* + \mathbf{v}_{1,\text{error}}^{(3)}.$$

For $\alpha^{(3)}$, we can utilize Lemma E.10 to derive that

$$\alpha^{(3)} \geq \frac{\eta \sigma_x}{4} \sqrt{\frac{2}{\pi}} D + \alpha^{(2)} - \eta \big\|\nabla_{\mathbf{v}_1} \mathcal{L}(\mathbf{v}^{(2)}, \mathbf{W}^{(2)}) - \mathcal{G}_v^{(2)}\big\|_2 \geq \frac{\eta \sigma_x}{5} \sqrt{\frac{2}{\pi}} D;$$

$$\alpha^{(3)} \leq \eta \sigma_x \sqrt{\frac{2}{\pi}} D + \alpha^{(2)} + \eta \big\|\nabla_{\mathbf{v}_1} \mathcal{L}(\mathbf{v}^{(2)}, \mathbf{W}^{(2)}) - \mathcal{G}_v^{(2)}\big\|_2 \leq \eta \sigma_x \sqrt{\frac{2}{\pi}} D.$$

And we also have $\big\|\mathbf{v}_{1,\text{error}}^{(3)}\big\|_2 \leq \eta \big\|\nabla_{\mathbf{v}_1} \mathcal{L}(\mathbf{v}^{(2)}, \mathbf{W}^{(2)}) - \mathcal{G}_v^{(2)}\big\|_2 \leq e^{-C_4 \sqrt{D}}$, which completes the proof at $t = 3$.

Since we have derived that (C.5) and (C.4) hold at $t = 3$. We will use induction to prove the case when $3 < t \leq T^*$. Instead of directly proving (C.4), we prove the following inequality

$$\big\|\mathbf{v}_{1,\text{error}}^{(t)}\big\|_2 \leq \frac{\eta t}{e^{c \sqrt{D}}}, \tag{C.12}$$

where $c$ is some positive constant solely depending on $\sigma_x$ and $\eta$. Then we assume (C.5) and (C.12) hold at any $3 < t \leq T^* - 1$. Similarly, we can bound $I_1$ in $\big\|\nabla_{\mathbf{v}_1} \mathcal{L}(\mathbf{v}^{(t)}, \mathbf{W}^{(t)}) - \mathcal{G}_v^{(t)}\big\|_2$ as

$$I_1 \leq D \mathbb{E}\Big[\Big|\ell'\big(D\alpha^{(t)} y \langle \mathbf{v}^*, \mathbf{x}_{j^*} \rangle + D y \langle \mathbf{v}_{1,\text{error}}^{(t)}, \mathbf{x}_{j^*} \rangle\big) - \ell'^{(t)}\Big| \|\mathbf{x}_{j^*}\|_2 \mathbf{1}_{\{E_t\}}\Big]$$

$$\leq \frac{D}{4} \mathbb{E}\Big[\Big| D\alpha^{(t)} y \langle \mathbf{v}^*, \mathbf{x}_{j^*} \rangle + D y \langle \mathbf{v}_{1,\text{error}}^{(t)}, \mathbf{x}_{j^*} \rangle - \sum_{j'=1}^{D} \sum_{j=1}^{D} y \langle \alpha^{(t)} \mathbf{v}^* + \mathbf{v}_{1,\text{error}}^{(t)}, \mathbf{x}_{j'} \rangle \mathbf{S}_{j',j}^{(t)}\Big| \|\mathbf{x}_{j^*}\|_2 \mathbf{1}_{\{E_t\}}\Big]$$

$$\leq \frac{D\alpha^{(t)}}{4} \mathbb{E}\Big[\Big(D - \sum_{j=1}^{D} \mathbf{S}_{j^*,j}^{(t)}\Big) |\langle \mathbf{v}^*, \mathbf{x}_{j^*} \rangle| \|\mathbf{x}_{j^*}\|_2 \mathbf{1}_{\{E_t\}}\Big] + \frac{D\alpha^{(t)}}{4} \sum_{j' \neq j^*} \sum_{j=1}^{D} \mathbb{E}\Big[|\langle \mathbf{v}^*, \mathbf{x}_{j'} \rangle| \|\mathbf{x}_{j^*}\|_2 \mathbf{S}_{j',j}^{(t)} \mathbf{1}_{\{E_t\}}\Big]$$

$$+ \frac{D}{4} \mathbb{E}\Big[\Big(D - \sum_{j=1}^{D} \mathbf{S}_{j^*,j}^{(t)}\Big) \|\mathbf{v}_{1,\text{error}}^{(t)}\|_2 \|\mathbf{x}_{j^*}\|_2^2 \mathbf{1}_{\{E_t\}}\Big] + \frac{D}{4} \sum_{j' \neq j^*} \sum_{j=1}^{D} \mathbb{E}\Big[\|\mathbf{v}_{1,\text{error}}^{(t)}\|_2 \|\mathbf{x}_{j'}\|_2 \|\mathbf{x}_{j^*}\|_2 \mathbf{1}_{\{E_t\}}\Big]$$

$$\leq \frac{c_3 \sigma_x D^{\frac{13}{4}} \alpha^{(t)} (\sqrt{d} + \sqrt{D}) + 4\sigma_x^2 D^3 \|\mathbf{v}_{1,\text{error}}^{(t)}\|_2 \max\{d, D\}}{2e^{C_9 D}} \leq \frac{1}{5e^{c\sqrt{D}}}$$

The first inequality is because $\ell'$ is Lipschitz continuous with $\frac{1}{4}$. The penultimate inequality holds since $|\langle \mathbf{v}^*, \mathbf{x}_j \rangle| \leq c_3 D^{1/4}$, $\|\mathbf{x}_j\|_2 \leq \sigma_x(\sqrt{d} + \sqrt{D})$, $\mathbf{S}_{j^*,j}^{(t)} \geq 1 - D\exp(-C_9 D)$ and $\mathbf{S}_{j',j}^{(t)} \leq \exp(-C_9 D)$ when $\mathbf{1}_{\{E_t\}} = 1$. And the last inequality holds since $\alpha^{(t)} \leq O((T^*)^{1/3} + D) \ll \exp(C_9 D)$ by (C.5) and our definition of $T^*$ in Theorem 2.2. For other terms in $\left\|\nabla_{\mathbf{v}_1}\mathcal{L}(\mathbf{v}^{(t)}, \mathbf{W}^{(t)}) - \mathcal{G}_v^{(t)}\right\|_2$, we can obtain the same upper bound with $t = 2$. Therefore, by combining these results, we obtain that $\left\|\nabla_{\mathbf{v}_1}\mathcal{L}(\mathbf{v}^{(t)}, \mathbf{W}^{(t)}) - \mathcal{G}_v^{(t)}\right\|_2 \leq \frac{1}{e^{c\sqrt{D}}}$ for some constant $c$ only depending on $\sigma_x$ and $\eta$. Now, we are ready to derive the gradient descent update for $\mathbf{v}^{(t+1)}$ as

$$\mathbf{v}_1^{(t+1)} = \mathbf{v}_1^{(t)} - \eta\nabla_{\mathbf{v}_1}\mathcal{L}(\mathbf{v}^{(t)}, \mathbf{W}^{(t)}) = \alpha^{(t)}\mathbf{v}^* + \mathbf{v}_{1,\text{error}}^{(t)} - \eta\mathcal{G}_v^{(t)} - \eta\left(\nabla_{\mathbf{v}_1}\mathcal{L}(\mathbf{v}^{(t)}, \mathbf{W}^{(t)}) - \mathcal{G}_v^{(t)}\right)$$

$$= \left(\alpha^{(t)} + D\eta\mathbb{E}\left[-\ell'\left(D\alpha^{(t)}y\langle\mathbf{v}^*, \mathbf{x}_{j^*}\rangle + Dy\langle\mathbf{v}_{1,\text{error}}^{(t)}, \mathbf{x}_{j^*}\rangle\right)y\langle\mathbf{v}^*, \mathbf{x}_{j^*}\rangle\right] + \left\langle\mathbf{v}^*, \eta\mathcal{G}_v^{(t)} - \eta\nabla_{\mathbf{v}_1}\mathcal{L}\right\rangle\right)\mathbf{v}^*$$

$$+ \sum_{i=2}^{d}\left(a_i^{(t)} + \eta D\mathbb{E}\left[-\ell'\left(D\alpha^{(t)}y\langle\mathbf{v}^*, \mathbf{x}_{j^*}\rangle + D\sum_{i=2}^{d}a_i^{(t)}y\langle\boldsymbol{\xi}_i, \mathbf{x}_{j^*}\rangle\right)y\langle\boldsymbol{\xi}_i, \mathbf{x}_{j^*}\rangle\right]\right)\boldsymbol{\xi}_i + \eta\mathcal{P}_{\boldsymbol{\xi}}\left(\mathcal{G}_v^{(t)} - \nabla_{\mathbf{v}_1}\mathcal{L}\right)$$

$$= \alpha^{(t+1)}\mathbf{v}^* + \mathbf{v}_{1,\text{error}}^{(t+1)}.$$

For $\alpha^{(t+1)}$, we can utilize Lemma E.10 to derive the following iterative formulas

$$\alpha^{(t+1)} \geq \alpha^{(t)} + \eta D\sqrt{\frac{2}{\pi}}\left(\frac{1}{\sigma_x D^2 (\alpha^{(t)})^2} - \frac{1}{\sigma_x^3 D^4 (\alpha^{(t)})^4}\right) - \eta\left\|\nabla_{\mathbf{v}_1}\mathcal{L}(\mathbf{v}^{(t)}, \mathbf{W}^{(t)}) - \mathcal{G}_v^{(t)}\right\|_2$$

$$\geq \alpha^{(t)} + \sqrt{\frac{2}{\pi}}\frac{\eta}{2\sigma_x D}\frac{1}{(\alpha^{(t)})^2},$$

and

$$\alpha^{(t+1)} \leq \alpha^{(t)} + \eta D\sqrt{\frac{2}{\pi}}\frac{1}{\sigma_x D^2 (\alpha^{(t)})^2}e^{\frac{\sigma_x^2 D^2\|\mathbf{v}_{1,\text{error}}^{(t)}\|_2^2}{2}} + \eta\left\|\nabla_{\mathbf{v}_1}\mathcal{L}(\mathbf{v}^{(t)}, \mathbf{W}^{(t)}) - \mathcal{G}_v^{(t)}\right\|_2$$

$$\leq \alpha^{(t)} + \sqrt{\frac{2}{\pi}}\frac{2\eta}{\sigma_x D}\frac{1}{(\alpha^{(t)})^2}$$

hold for all $t \geq 3$. Then combined with the fact about initialization that $\frac{\eta\sigma_x}{5}\sqrt{\frac{2}{\pi}}D \leq \alpha^{(3)} \leq \eta\sigma_x\sqrt{\frac{2}{\pi}}D$, Lemma E.19 and comparison theorem, we finally get that

$$\left(\sqrt{\frac{2}{\pi}}\frac{3\eta}{2\sigma_x D}(t-2) + \frac{2\eta^3\sigma_x^3}{125\pi}\sqrt{\frac{2}{\pi}}D^3\right)^{\frac{1}{3}} \leq \alpha^{(t+1)} \leq \sqrt{\frac{\pi}{2}}\frac{2}{\eta\sigma_x^3 D^3} + \left(\sqrt{\frac{2}{\pi}}\frac{6\eta}{\sigma_x D}(t-2) + \frac{2\eta^3\sigma_x^3}{\pi}\sqrt{\frac{2}{\pi}}D^3\right)^{\frac{1}{3}},$$

which completes the proof for (C.5). Next, we demonstrate the upper bound for $\|\mathbf{v}_{1,\text{error}}^{(t+1)}\|_2$. In order to provide this result, we first analyze the coefficient of each $\boldsymbol{\xi}_i$, and W.L.O.G, we assume $a_i^{(t)} \geq 0$. Then by Lemma E.11, we have

$$\mathbb{E}\left[-\ell'\left(D\alpha^{(t)}y\langle\mathbf{v}^*, \mathbf{x}_{j^*}\rangle + Dy\langle\mathbf{v}_{1,\text{error}}^{(t)}, \mathbf{x}_{j^*}\rangle\right)y\langle\mathbf{v}^*, \mathbf{x}_{j^*}\rangle\right] \leq 0,$$

and

$$\mathbb{E}\left[-\ell'\left(D\alpha^{(t)}y\langle\mathbf{v}^*, \mathbf{x}_{j^*}\rangle + Dy\langle\mathbf{v}_{1,\text{error}}^{(t)}, \mathbf{x}_{j^*}\rangle\right)y\langle\mathbf{v}^*, \mathbf{x}_{j^*}\rangle\right] \geq -\sqrt{\frac{2}{\pi}}\frac{\sigma_x^2 a_i^{(t)}}{\alpha^{(t)}}e^{\frac{\sigma_x^2 D^2\|\mathbf{v}_{1,\text{error}}^{(t)}\|_2^2}{2}} \geq -\frac{6\sigma_x a_i^{(t)}}{\eta D},$$

where the last inequality holds by $\alpha^{(t)} \geq \frac{\eta\sigma_x}{5}\sqrt{\frac{2}{\pi}}D$. Therefore the coefficient of $\boldsymbol{\xi}_i$ follows that

$$\left|a_i^{(t)} + \eta D\mathbb{E}\left[-\ell'\left(D\alpha^{(t)}y\langle\mathbf{v}^*, \mathbf{x}_{j^*}\rangle + D\sum_{i=2}^{d}a_i^{(t)}y\langle\boldsymbol{\xi}_i, \mathbf{x}_{j^*}\rangle\right)y\langle\boldsymbol{\xi}_i, \mathbf{x}_{j^*}\rangle\right]\right| \leq |a_i^{(t)}|,$$

by $\sigma_x \leq \frac{1}{3}$ from conditions of Theorem 2.2 Based on these results, we can finally provide the upper-bound for $\left\|\mathbf{v}_{1,\text{error}}^{(t+1)}\right\|_2$ as

$$
\begin{aligned}
\left\|\mathbf{v}_{1,\text{error}}^{(t+1)}\right\|_2 &\leq \left\|\sum_{i=2}^{d}\left(a_i^{(t)} + \eta D\mathbb{E}\Big[-\ell'\big(D\alpha^{(t)}y\langle\mathbf{v}^*,\mathbf{x}_{j^*}\rangle + D\sum_{i=2}^{d}a_i^{(t)}y\langle\boldsymbol{\xi}_i,\mathbf{x}_{j^*}\rangle\big)y\langle\boldsymbol{\xi}_i,\mathbf{x}_{j^*}\rangle\Big]\right)\boldsymbol{\xi}_i\right\|_2 \\
&\quad + \eta\left\|\mathcal{G}_v^{(t)} - \nabla_{\mathbf{v}_1}\mathcal{L}(\mathbf{v}^{(t)},\mathbf{W}^{(t)})\right\|_2 \\
&\leq \sqrt{\sum_{i=2}^{d}\left(a_i^{(t)} + \eta D\mathbb{E}\Big[-\ell'\big(D\alpha^{(t)}y\langle\mathbf{v}^*,\mathbf{x}_{j^*}\rangle + D\sum_{i=2}^{d}a_i^{(t)}y\langle\boldsymbol{\xi}_i,\mathbf{x}_{j^*}\rangle\big)y\langle\boldsymbol{\xi}_i,\mathbf{x}_{j^*}\rangle\Big]\right)^2 + \frac{\eta}{e^{c\sqrt{D}}}} \\
&\leq \sqrt{\sum_{i=2}^{d}\big(a_i^{(t)}\big)^2 + \frac{\eta}{e^{c\sqrt{D}}}} = \left\|\mathbf{v}_{1,\text{error}}^{(t)}\right\|_2 + \frac{\eta}{e^{c\sqrt{D}}} \leq \frac{(t+1)\eta}{e^{c\sqrt{D}}},
\end{aligned}
$$

which completes the proof for (C.12). Further by our definition of $T^*$ in Theorem 2.2, we can obtain

$$
\left\|\mathbf{v}_{1,\text{error}}^{(t)}\right\|_2 \leq \frac{T^*\eta}{e^{c\sqrt{D}}} \leq e^{-C_4\sqrt{D}}.
$$

This completes the proof for (C.4).

$\square$

Next, we present the proof for Lemma C.3.

*Proof of Lemma C.3.* By calculations in Lemma C.11, we have obtained that $\mathbf{W}_{1,1}^{(2)}$ follows (C.6) with $\mathbf{W}_{1,1,\text{error}}^{(2)} = 0$. Instead of directly proving (C.7), we prove the following inequality,

$$
\|\mathbf{W}_{1,1,\text{error}}^{(t)}\|_2 \leq \frac{\eta t}{e^{c\sqrt{D}}}. \tag{C.13}
$$

for some constant $c$ solely depending on $\eta$ and $\sigma_x$. Therefore by induction, we assume it holds for $t$, and prove it still holds for $t+1$. Actually, it suffices to show that $\left\|\nabla_{\mathbf{W}_{1,1}}\mathcal{L}(\mathbf{v}^{(t)},\mathbf{W}^{(t)})\right\|_2 \leq e^{-c\sqrt{D}}$. By (C.2), we have

$$
\begin{aligned}
&\nabla_{\mathbf{W}_{1,1}}\mathcal{L}(\mathbf{v}^{(t)},\mathbf{W}^{(t)}) \\
={}&\mathbb{E}\left[\ell'^{(t)}\sum_{j=1}^{D}\sum_{j'=1}^{D}\sum_{j''\neq j'}\mathbf{S}_{j',j}^{(t)}\mathbf{S}_{j'',j}^{(t)}y\langle\mathbf{v}_1^{(t)},\mathbf{x}_{j'}\rangle(\mathbf{x}_{j'}-\mathbf{x}_{j''})\mathbf{x}_j^{\top}\right] \\
={}&\underbrace{\mathbb{E}\left[\ell'^{(t)}\sum_{j=1}^{D}\sum_{j'=1}^{D}\mathbf{S}_{j',j}^{(t)}\big(1-\mathbf{S}_{j',j}^{(t)}\big)y\langle\mathbf{v}_1^{(t)},\mathbf{x}_{j'}\rangle\mathbf{x}_{j'}\mathbf{x}_j^{\top}\right]}_{I_1} - \underbrace{\mathbb{E}\left[\ell'^{(t)}\sum_{j=1}^{D}\sum_{j'=1}^{D}\sum_{j''\neq j'}\mathbf{S}_{j',j}^{(t)}\mathbf{S}_{j'',j}^{(t)}y\langle\mathbf{v}_1^{(t)},\mathbf{x}_{j'}\rangle\mathbf{x}_{j''}\mathbf{x}_j^{\top}\right]}_{I_2} \\
={}&\underbrace{\mathbb{E}\left[\ell'^{(t)}\sum_{j=1}^{D}\sum_{j'=1}^{D}\mathbf{S}_{j',j}^{(t)}\big(1-\mathbf{S}_{j',j}^{(t)}\big)y\langle\mathbf{v}_1^{(t)},\mathbf{x}_{j'}\rangle\mathbf{x}_{j'}\mathbf{x}_j^{\top}\mathbf{1}_{\{E_t\}}\right]}_{I_{1,1}} + \underbrace{\mathbb{E}\left[\ell'^{(t)}\sum_{j=1}^{D}\sum_{j'=1}^{D}\mathbf{S}_{j',j}^{(t)}\big(1-\mathbf{S}_{j',j}^{(t)}\big)y\langle\mathbf{v}_1^{(t)},\mathbf{x}_{j'}\rangle\mathbf{x}_{j'}\mathbf{x}_j^{\top}\mathbf{1}_{\{E_t^c\}}\right]}_{I_{1,2}} \\
&-\underbrace{\mathbb{E}\left[\ell'^{(t)}\sum_{j=1}^{D}\sum_{j'=1}^{D}\sum_{j''\neq j'}\mathbf{S}_{j',j}^{(t)}\mathbf{S}_{j'',j}^{(t)}y\langle\mathbf{v}_1^{(t)},\mathbf{x}_{j'}\rangle\mathbf{x}_{j''}\mathbf{x}_j^{\top}\mathbf{1}_{\{E_t\}}\right]}_{I_{2,1}} - \underbrace{\mathbb{E}\left[\ell'^{(t)}\sum_{j=1}^{D}\sum_{j'=1}^{D}\sum_{j''\neq j'}\mathbf{S}_{j',j}^{(t)}\mathbf{S}_{j'',j}^{(t)}y\langle\mathbf{v}_1^{(t)},\mathbf{x}_{j'}\rangle\mathbf{x}_{j''}\mathbf{x}_j^{\top}\mathbf{1}_{\{E_t^c\}}\right]}_{I_{2,2}}
\end{aligned}
$$

For $I_{1,1}$, we have

$$
\|I_{1,1}\|_2 \leq \sum_{j=1}^{D}\sum_{j'=1}^{D}\mathbb{E}\left[\mathbf{S}_{j',j}^{(t)}\big(1-\mathbf{S}_{j',j}^{(t)}\big)\big(\alpha^{(t)}|\langle\mathbf{v}^*,\mathbf{x}_{j'}\rangle| + \|\mathbf{v}_{1,\text{error}}^{(t)}\|_2\|\mathbf{x}_{j'}\|_2\big)\|\mathbf{x}_{j'}\|_2\|\mathbf{x}_j\|_2\mathbf{1}_{\{E_t\}}\right]
$$

$$\leq \frac{8D^2 \max\{d,D\}\big(c_3\alpha^{(t)}D^{1/4} + \big\|\mathbf{v}_{1,\text{error}}^{(t)}\big\|_2(\sqrt{d}+\sqrt{D})\big)}{e^{c_9 D}} \leq \frac{1}{4e^{c\sqrt{D}}}.$$

The first inequality is by triangle inequality and $|y|, |\ell'| \leq 1$. The second inequality holds because $|\langle \mathbf{v}^*, \mathbf{x}_j\rangle| \leq c_3 D^{1/4}$, $\|\mathbf{x}_j\|_2 \leq \sigma_x(\sqrt{d}+\sqrt{D})$, $\mathbf{S}_{j^*,j}^{(t)} \geq 1 - D\exp(-cD)$ and $\mathbf{S}_{j',j}^{(t)} \leq \exp(-cD)$ when $\mathbf{1}_{\{E_t\}} = 1$. The second inequality holds since $\alpha^{(t)} \leq O((T^*)^{1/3} + D) \ll \exp(C_9 D)$ and $\big\|\mathbf{v}_{1,\text{error}}^{(t)}\big\|_2 \leq e^{-C_4\sqrt{D}}$ by Lemma C.2 and definition of $T^*$ in Theorem 2.2. Similarly, for $I_{2,1}$, we can obtain that

$$\big\|I_{2,1}\big\|_2 \leq \sum_{j=1}^{D}\sum_{j'=1}^{D}\sum_{j''\neq j'}^{D} \mathbb{E}\bigg[\mathbf{S}_{j',j}^{(t)}\mathbf{S}_{j'',j}^{(t)}\Big(\alpha^{(t)}|\langle\mathbf{v}^*,\mathbf{x}_{j'}\rangle| + \big\|\mathbf{v}_{1,\text{error}}^{(t)}\big\|_2\|\mathbf{x}_{j'}\|_2\Big)\|\mathbf{x}_{j''}\|_2\|\mathbf{x}_j\|_2\mathbf{1}_{\{E_t\}}\bigg]$$

$$\leq \frac{9D^2 \max\{d,D\}\big(c_3\alpha^{(t)}D^{1/4} + \big\|\mathbf{v}_{1,\text{error}}^{(t)}\big\|_2(\sqrt{d}+\sqrt{D})\big)}{e^{cD}} \leq \frac{1}{4e^{c\sqrt{D}}}.$$

In the following, we also denote $\mathbf{v}^*$ by $\boldsymbol{\xi}_1$ in summation calculation for simplicity of expression. Then by applying Cauchy–Schwarz inequality, we can obtain an upper bound as

$$\big\|I_{1,2}\big\|_2 = \left\|\mathbf{A}\mathbb{E}\bigg[\ell'^{(t)}\sum_{j=1}^{D}\sum_{j'=1}^{D}\mathbf{S}_{j',j}^{(t)}\big(1-\mathbf{S}_{j',j}^{(t)}\big)y\langle\mathbf{v}_1^{(t)},\mathbf{x}_{j'}\rangle \begin{bmatrix}\langle\boldsymbol{\xi}_1,\mathbf{x}_{j'}\rangle\\\langle\boldsymbol{\xi}_2,\mathbf{x}_{j'}\rangle\\\vdots\\\langle\boldsymbol{\xi}_d,\mathbf{x}_{j'}\rangle\end{bmatrix}\big[\langle\boldsymbol{\xi}_1,\mathbf{x}_j\rangle,\langle\boldsymbol{\xi}_2,\mathbf{x}_j\rangle,\cdots,\langle\boldsymbol{\xi}_d,\mathbf{x}_j\rangle\big]\mathbf{1}_{\{E_t^c\}}\bigg]\mathbf{A}^\top\right\|_2$$

$$= \left\|\sum_{i=1}^{d}\sum_{i'=1}^{d}\mathbb{E}\bigg[\ell'^{(t)}\sum_{j=1}^{D}\sum_{j'=1}^{D}\mathbf{S}_{j',j}^{(t)}\big(1-\mathbf{S}_{j',j}^{(t)}\big)y\langle\mathbf{v}_1^{(t)},\mathbf{x}_{j'}\rangle\langle\boldsymbol{\xi}_i,\mathbf{x}_{j'}\rangle\langle\boldsymbol{\xi}_{i'},\mathbf{x}_j\rangle\mathbf{1}_{\{E_t^c\}}\bigg]\boldsymbol{\xi}_i\boldsymbol{\xi}_{i'}\right\|_2$$

$$= \sqrt{\sum_{i=1}^{d}\sum_{i'=1}^{d}\mathbb{E}\bigg[\ell'^{(t)}\sum_{j=1}^{D}\sum_{j'=1}^{D}\mathbf{S}_{j',j}^{(t)}\big(1-\mathbf{S}_{j',j}^{(t)}\big)y\langle\mathbf{v}_1^{(t)},\mathbf{x}_{j'}\rangle\langle\boldsymbol{\xi}_i,\mathbf{x}_{j'}\rangle\langle\boldsymbol{\xi}_{i'},\mathbf{x}_j\rangle\mathbf{1}_{\{E_t^c\}}\bigg]^2}$$

$$\leq \sqrt{\sum_{i=1}^{d}\sum_{i'=1}^{d}\mathbb{E}\bigg[\Big(\sum_{j=1}^{D}\sum_{j'=1}^{D}\mathbf{S}_{j',j}^{(t)}\big(1-\mathbf{S}_{j',j}^{(t)}\big)\langle\mathbf{v}_1^{(t)},\mathbf{x}_{j'}\rangle\langle\boldsymbol{\xi}_i,\mathbf{x}_{j'}\rangle\langle\boldsymbol{\xi}_{i'},\mathbf{x}_j\rangle\Big)^2\bigg]}\sqrt{\mathbb{P}(E_t^c)}$$

$$\leq \sqrt{\sum_{i=1}^{d}\sum_{i'=1}^{d}\mathbb{E}\bigg[\Big(\sum_{j=1}^{D}\langle\boldsymbol{\xi}_{i'},\mathbf{x}_j\rangle\Big)^2\Big(\sum_{j'=1}^{D}\langle\mathbf{v}_1^{(t)},\mathbf{x}_{j'}\rangle^2\langle\boldsymbol{\xi}_i,\mathbf{x}_{j'}\rangle^2\Big)\bigg]}\sqrt{\mathbb{P}(E_t^c)}$$

$$\leq \sqrt{\sum_{i=1}^{d}\sum_{i'=1}^{d}\mathbb{E}\bigg[\Big(\sum_{j=1}^{D}|\langle\boldsymbol{\xi}_{i'},\mathbf{x}_j\rangle|\Big)^2\Big(\sum_{j'=1}^{D}\langle\mathbf{v}_1^{(t)},\mathbf{x}_{j'}\rangle^2\langle\boldsymbol{\xi}_i,\mathbf{x}_{j'}\rangle^2\Big)\bigg]}\sqrt{\mathbb{P}(E_t^c)}$$

$$\leq \big(\alpha^{(t)} + \big\|\mathbf{v}_{1,\text{error}}^{(t)}\big\|_2\big)\sqrt{\sum_{i=1}^{d}\sum_{i'=1}^{d}\sum_{j_1=1}^{D}\sum_{j_2=1}^{D}\sum_{j'=1}^{D}\mathbb{E}\Big[|\langle\boldsymbol{\xi}_{i'},\mathbf{x}_{j_1}\rangle||\langle\boldsymbol{\xi}_{i'},\mathbf{x}_{j_2}\rangle|\langle\mathbf{v}_1^{(t)}/\|\mathbf{v}_1^{(t)}\|_2,\mathbf{x}_{j'}\rangle^2\langle\boldsymbol{\xi}_i,\mathbf{x}_{j'}\rangle^2\Big]}\sqrt{\mathbb{P}(E_t^c)}$$

$$\leq \frac{\sqrt{15}\sigma_x^3 dD^{\frac{3}{2}}\big(\alpha^{(t)} + \big\|\mathbf{v}_{1,\text{error}}^{(t)}\big\|_2\big)}{e^{C_8\sqrt{D}}} \leq \frac{1}{4e^{c\sqrt{D}}}.$$

The first and second inequality is derived by Cauchy–Schwarz inequality, and the facts that $\sum_{j'=1}^{D}\big(\mathbf{S}_{j',j}^{(t)}\big)^2 \leq 1$ and $\big(1-\mathbf{S}_{j',j}^{(t)}\big)^2, y^2, \ell'^2 \leq 1$. The last inequality holds since $\alpha^{(t)} \leq O((T^*)^{1/3} + D) \ll \exp(C_8\sqrt{D})$ and $\big\|\mathbf{v}_{1,\text{error}}^{(t)}\big\|_2 \leq e^{-C_4\sqrt{D}}$ by Lemma C.2 and definition of $T^*$ in Theorem 2.2. Similarly for $I_{2,2}$, we also have

$$\big\|I_{2,2}\big\|_2 = \left\|\mathbf{A}\mathbb{E}\bigg[\ell'^{(t)}\sum_{j=1}^{D}\sum_{j'=1}^{D}\sum_{j\neq j'}\mathbf{S}_{j',j}^{(t)}\mathbf{S}_{j'',j}^{(t)}y\langle\mathbf{v}_1^{(t)},\mathbf{x}_{j'}\rangle\begin{bmatrix}\langle\boldsymbol{\xi}_1,\mathbf{x}_{j''}\rangle\\\langle\boldsymbol{\xi}_2,\mathbf{x}_{j''}\rangle\\\vdots\\\langle\boldsymbol{\xi}_d,\mathbf{x}_{j''}\rangle\end{bmatrix}\big[\langle\boldsymbol{\xi}_1,\mathbf{x}_j\rangle,\langle\boldsymbol{\xi}_2,\mathbf{x}_j\rangle,\cdots,\langle\boldsymbol{\xi}_d,\mathbf{x}_j\rangle\big]\mathbf{1}_{\{E_t^c\}}\bigg]\mathbf{A}^\top\right\|_2$$

$$= \left\|\sum_{i=1}^{d}\sum_{i'=1}^{d}\mathbb{E}\bigg[\ell'^{(t)}\sum_{j=1}^{D}\sum_{j'=1}^{D}\sum_{j\neq j'}\mathbf{S}_{j',j}^{(t)}\mathbf{S}_{j'',j}^{(t)}y\langle\mathbf{v}_1^{(t)},\mathbf{x}_{j'}\rangle\langle\boldsymbol{\xi}_i,\mathbf{x}_{j''}\rangle\langle\boldsymbol{\xi}_{i'},\mathbf{x}_j\rangle\mathbf{1}_{\{E_t^c\}}\bigg]\boldsymbol{\xi}_i\boldsymbol{\xi}_{i'}\right\|_2$$

$$
= \sqrt{\sum_{i=1}^{d}\sum_{i'=1}^{d}\mathbb{E}\left[\ell'^{(t)}\sum_{j=1}^{D}\sum_{j'=1}^{D}\sum_{j\neq j'}\mathbf{S}_{j',j}^{(t)}\mathbf{S}_{j'',j}^{(t)}y\langle\mathbf{v}_1^{(t)},\mathbf{x}_{j'}\rangle\langle\boldsymbol{\xi}_i,\mathbf{x}_{j''}\rangle\langle\boldsymbol{\xi}_{i'},\mathbf{x}_j\rangle\mathbf{1}_{\{E_t^c\}}\right]^2}
$$

$$
\leq \sqrt{\sum_{i=1}^{d}\sum_{i'=1}^{d}\mathbb{E}\left[\left(\sum_{j=1}^{D}\sum_{j'=1}^{D}\sum_{j\neq j'}\mathbf{S}_{j',j}^{(t)}\mathbf{S}_{j'',j}^{(t)}\langle\mathbf{v}_1^{(t)},\mathbf{x}_{j'}\rangle\langle\boldsymbol{\xi}_i,\mathbf{x}_{j''}\rangle\langle\boldsymbol{\xi}_{i'},\mathbf{x}_j\rangle\right)^2\right]}\sqrt{\mathbb{P}(E_t^c)}
$$

$$
\leq \sqrt{\sum_{i=1}^{d}\sum_{i'=1}^{d}\mathbb{E}\left[\left(\sum_{j=1}^{D}\langle\boldsymbol{\xi}_{i'},\mathbf{x}_j\rangle\right)^2\left(\sum_{j'=1}^{D}\langle\mathbf{v}_1^{(t)},\mathbf{x}_{j'}\rangle^2\right)\left(\sum_{j''=1}^{D}\langle\boldsymbol{\xi}_i,\mathbf{x}_{j''}\rangle^2\right)\right]}\sqrt{\mathbb{P}(E_t^c)}
$$

$$
\leq \sqrt{\sum_{i=1}^{d}\sum_{i'=1}^{d}\mathbb{E}\left[\left(\sum_{j=1}^{D}|\langle\boldsymbol{\xi}_{i'},\mathbf{x}_j\rangle|\right)^2\left(\sum_{j'=1}^{D}\langle\mathbf{v}_1^{(t)},\mathbf{x}_{j'}\rangle^2\right)\left(\sum_{j''=1}^{D}\langle\boldsymbol{\xi}_i,\mathbf{x}_{j''}\rangle^2\right)\right]}\sqrt{\mathbb{P}(E_t^c)}
$$

$$
= \|\mathbf{v}_1^{(t)}\|_2\sqrt{\sum_{i=1}^{d}\sum_{i'=1}^{d}\sum_{j_1=1}^{D}\sum_{j_2=1}^{D}\sum_{j'=1}^{D}\sum_{j''=1}^{D}\mathbb{E}\left[|\langle\boldsymbol{\xi}_{i'},\mathbf{x}_{j_1}\rangle||\langle\boldsymbol{\xi}_{i'},\mathbf{x}_{j_2}\rangle|\langle\mathbf{v}_1^{(t)}/\|\mathbf{v}_1^{(t)}\|_2,\mathbf{x}_{j'}\rangle^2\langle\boldsymbol{\xi}_i,\mathbf{x}_{j''}\rangle^2\right]}\sqrt{\mathbb{P}(E_t^c)}
$$

$$
\leq \frac{\sqrt{15}\sigma_x^3 dD^2(\alpha^{(t)}+\|\mathbf{v}_{1,\text{error}}^{(t)}\|_2)}{e^{C_8\sqrt{D}}}\leq\frac{1}{4e^{c\sqrt{D}}}.
$$

Combining all the preceding results, we have

$$
\left\|\mathbf{W}_{1,1,\text{error}}^{(t+1)}\right\|_2 \leq \left\|\mathbf{W}_{1,1,\text{error}}^{(t)}\right\|_2 + \eta\|\nabla_{\mathbf{W}_{1,1}}\mathcal{L}(\mathbf{v}^{(t)},\mathbf{W}^{(t)})\|_2
$$

$$
\leq \left\|\mathbf{W}_{1,1,\text{error}}^{(t)}\right\|_2 + \eta\|I_{1,1}\|_2 + \eta\|I_{1,2}\|_2 + \eta\|I_{2,1}\|_2 + \eta\|I_{2,2}\|_2
$$

$$
\leq \frac{\eta t}{e^{c\sqrt{D}}} + \frac{4\eta}{4e^{c\sqrt{D}}} = \frac{\eta(t+1)}{e^{c\sqrt{D}}},
$$

which completes the proof for (C.13). Further by our definition of $T^*$ in Theorem 2.2, we can obtain

$$
\left\|\mathbf{W}_{1,1,\text{error}}^{(t)}\right\|_2 \leq \frac{T^*\eta}{e^{c\sqrt{D}}} \leq e^{-C_7\sqrt{D}}.
$$

This completes the proof for (C.7). $\qquad\qquad\square$

Next, we present the proof for Lemma C.4.

*Proof of Lemma C.4.* By calculations in Lemma C.12, we have obtained that $\mathbf{W}_{2,2}^{(2)}$ follows (C.8) with $\mathbf{W}_{2,2,\text{error}}^{(2)} = 0$. Instead of directly proving (C.9), we prove the following inequality,

$$
\|\mathbf{W}_{2,2,\text{error}}^{(t)}\|_2 \leq \frac{\eta t}{e^{c\sqrt{D}}}. \tag{C.14}
$$

for some constant $c$ solely depending on $\eta$ and $\sigma_x$. Therefore by induction, we assume it holds for $t$, and prove it still holds for $t+1$. Actually, it suffices to show that $\left\|\nabla_{\mathbf{W}_{2,2}}\mathcal{L}(\mathbf{v}^{(t)},\mathbf{W}^{(t)})\right\|_2 \leq e^{-c\sqrt{D}}$. By (C.2), we have

$$
\nabla_{\mathbf{W}_{2,2}}\mathcal{L}(\mathbf{v}^{(t)},\mathbf{W}^{(t)})
$$

$$
=\mathbb{E}\left[\ell'^{(t)}\sum_{j=1}^{D}\sum_{j'=1}^{D}\sum_{j''\neq j'}\mathbf{S}_{j',j}^{(t)}\mathbf{S}_{j'',j}^{(t)}y\langle\mathbf{v}_1^{(t)},\mathbf{x}_{j'}\rangle(\mathbf{p}_{j'}-\mathbf{p}_{j''})\mathbf{p}_j^\top\right]
$$

$$
=\underbrace{\mathbb{E}\left[\ell'^{(t)}\sum_{j=1}^{D}\sum_{j'=1}^{D}\mathbf{S}_{j',j}^{(t)}(1-\mathbf{S}_{j',j}^{(t)})y\langle\mathbf{v}_1^{(t)},\mathbf{x}_{j'}\rangle\mathbf{p}_{j'}\mathbf{p}_j^\top\right]}_{I_1} - \underbrace{\mathbb{E}\left[\ell'^{(t)}\sum_{j=1}^{D}\sum_{j'=1}^{D}\sum_{j''\neq j'}\mathbf{S}_{j',j}^{(t)}\mathbf{S}_{j'',j}^{(t)}y\langle\mathbf{v}_1^{(t)},\mathbf{x}_{j'}\rangle\mathbf{p}_{j''}\mathbf{p}_j^\top\right]}_{I_2}
$$

$$
= \underbrace{\sum_{j=1}^{D}\sum_{j'=1}^{D} \mathbb{E}\Big[\ell'^{(t)}\mathbf{S}_{j',j}^{(t)}\big(1-\mathbf{S}_{j',j}^{(t)}\big)y\langle\mathbf{v}_1^{(t)},\mathbf{x}_{j'}\rangle\mathbf{1}_{\{E_t\}}\Big]\mathbf{p}_{j'}\mathbf{p}_j^\top}_{I_{1,1}} + \underbrace{\sum_{j=1}^{D}\sum_{j'=1}^{D} \mathbb{E}\Big[\ell'^{(t)}\mathbf{S}_{j',j}^{(t)}\big(1-\mathbf{S}_{j',j}^{(t)}\big)y\langle\mathbf{v}_1^{(t)},\mathbf{x}_{j'}\rangle\mathbf{1}_{\{E_t^c\}}\Big]\mathbf{p}_{j'}\mathbf{p}_j^\top}_{I_{1,2}}
$$

$$
- \underbrace{\sum_{j=1}^{D}\sum_{j'=1}^{D} \mathbb{E}\Big[\ell'^{(t)}\mathbf{S}_{j',j}^{(t)}\big(1-\mathbf{S}_{j',j}^{(t)}\big)y\langle\mathbf{v}_1^{(t)},\mathbf{x}_{j'}\rangle\mathbf{1}_{\{E_t\}}\Big]\mathbf{p}_{j'}\mathbf{p}_j^\top}_{I_{2,1}} - \underbrace{\sum_{j=1}^{D}\sum_{j'=1}^{D}\sum_{j''\neq j'} \mathbb{E}\Big[\ell'^{(t)}\mathbf{S}_{j',j}^{(t)}\mathbf{S}_{j'',j}^{(t)}y\langle\mathbf{v}_1^{(t)},\mathbf{x}_{j'}\rangle\mathbf{1}_{\{E_t^c\}}\Big]\mathbf{p}_{j''}\mathbf{p}_j^\top}_{I_{2,2}}
$$

For $I_{1,1}$, we have

$$
\big\|I_{1,1}\big\|_2 \leq \sum_{j=1}^{D}\sum_{j'=1}^{D} \mathbb{E}\Big[\mathbf{S}_{j',j}^{(t)}\big(1-\mathbf{S}_{j',j}^{(t)}\big)\Big(\alpha^{(t)}|\langle\mathbf{v}^*,\mathbf{x}_{j'}\rangle| + \big\|\mathbf{v}_{1,\text{error}}^{(t)}\big\|_2\|\mathbf{x}_{j'}\|_2\Big)\mathbf{1}_{\{E_t\}}\Big]\|\mathbf{p}_{j'}\|_2\|\mathbf{p}_j\|_2
$$

$$
\leq \frac{D^2(D+1)\big(c_3\alpha^{(t)}D^{1/4} + \big\|\mathbf{v}_{1,\text{error}}^{(t)}\big\|_2(\sqrt{d}+\sqrt{D})\big)}{e^{C_9 D}} \leq \frac{1}{4e^{c\sqrt{D}}}.
$$

The first inequality is by triangle inequality and $|y|, |\ell'| \leq 1$. The second inequality holds because $|\langle\mathbf{v}^*,\mathbf{x}_j\rangle| \leq c_3 D^{1/4}$, $\|\mathbf{x}_j\|_2 \leq \sigma_x(\sqrt{d}+\sqrt{D})$, $\mathbf{S}_{j^*,j}^{(t)} \geq 1 - D\exp(-C_9 D)$ and $\mathbf{S}_{j',j}^{(t)} \leq \exp(-C_9 D)$ when $\mathbf{1}_{\{E_t\}} = 1$, and $\|\mathbf{p}_j\|_2 = \sqrt{(D+1)/2}$ by Lemma E.18. The second inequality holds since $\alpha^{(t)} \leq O((T^*)^{1/3}+D) \ll \exp(C_9 D)$ and $\big\|\mathbf{v}_{1,\text{error}}^{(t)}\big\|_2 \leq e^{-C_4\sqrt{D}}$ by Lemma C.2 and definition of $T^*$ in Theorem 2.2. Similarly, for $I_{2,1}$, we can obtain that

$$
\big\|I_{2,1}\big\|_2 \leq \sum_{j=1}^{D}\sum_{j'=1}^{D} \mathbb{E}\Big[\mathbf{S}_{j',j}^{(t)}\mathbf{S}_{j'',j}^{(t)}\Big(\alpha^{(t)}\langle\mathbf{v}^*,\mathbf{x}_{j'}\rangle + \big\|\mathbf{v}_{1,\text{error}}^{(t)}\big\|_2\|\mathbf{x}_{j'}\|_2\Big)\mathbf{1}_{\{E_t\}}\Big]\|\mathbf{p}_{j''}\|_2\|\mathbf{p}_j\|_2
$$

$$
\leq \frac{2D^2(D+1)\big(c_3\alpha^{(t)}D^{1/4} + \big\|\mathbf{v}_{1,\text{error}}^{(t)}\big\|_2(\sqrt{d}+\sqrt{D})\big)}{e^{C_9 D}} \leq \frac{1}{4e^{c\sqrt{D}}}.
$$

By applying Cauchy–Schwarz inequality, we can obtain an upper bound for $I_{1,2}$ as

$$
\big\|I_{1,2}\big\|_2 = \left\|\sum_{j=1}^{D}\sum_{j'=1}^{D} \mathbb{E}\Big[\ell'^{(t)}\mathbf{S}_{j',j}^{(t)}\big(1-\mathbf{S}_{j',j}^{(t)}\big)y\langle\mathbf{v}_1^{(t)},\mathbf{x}_{j'}\rangle\mathbf{1}_{\{E_t^c\}}\Big]\mathbf{p}_{j'}\mathbf{p}_j^\top\right\|_2
$$

$$
= \sqrt{\sum_{j=1}^{D}\sum_{j'=1}^{D} \mathbb{E}\Big[\ell'^{(t)}\mathbf{S}_{j',j}^{(t)}\big(1-\mathbf{S}_{j',j}^{(t)}\big)y\langle\mathbf{v}_1^{(t)},\mathbf{x}_{j'}\rangle\mathbf{1}_{\{E_t^c\}}\Big]^2\|\mathbf{p}_{j'}\|_2^2\|\mathbf{p}_j\|_2^2}
$$

$$
\leq \frac{(D+1)\sqrt{\mathbb{P}(E_t^c)}}{2}\sqrt{\sum_{j=1}^{D}\sum_{j'=1}^{D} \mathbb{E}\Big[\big(\mathbf{S}_{j',j}^{(t)}\big)^2\langle\mathbf{v}_1^{(t)},\mathbf{x}_{j'}\rangle^2\Big]}
$$

$$
\leq \frac{(D+1)\big(\alpha^{(t)} + \big\|\mathbf{v}_{1,\text{error}}^{(t)}\big\|_2\big)\sqrt{\mathbb{P}(E_t^c)}}{2}\sqrt{\sum_{j=1}^{D}\sqrt{\sum_{j'=1}^{D} \mathbb{E}\big[\langle\mathbf{v}_1^{(t)}/\|\mathbf{v}_1^{(t)}\|_2,\mathbf{x}_{j'}\rangle^4\big]}}
$$

$$
\leq \frac{3^{\frac{1}{4}}\sigma_x(D+1)D^{\frac{3}{4}}\big(\alpha^{(t)} + \big\|\mathbf{v}_{1,\text{error}}^{(t)}\big\|_2\big)}{2e^{C_8\sqrt{D}}} \leq \frac{1}{4e^{c\sqrt{D}}}.
$$

The first inequality is derived by Cauchy–Schwarz inequality, $\|\mathbf{p}_j\|_2 = \sqrt{(D+1)/2}$ by Lemma E.18, The second inequality is derived by Cauchy–Schwarz inequality, and $\sum_{j'=1}^{D}\big(\mathbf{S}_{j',j}^{(t)}\big)^4 \leq 1$. The last inequality holds since $\alpha^{(t)} \leq O((T^*)^{1/3}+D) \ll \exp(C_8\sqrt{D})$ and $\big\|\mathbf{v}_{1,\text{error}}^{(t)}\big\|_2 \leq e^{-C_4\sqrt{D}}$ by Lemma C.2 and definition of $T^*$ in Theorem 2.2. Similarly for $I_{2,2}$, we also have

$$
\big\|I_{2,2}\big\|_2 = \left\|\sum_{j=1}^{D}\sum_{j'=1}^{D}\sum_{j''\neq j'} \mathbb{E}\Big[\ell'^{(t)}\mathbf{S}_{j',j}^{(t)}\mathbf{S}_{j'',j}^{(t)}y\langle\mathbf{v}_1^{(t)},\mathbf{x}_{j'}\rangle\mathbf{1}_{\{E_t^c\}}\Big]\mathbf{p}_{j''}\mathbf{p}_j^\top\right\|_2
$$

$$= \sqrt{\sum_{j=1}^{D}\sum_{j'=1}^{D}\sum_{j''\neq j'}\mathbb{E}\Big[\ell'^{(t)}\mathbf{S}_{j',j}^{(t)}\mathbf{S}_{j'',j}^{(t)}y\langle\mathbf{v}_1^{(t)},\mathbf{x}_{j'}\rangle\mathbf{1}_{\{E_t^c\}}\Big]^2\|\mathbf{p}_{j'}\|_2^2\|\mathbf{p}_j\|_2^2}$$

$$\leq \frac{(D+1)\sqrt{\mathbb{P}(E_t^c)}}{2}\sqrt{\sum_{j=1}^{D}\sum_{j'=1}^{D}\mathbb{E}\Big[(\mathbf{S}_{j',j}^{(t)})^2\langle\mathbf{v}_1^{(t)},\mathbf{x}_{j'}\rangle^2\sum_{j''\neq j'}(\mathbf{S}_{j'',j}^{(t)})^2\Big]}$$

$$\leq \frac{(D+1)\big(\alpha^{(t)}+\|\mathbf{v}_{1,\mathrm{error}}^{(t)}\|_2\big)\sqrt{\mathbb{P}(E_t^c)}}{2}\sqrt{\sum_{j=1}^{D}\sqrt{\sum_{j'=1}^{D}\mathbb{E}\big[\langle\mathbf{v}_1^{(t)}/\|\mathbf{v}_1^{(t)}\|_2,\mathbf{x}_{j'}\rangle^4\big]}}$$

$$\leq \frac{3^{\frac{1}{4}}\sigma_x(D+1)D^{\frac{3}{4}}\big(\alpha^{(t)}+\|\mathbf{v}_{1,\mathrm{error}}^{(t)}\|_2\big)}{2e^{C_8\sqrt{D}}}\leq\frac{1}{4e^{c\sqrt{D}}}.$$

Combining all the preceding results, we have

$$\Big\|\mathbf{W}_{2,2,\mathrm{error}}^{(t+1)}\Big\|_2 \leq \Big\|\mathbf{W}_{2,2,\mathrm{error}}^{(t)}\Big\|_2 + \eta\|\nabla_{\mathbf{W}_{2,2}^{(t)}}\mathcal{L}\|_2$$

$$\leq \Big\|\mathbf{W}_{2,2,\mathrm{error}}^{(t)}\Big\|_2 + \eta\big\|I_{1,1}\big\|_2 + \eta\big\|I_{1,2}\big\|_2 + \eta\big\|I_{2,1}\big\|_2 + \eta\big\|I_{2,2}\big\|_2$$

$$\leq \frac{\eta t}{e^{c\sqrt{D}}} + \frac{4\eta}{4e^{c\sqrt{D}}} = \frac{\eta(t+1)}{e^{c\sqrt{D}}},$$

which completes the proof for (C.14). Further by our definition of $T^*$ in Theorem 2.2, we can obtain

$$\Big\|\mathbf{W}_{2,2,\mathrm{error}}^{(t)}\Big\|_2 \leq \frac{T^*\eta}{e^{c\sqrt{D}}} \leq e^{-C_7\sqrt{D}}.$$

This completes the proof for (C.9). □

Finally, we provide the proof for Lemma 4.3

*Proof of Lemma 4.3.* By Lemma C.2 and Proposition C.1, we can re-write $yf(\mathbf{Z},\mathbf{W}^{(t)},\mathbf{v}^{(t)})$ as

$$yf(\mathbf{Z},\mathbf{W}^{(t)},\mathbf{v}^{(t)}) = \alpha^{(t)}\sum_{j=1}^{D}\sum_{j'=1}^{D}y\langle\mathbf{v}^*,\mathbf{x}_{j'}\rangle\mathbf{S}_{j',j}^{(t)} + \sum_{j=1}^{D}\sum_{j'=1}^{D}y\langle\mathbf{v}_{1,\mathrm{error}}^{(t)},\mathbf{x}_{j'}\rangle\mathbf{S}_{j',j}^{(t)}.$$

By the fact that $\sum_{j'=1}^{D}(\mathbf{S}_{j',j}^{(t)})^2 \leq 1$ and Cauchy-Schwarz inequality, we can get a worst-case lower bound for $yf(\mathbf{Z},\mathbf{W}^{(t)},\mathbf{v}^{(t)})$ as

$$yf(\mathbf{Z},\mathbf{W}^{(t)},\mathbf{v}^{(t)}) \geq -\alpha^{(t)}\sum_{j=1}^{D}\left(\Big(\sum_{j'=1}^{D}\langle\mathbf{v}^*,\mathbf{x}_{j'}\rangle^2\Big)^{\frac{1}{2}}\Big(\sum_{j'=1}^{D}(\mathbf{S}_{j',j}^{(t)})^2\Big)^{\frac{1}{2}}\right)$$

$$-\|\mathbf{v}_{1,\mathrm{error}}^{(t)}\|_2\sum_{j=1}^{D}\left(\Big(\sum_{j'=1}^{D}\Big\langle\frac{\mathbf{v}_{1,\mathrm{error}}^{(t)}}{\|\mathbf{v}_{1,\mathrm{error}}^{(t)}\|_2},\mathbf{x}_{j'}\Big\rangle^2\Big)^{\frac{1}{2}}\Big(\sum_{j'=1}^{D}(\mathbf{S}_{j',j}^{(t)})^2\Big)^{\frac{1}{2}}\right)$$

$$\geq -D\alpha^{(t)}\Big(\sum_{j'=1}^{D}\langle\mathbf{v}^*,\mathbf{x}_{j'}\rangle^2\Big)^{\frac{1}{2}} - D\|\mathbf{v}_{1,\mathrm{error}}^{(t)}\|_2\Big(\sum_{j'=1}^{D}\Big\langle\frac{\mathbf{v}_{1,\mathrm{error}}^{(t)}}{\|\mathbf{v}_{1,\mathrm{error}}^{(t)}\|_2},\mathbf{x}_{j'}\Big\rangle^2\Big)^{\frac{1}{2}}.$$

Since $\langle\mathbf{v}^*,\mathbf{v}_{1,\mathrm{error}}^{(t)}\rangle = 0$, we can derive that $\sum_{j'=1}^{D}\langle\mathbf{v}^*,\mathbf{x}_{j'}\rangle^2$ and $\sum_{j'=1}^{D}\Big\langle\frac{\mathbf{v}_{1,\mathrm{error}}^{(t)}}{\|\mathbf{v}_{1,\mathrm{error}}^{(t)}\|_2},\mathbf{x}_{j'}\Big\rangle^2$ are i.i.d. random variables and $\frac{1}{\sigma_x^2}\sum_{j'=1}^{D}\langle\mathbf{v}^*,\mathbf{x}_{j'}\rangle^2, \frac{1}{\sigma_x^2}\sum_{j'=1}^{D}\Big\langle\frac{\mathbf{v}_{1,\mathrm{error}}^{(t)}}{\|\mathbf{v}_{1,\mathrm{error}}^{(t)}\|_2},\mathbf{x}_{j'}\Big\rangle^2 \sim \chi_D^2$ by the properties of Gaussian distribution. Besides, when $E_t$ holds, we also have

$$yf(\mathbf{Z},\mathbf{W}^{(t)},\mathbf{v}^{(t)}) \geq \alpha^{(t)}y\langle\mathbf{v}^*,\mathbf{x}_{j^*}\rangle\sum_{j=1}^{D}\mathbf{S}_{j^*,j}^{(t)} - \alpha^{(t)}\sum_{j'\neq j^*}\|\mathbf{x}_{j'}\|_2\sum_{j=1}^{D}\mathbf{S}_{j',j}^{(t)} - \|\mathbf{v}_{1,\mathrm{error}}^{(t)}\|_2\sum_{j'=1}^{D}\|\mathbf{x}_{j'}\|_2\sum_{j=1}^{D}\mathbf{S}_{j',j}^{(t)}$$

$$\geq D\alpha^{(t)}\left(1 - \frac{D}{e^{C_9 D}}\right)y\langle\mathbf{v}^*, \mathbf{x}_{j^*}\rangle - \frac{\sigma_x\alpha^{(t)}D^2\left(\sqrt{d}+\sqrt{D}\right)}{e^{C_9 D}} - \frac{\sigma_x D\left(\sqrt{d}+\sqrt{D}\right)}{e^{C_4\sqrt{D}}}$$

$$\geq D\alpha^{(t)}\left(1 - \frac{D}{e^{cD}}\right)y\langle\mathbf{v}^*, \mathbf{x}_{j^*}\rangle - 1.$$

The second inequality holds because $E_t$ implies that $\mathbf{S}_{j^*,j}^{(t)} \geq 1 - De^{-C_9 D}$, $\mathbf{S}_{j',j}^{(t)} \leq e^{-C_9 D}$, and $\|\mathbf{x}_{j'}\|_2 \leq \sigma_x(\sqrt{d}+\sqrt{D})$ by Lemma C.5 and Lemma E.16. Besides, Lemma 4.2 guarantees that $\|\mathbf{v}_{1,\text{error}}^{(t)}\|_2 \leq \frac{1}{e^{C_4\sqrt{D}}}$. The last inequality holds because $\alpha^{(t)} \leq O\left((T^*)^{1/3} + D\right)$ by Lemma C.2, which is much smaller than $e^{C_9 D}$ by our definition of $T^*$ in Theorem 2.2. Therefore, the last two terms are much smaller than 1. Similarly, we can also obtain that

$$yf(\mathbf{Z}, \mathbf{W}^{(t)}, \mathbf{v}^{(t)}) \leq \alpha^{(t)}y\langle\mathbf{v}^*, \mathbf{x}_{j^*}\rangle\sum_{j=1}^{D}\mathbf{S}_{j^*,j}^{(t)} + \alpha^{(t)}\sum_{j'\neq j^*}\|\mathbf{x}_{j'}\|_2\sum_{j=1}^{D}\mathbf{S}_{j',j}^{(t)} + \|\mathbf{v}_{1,\text{error}}^{(t)}\|_2\sum_{j'=1}^{D}\|\mathbf{x}_{j'}\|_2\sum_{j=1}^{D}\mathbf{S}_{j',j}^{(t)}$$

$$\leq D\alpha^{(t)}y\langle\mathbf{v}^*, \mathbf{x}_{j^*}\rangle + \frac{\sigma_x\alpha^{(t)}D^2\left(\sqrt{d}+\sqrt{D}\right)}{e^{C_9 D}} + \frac{\sigma_x D\left(\sqrt{d}+\sqrt{D}\right)}{e^{C_4\sqrt{D}}}$$

$$\leq D\alpha^{(t)}y\langle\mathbf{v}^*, \mathbf{x}_{j^*}\rangle + 1.$$

This completes the proof. $\square$

## D  PROOF OF THEOREM 3.2

This section provides a complete proof for Theorem 3.2. Before we demonstrate the proof for Theorem 3.2, we first introduce and prove several lemmas which will be utilized for further proof. The following two lemmas are very similar to Lemma C.3, Lemma C.4 and Lemma C.5.

**Lemma D.1.** Under the same conditions of Theorem 3.2, with probability at least $1 - ne^{-C_3\sqrt{D}}$ over the randomness of $\left(\mathbf{X}^{(i)}y^{(i)}\right)_{i=1}^{n}$, it holds that

$$\widetilde{\mathbf{W}}^{(i)} = \begin{bmatrix} \beta_1\mathbf{v}^*\mathbf{v}^{*\top} & \mathbf{0} \\ \mathbf{0} & \beta_2\left(\sum_{j\neq j^*}\left(\mathbf{p}_{j^*} - \mathbf{p}_j\right)\right)\left(\sum_{j=1}^{D}\mathbf{p}_j^\top\right) \end{bmatrix} + \widetilde{\mathbf{W}}_{\text{error}}^{(i)}, \tag{D.1}$$

where $|\beta_1| \leq c_1\sqrt{D}$, $\beta_2 \geq \frac{c_2}{D^2}$ for some constant $c_1, c_2$ solely depending on $\eta, \sigma_x$, and the error term satisfies that

$$\left\|\widetilde{\mathbf{W}}_{\text{error}}^{(i)}\right\|_2 \leq \exp\left(-C_{10}\sqrt{D}\right). \tag{D.2}$$

The coefficients $C_3, C_{10}$ are both positive constants solely depending on $\sigma_x, \widetilde{\sigma}_x$ and $\eta$.

**Lemma D.2.** Under the same conditions of Theorem 3.2, then with probability at least $1 - ne^{-C_3\sqrt{D}}$ over the randomness of $\left(\mathbf{X}^{(i)}y^{(i)}\right)_{i=1}^{n}$, it holds that

$$\mathbf{S}_{j^*,j}^{(i)} \geq 1 - D\exp(-C_{11}D);$$
$$\mathbf{S}_{j',j}^{(i)} \leq \exp(-C_{11}D)$$

for all $i \in [n]$, $j \in [D]$ and $j' \neq j^*$. The coefficients $C_3, C_{11}$ are both positive constants solely depending on $\sigma_x, \widetilde{\sigma}_x$ and $\eta$.

*Proof of Lemma D.2.* By Lemma D.1, there exists constants $c_1, c_2$ solely depending on $\eta, \sigma_x$ such that $|\beta_1| \leq c_1\sqrt{D}$ and $\beta_2 \geq c_2\frac{1}{D^2}$. Then further combining with Lemma E.17 and Lemma E.18, with probability at least $1 - ne^{-C_1\sqrt{D}}$, we can obtain that

$$\left(\mathbf{z}_{j'}^{(i)}\right)^\top\widetilde{\mathbf{W}}^{(i)}\mathbf{z}_j^{(i)} = \beta_1\langle\mathbf{v}^*, \mathbf{x}_{j'}^{(i)}\rangle\langle\mathbf{v}^*, \mathbf{x}_j^{(i)}\rangle - \frac{(D+1)^2}{4}\beta_2 + \left(\mathbf{z}_{j'}^{(i)}\right)^\top\widetilde{\mathbf{W}}_{\text{error}}^{(i)}\mathbf{z}_j^{(i)}$$

$$\leq \left|\beta_1\langle\mathbf{v}^*, \mathbf{x}_{j'}^{(i)}\rangle\langle\mathbf{v}^*, \mathbf{x}_j^{(i)}\rangle\right| + \left\|\widetilde{\mathbf{W}}_{\text{error}}^{(i)}\right\|_2\left\|\mathbf{z}_j^{(i)}\right\|_2\left\|\mathbf{z}_{j'}^{(i)}\right\|_2$$

$$\leq c_1\sqrt{D}\left(\sqrt{\frac{c_2}{13c_1\widetilde{\sigma}_x^2}}\widetilde{\sigma}_x D^{\frac{1}{4}}\right)^2 + \frac{2\widetilde{\sigma}_x^2(\sqrt{d}+\sqrt{D})^2}{e^{C_{10}\sqrt{D}}}$$

$$\leq \frac{c_2}{12}D$$

holds for all $j' \neq j^*$, $j \in [D]$ and $i \in [D]$, and

$$\left(\mathbf{z}_{j^*}^{(i)}\right)^\top \widetilde{\mathbf{W}}^{(i)}\mathbf{z}_j^{(i)} = \beta_1\langle\mathbf{v}^*,\mathbf{x}_{j^*}^{(i)}\rangle\langle\mathbf{v}^*,\mathbf{x}_j^{(i)}\rangle + \frac{(D-1)(D+1)^2}{4}\beta_2 + \left(\mathbf{z}_{j'}^{(i)}\right)^\top \widetilde{\mathbf{W}}_{\text{error}}^{(i)}\mathbf{z}_j^{(i)}$$

$$\geq \frac{c_2 D}{4} - \left|\beta_1\langle\mathbf{v}^*,\mathbf{x}_{j^*}^{(i)}\rangle\langle\mathbf{v}^*,\mathbf{x}_j^{(i)}\rangle\right| - \|\widetilde{\mathbf{W}}_{\text{error}}^{(i)}\|_2\|\mathbf{z}_j^{(i)}\|_2\|\mathbf{z}_{j'}^{(i)}\|_2$$

$$\geq \frac{c_2 D}{4} - c_1\sqrt{D}\left(\sqrt{\frac{c_2}{13c_1\sigma_x^2}}\sigma_x D^{\frac{1}{4}}\right)^2 - \frac{2\widetilde{\sigma}_x^2(\sqrt{d}+\sqrt{D})^2}{e^{C_{10}\sqrt{D}}}$$

$$\geq \frac{c_2}{6}D$$

holds for all $j \in [D]$ and $i \in [D]$. Therefore, we can obtain that

$$\mathbf{S}_{j^*,j}^{(i)} \geq \frac{e^{\frac{c_2}{6}D}}{e^{\frac{c_2}{6}D} + (D-1)e^{\frac{c_2}{12}D}} \geq 1 - De^{-\frac{c_2}{12}D};$$

$$\mathbf{S}_{j',j}^{(i)} \leq \frac{e^{\frac{c_2}{12}D}}{e^{\frac{c_2}{6}D} + (D-1)e^{\frac{c_2}{12}D}} \leq e^{-\frac{c_2}{12}D}.$$

This completes the proof. $\qquad\square$

*Proof of Lemma D.1.* By Lemma C.3 and Lemma C.4, we have obtained that $\widetilde{\mathbf{W}}^{(1)} = \mathbf{W}^{(T^*)}$ follows (D.1) with $\|\widetilde{\mathbf{W}}_{\text{error}}^{(1)}\|_2 \leq \max\left\{\|\mathbf{W}_{1,1,\text{error}}^{(T^*)}\|_2, \|\mathbf{W}_{2,2,\text{error}}^{(T^*)}\|_2\right\} \leq \exp\left(-C_7\sqrt{D}\right)$. Instead of directly proving (D.2), we prove the following inequality,

$$\|\widetilde{\mathbf{W}}_{\text{error}}^{(i)}\|_2 \leq \exp\left(-C_7\sqrt{D}\right) + \frac{\widetilde{\eta}i}{\exp(cD)}, \tag{D.3}$$

where $c$ is a constant solely depending on $\sigma_x, \widetilde{\sigma}_x, \eta$. Based on our previous discussion, we know it holds for $i = 1$. Therefore by induction, we assume it holds for $i \leq n-1$, and prove it still holds for $i+1$. Actually, it suffices to show that $\|\nabla_{\mathbf{W}}\widetilde{\mathcal{L}}_i(\widetilde{\mathbf{v}}^{(i)}, \widetilde{\mathbf{W}}^{(i)})\|_2 \leq e^{-cD}$. And we firstly provide an upper-bound for $\|\widetilde{\mathbf{v}}^{(i)}\|_2$ as

$$\|\widetilde{\mathbf{v}}^{(i)}\|_2 = \left\|\widetilde{\eta}\sum_{k=1}^{i}\sum_{j=1}^{D}\sum_{j'=1}^{D}\ell'^{(i)}y^{(i)}\cdot\mathbf{z}_j^{(i)}\mathbf{S}_{j,j'}^{(t)}\right\|_2$$

$$\leq \left(\sqrt{2}\widetilde{\sigma}_x + 1\right)\widetilde{\eta}nD\left(\sqrt{d}+\sqrt{D}\right) \leq \frac{n}{\left(\sqrt{2}\widetilde{\sigma}_x + 1\right)D\left(\sqrt{d}+\sqrt{D}\right)}.$$

The first inequality is because $\|\mathbf{z}_j^{(i)}\| \leq \left(\sqrt{2}\widetilde{\sigma}_x + 1\right)\left(\sqrt{d}+\sqrt{D}\right)$ by Lemma E.17 and Lemma E.18. The last inequality holds by the scale of $\widetilde{\eta}$ from the conditions of Theorem 3.2. Based on this upper-bound, we can further demonstrate the upper-bound of $\|\nabla_{\mathbf{W}}\widetilde{\mathcal{L}}_i(\widetilde{\mathbf{v}}^{(i)}, \widetilde{\mathbf{W}}^{(i)})\|_2$ Then by (C.2), we have

$$\nabla_{\mathbf{W}}\widetilde{\mathcal{L}}_i(\widetilde{\mathbf{v}}^{(i)}, \widetilde{\mathbf{W}}^{(i)})$$

$$= \ell'^{(i)}\sum_{j=1}^{D}\sum_{j'=1}^{D}\sum_{j''\neq j'}\mathbf{S}_{j',j}^{(i)}\mathbf{S}_{j'',j}^{(i)}y\langle\widetilde{\mathbf{v}}^{(i)},\mathbf{z}_{j'}^{(i)}\rangle\left(\mathbf{z}_{j'}^{(i)} - \mathbf{z}_{j''}^{(i)}\right)\left(\mathbf{z}_j^{(i)}\right)^\top$$

$$= \underbrace{\ell'^{(i)}\sum_{j=1}^{D}\sum_{j'=1}^{D}\mathbf{S}_{j',j}^{(i)}\left(1 - \mathbf{S}_{j',j}^{(i)}\right)y\langle\widetilde{\mathbf{v}}^{(i)},\mathbf{z}_{j'}^{(i)}\rangle\mathbf{z}_{j'}^{(i)}\left(\mathbf{z}_j^{(i)}\right)^\top}_{I_1} - \underbrace{\ell'^{(i)}\sum_{j=1}^{D}\sum_{j'=1}^{D}\sum_{j''\neq j'}\mathbf{S}_{j',j}^{(i)}\mathbf{S}_{j'',j}^{(i)}y\langle\widetilde{\mathbf{v}}^{(i)},\mathbf{z}_{j'}^{(i)}\rangle\mathbf{z}_{j''}^{(i)}\left(\mathbf{z}_j^{(i)}\right)^\top}_{I_2}.$$

For $I_1$, we have

$$\|I_1\|_2 \leq \sum_{j=1}^{D} \sum_{j'=1}^{D} \mathbf{S}_{j',j}^{(i)} (1 - \mathbf{S}_{j',j}^{(i)}) \|\widetilde{\mathbf{v}}^{(i)}\|_2 \|\mathbf{z}_{j'}^{(i)}\|_2^2 \|\mathbf{z}_j^{(i)}\|_2 \leq \frac{2(\sqrt{2}\widetilde{\sigma}_x + 1)^2 Dn(\sqrt{d} + \sqrt{D})^2}{e^{C_{11}D}} \leq \frac{1}{2e^{cD}}.$$

The first inequality is by triangle inequality and $|y|, |\ell'| \leq 1$. The second inequality holds because $\|\mathbf{z}_j^{(i)}\| \leq (\sqrt{2}\widetilde{\sigma}_x + 1)(\sqrt{d} + \sqrt{D})$, $\mathbf{S}_{j^*,j}^{(i)} \geq 1 - D\exp(-C_{11}D)$ and $\mathbf{S}_{j',j}^{(i)} \leq \exp(-C_{11}D)$ by Lemma D.2 and upper-bound for $\|\widetilde{\mathbf{v}}^{(i)}\|_2$. The second inequality holds by the condition concerning $n$ from Theorem 3.2. Similarly, for $I_2$, we can obtain that

$$\|I_2\|_2 \leq \sum_{j=1}^{D} \sum_{j'=1}^{D} \sum_{j'' \neq j'} \mathbf{S}_{j',j}^{(i)} \mathbf{S}_{j'',j}^{(i)} \|\widetilde{\mathbf{v}}^{(i)}\|_2 \|\mathbf{z}_{j'}^{(i)}\|_2 \|\mathbf{z}_{j''}^{(i)}\|_2 \|\mathbf{z}_j^{(i)}\|_2 \leq \frac{2(\sqrt{2}\widetilde{\sigma}_x + 1)^2 Dn(\sqrt{d} + \sqrt{D})^2}{e^{C_{11}D}} \leq \frac{1}{2e^{cD}}.$$

Combining all the preceding results, we have

$$
\begin{aligned}
\left\|\widetilde{\mathbf{W}}_{\text{error}}^{(i+1)}\right\|_2 &\leq \left\|\widetilde{\mathbf{W}}_{\text{error}}^{(i)}\right\|_2 + \widetilde{\eta}\left\|\nabla_{\mathbf{W}}\widetilde{\mathcal{L}}_i(\widetilde{\mathbf{v}}^{(i)}, \widetilde{\mathbf{W}}^{(i)})\right\|_2 \\
&\leq \left\|\widetilde{\mathbf{W}}_{\text{error}}^{(i)}\right\|_2 + \widetilde{\eta}\|I_1\|_2 + \widetilde{\eta}\|I_2\|_2 \\
&\leq \exp\left(-C_7\sqrt{D}\right) + \frac{\widetilde{\eta}i}{e^{cD}} + \frac{2\widetilde{\eta}}{2e^{cD}} = \exp\left(-C_7\sqrt{D}\right) + \frac{\widetilde{\eta}(i+1)}{e^{cD}},
\end{aligned}
$$

which proves that (D.3). Then by conditions concerning $n$ from Theorem 3.2, we further have

$$\left\|\widetilde{\mathbf{W}}_{\text{error}}^{(i)}\right\|_2 \leq \exp\left(-C_7\sqrt{D}\right) + \frac{\widetilde{\eta}i}{e^{cD}} \leq \exp(-C_{10}\sqrt{D}).$$

This completes the proof. $\qquad\square$

For further proof in this section, we introduce the following several notations. We denote $\widetilde{\mathbf{v}}^* = \arg\max_{\mathbf{v}:\|\mathbf{v}\|_2 \leq 1} \min_{i \in [n]} y^{(i)} \cdot \langle \mathbf{v}, \mathbf{x}_{j^*}^{(i)} \rangle$, and $\widehat{\mathbf{v}} = \left[(\frac{\log n}{\gamma D}\widetilde{\mathbf{v}}^*)^\top, \mathbf{0}_D^\top\right]^\top \in \mathbb{R}^{d+D}$. And we define $\widehat{\mathbf{W}}$ is an idealized matrix such that $\mathbf{S}_{j^*,j}^{(i)} = 1$ and $\mathbf{S}_{j',j}^{(i)} = 0$ for all $j' \neq j^*$ and $j \in [D]$ (Such a matrix exists by Lemma E.18). Furthermore, we define a proxy loss by $-\ell'$ as $\mathcal{R}_i(\widetilde{\mathbf{v}}, \widetilde{\mathbf{W}}) = -\ell'(y^{(i)} \cdot f(\mathbf{Z}^{(i)}, \widetilde{\mathbf{W}}, \widetilde{\mathbf{v}}))$ and $\mathcal{R}(\widetilde{\mathbf{v}}, \widetilde{\mathbf{W}}) = \mathbb{E}_{(\mathbf{X},y)\sim\widetilde{\mathcal{D}}}\left[-\ell'(y \cdot f(\mathbf{Z}, \widetilde{\mathbf{W}}, \widetilde{\mathbf{v}}))\right]$. Based on these notations, we can finish the remaining proof. Firstly, we prove that $\widetilde{\mathcal{L}}_i(\widetilde{\mathbf{v}}, \widetilde{\mathbf{W}})$ is nearly convex.

**Lemma D.3.** With probability at least $1 - ne^{-C_3\sqrt{D}}$, it holds that

$$\mathcal{L}_i(\widehat{\mathbf{v}}, \widehat{\mathbf{W}}) - \mathcal{L}_i(\widetilde{\mathbf{v}}^{(i)}, \widetilde{\mathbf{W}}^{(i)}) \geq \left\langle \nabla_{\mathbf{v}}\mathcal{L}_i(\widetilde{\mathbf{v}}^{(i)}, \widetilde{\mathbf{W}}^{(i)}), \widehat{\mathbf{v}} - \widetilde{\mathbf{v}}^{(i)} \right\rangle - \frac{1}{\gamma e^{C_{12}D}}$$

for all $i \in [n]$, where $C_{12}$ is a constant solely depending on $\sigma_x, \widetilde{\sigma}_x, \eta$

*Proof of Lemma D.3.* By convexity of $\ell$, we can obtain that

$$
\begin{aligned}
&\mathcal{L}_i(\widehat{\mathbf{v}}, \widehat{\mathbf{W}}) - \mathcal{L}_i(\widetilde{\mathbf{v}}^{(i)}, \widetilde{\mathbf{W}}^{(i)}) \\
=&\ell\left(y^{(i)}f(\mathbf{Z}^{(i)}, \widehat{\mathbf{W}}, \widehat{\mathbf{v}})\right) - \ell\left(y^{(i)}f(\mathbf{Z}^{(i)}, \widetilde{\mathbf{W}}^{(i)}, \widetilde{\mathbf{v}}^{(i)})\right) \\
\geq&\ell'\left(y^{(i)}f(\mathbf{Z}^{(i)}, \widetilde{\mathbf{W}}^{(i)}, \widetilde{\mathbf{v}}^{(i)})\right)y^{(i)}\left(f(\mathbf{Z}^{(i)}, \widehat{\mathbf{W}}, \widehat{\mathbf{v}}) - f(\mathbf{Z}^{(i)}, \widetilde{\mathbf{W}}^{(i)}, \widetilde{\mathbf{v}}^{(i)})\right) \\
=&\ell'^{(i)}y^{(i)}\left(\sum_{j=1}^{D}\sum_{j'=1}^{D}\mathbf{S}_{j,j'}^{(i)}\langle\widehat{\mathbf{v}} - \widetilde{\mathbf{v}}^{(i)}, \mathbf{z}_j^{(i)}\rangle\right) + \ell'^{(i)}y^{(i)}\left(D - \sum_{j'=1}^{D}\mathbf{S}_{j^*,j'}^{(i)}\right)\langle\widehat{\mathbf{v}}, \mathbf{z}_{j^*}^{(i)}\rangle + \ell'^{(i)}y^{(i)}\sum_{j\neq j^*}\sum_{j'=1}^{D}\mathbf{S}_{j,j'}^{(i)}\langle\widehat{\mathbf{v}}, \mathbf{z}_j^{(i)}\rangle \\
=&\left\langle\nabla_{\mathbf{v}}\mathcal{L}_i(\widetilde{\mathbf{v}}^{(i)}, \widetilde{\mathbf{W}}^{(i)}), \widehat{\mathbf{v}} - \widetilde{\mathbf{v}}^{(i)}\right\rangle + \underbrace{\ell'^{(i)}y^{(i)}\left(D - \sum_{j'=1}^{D}\mathbf{S}_{j^*,j'}^{(i)}\right)\langle\widehat{\mathbf{v}}, \mathbf{z}_{j^*}^{(i)}\rangle + \ell'^{(i)}y^{(i)}\sum_{j\neq j^*}\sum_{j'=1}^{D}\mathbf{S}_{j,j'}^{(i)}\langle\widehat{\mathbf{v}}, \mathbf{z}_j^{(i)}\rangle}_{I}
\end{aligned}
$$

And we can bound $|I|$ as

$$|I| \leq \left(D - \sum_{j'=1}^{D}\mathbf{S}_{j^*,j'}^{(i)}\right)\|\widehat{\mathbf{v}}\|_2\|\mathbf{z}_{j^*}^{(i)}\|_2 + \sum_{j\neq j^*}\sum_{j'=1}^{D}\mathbf{S}_{j,j'}^{(i)}\|\widehat{\mathbf{v}}\|_2\|\mathbf{z}_j^{(i)}\|_2$$

$$\leq \frac{2\big(\sqrt{2}\widetilde{\sigma}_x + 1\big)\big(\sqrt{d} + \sqrt{D}\big)D\log n}{\gamma e^{C_{11}D}} \leq \frac{1}{\gamma e^{C_{12}D}},$$

where the penultimate inequality is by Lemma D.2, Lemma E.17, Lemma E.18 and definition of $\widehat{\mathbf{v}}$. This completes the proof. $\qquad\square$

Based on the preceding Lemmas, we are now ready to prove Theorem 3.2.

*Proof of Theorem 3.2.* By updating rule (3.1), we can obtain that

$$\big\|\widetilde{\mathbf{v}}^{(i+1)} - \widehat{\mathbf{v}}\big\|_2^2 = \big\|\widetilde{\mathbf{v}}^{(i)} - \widehat{\mathbf{v}}\big\|_2^2 + 2\big\langle\widetilde{\mathbf{v}}^{(i+1)} - \widetilde{\mathbf{v}}^{(i)}, \widetilde{\mathbf{v}}^{(i)} - \widehat{\mathbf{v}}\big\rangle + \big\|\widetilde{\mathbf{v}}^{(i+1)} - \widetilde{\mathbf{v}}^{(i)}\big\|_2^2$$

$$= \big\|\widetilde{\mathbf{v}}^{(i)} - \widehat{\mathbf{v}}\big\|_2^2 - 2\widetilde{\eta}\big\langle\nabla_{\mathbf{v}}\mathcal{L}_i(\widetilde{\mathbf{v}}^{(i)}, \widetilde{\mathbf{W}}^{(i)}), \widetilde{\mathbf{v}}^{(i)} - \widehat{\mathbf{v}}\big\rangle + \widetilde{\eta}^2\big\|\nabla_{\mathbf{v}}\mathcal{L}_i(\widetilde{\mathbf{v}}^{(i)}, \widetilde{\mathbf{W}}^{(i)})\big\|_2^2$$

$$\leq \big\|\widetilde{\mathbf{v}}^{(i)} - \widehat{\mathbf{v}}\big\|_2^2 + 2\widetilde{\eta}\bigg(\mathcal{L}_i(\widehat{\mathbf{v}}, \widehat{\mathbf{W}}) - \mathcal{L}_i(\widetilde{\mathbf{v}}^{(i)}, \widetilde{\mathbf{W}}^{(i)}) + \frac{1}{\gamma e^{C_{12}D}}\bigg) + \widetilde{\eta}^2\big\|\nabla_{\mathbf{v}}\mathcal{L}_i(\widetilde{\mathbf{v}}^{(i)}, \widetilde{\mathbf{W}}^{(i)})\big\|_2^2$$

$$\leq \big\|\widetilde{\mathbf{v}}^{(i)} - \widehat{\mathbf{v}}\big\|_2^2 + 2\widetilde{\eta}\bigg(\mathcal{L}_i(\widehat{\mathbf{v}}, \widehat{\mathbf{W}}) - \mathcal{L}_i(\widetilde{\mathbf{v}}^{(i)}, \widetilde{\mathbf{W}}^{(i)}) + \frac{1}{\gamma e^{C_{12}D}}\bigg) + \widetilde{\eta}\mathcal{L}_i(\widetilde{\mathbf{v}}^{(i)}, \widetilde{\mathbf{W}}^{(i)})$$

$$\leq \big\|\widetilde{\mathbf{v}}^{(i)} - \widehat{\mathbf{v}}\big\|_2^2 - \widetilde{\eta}\mathcal{L}_i(\widetilde{\mathbf{v}}^{(i)}, \widetilde{\mathbf{W}}^{(i)}) + \frac{2\widetilde{\eta}}{\gamma e^{C_{12}D}} + \frac{2\widetilde{\eta}}{n}. \qquad\text{(D.4)}$$

The first inequality holds by Lemma D.3. The second inequality holds because

$$\widetilde{\eta}^2\big\|\nabla_{\mathbf{v}}\mathcal{L}_i(\widetilde{\mathbf{v}}^{(i)}, \widetilde{\mathbf{W}}^{(i)})\big\|_2^2 = \widetilde{\eta}^2\Big(\ell'\big(y^{(i)}f(\mathbf{Z}^{(i)}, \widetilde{\mathbf{v}}^{(i)}, \widetilde{\mathbf{W}}^{(i)})\big)\Big)^2\bigg\|\sum_{j=1}^D\sum_{j'=1}^D\mathbf{S}_{j,j'}^{(i)}\mathbf{z}_j^{(i)}\bigg\|_2$$

$$\leq \widetilde{\eta}^2\ell\big(y^{(i)}f(\mathbf{Z}^{(i)}, \widetilde{\mathbf{v}}^{(i)}, \widetilde{\mathbf{W}}^{(i)})\big)\big(\sqrt{2}\widetilde{\sigma}_x + 1\big)^2\big(\sqrt{d} + \sqrt{D}\big)^2 D^2 \leq \widetilde{\eta}\mathcal{L}_i(\widetilde{\mathbf{v}}^{(i)}, \widetilde{\mathbf{W}}^{(i)}),$$

where the first inequality holds by the facts that $-\ell'(x) \leq \ell(x)$, $-\ell'(x) \leq 1$ and $\big\|\mathbf{z}_j^{(i)}\big\|_2 \leq \big(\sqrt{2}\widetilde{\sigma}_x + 1\big)\big(\sqrt{d} + \sqrt{D}\big)$, and the last inequality holds by our condition that $\widetilde{\eta} \leq \frac{1}{(\sqrt{2}\widetilde{\sigma}_x + 1)^2(\sqrt{d} + \sqrt{D})^2 D^2}$. And the third inequality is because

$$\mathcal{L}_i(\widehat{\mathbf{v}}, \widehat{\mathbf{W}}) = \ell\big(y^{(i)}f(\mathbf{Z}^{(i)}, \widehat{\mathbf{W}}, \widehat{\mathbf{v}})\big) = \ell\bigg(\frac{\log n}{\gamma}y^{(i)}\langle\widetilde{\mathbf{v}}^*, \mathbf{x}_{j^*}^{(i)}\rangle\bigg) \leq \ell(\log n) \leq \frac{1}{n}.$$

By rearranging and take a telescoping sum on both side of (D.4), we obtain that

$$\frac{1}{n}\sum_{i=1}^n\mathcal{L}_i(\widetilde{\mathbf{v}}^{(i)}, \widetilde{\mathbf{W}}^{(i)}) \leq \frac{\big\|\widetilde{\mathbf{v}}^{(1)} - \widehat{\mathbf{v}}\big\|_2^2}{n\widetilde{\eta}} + \frac{2}{\gamma e^{C_{12}D}} + \frac{2}{n} \leq \frac{2(\sqrt{2}\widetilde{\sigma}_x + 1)^2\log^2 n(d + D)}{\gamma^2 n} + \frac{2}{\gamma e^{C_{12}D}} + \frac{2}{n}.$$

Further by applying the fact that $\gamma \leq \min_{i\in[n]}\|\mathbf{x}_{j^*}^{(i)}\|_2 \leq \sqrt{2}\widetilde{\sigma}_x\big(\sqrt{d} + \sqrt{D}\big)$ and $-\ell(x) \leq \ell(x)$, we can get

$$\frac{1}{n}\sum_{i=1}^n\mathcal{R}_i(\widetilde{\mathbf{v}}^{(i)}, \widetilde{\mathbf{W}}^{(i)}) \leq \frac{1}{n}\sum_{i=1}^n\mathcal{L}_i(\widetilde{\mathbf{v}}^{(i)}, \widetilde{\mathbf{W}}^{(i)}) \leq \frac{4(\sqrt{2}\widetilde{\sigma}_x + 1)^2\log^2 n(d + D)}{\gamma^2 n} + \frac{2}{\gamma e^{C_{12}D}}.$$
$$\text{(D.5)}$$

Notice that the quantity $\sum_{i\leq k}\big(\mathcal{R}_i(\widetilde{\mathbf{v}}^{(i)}, \widetilde{\mathbf{W}}^{(i)}) - \mathcal{R}(\widetilde{\mathbf{v}}^{(i)}, \widetilde{\mathbf{W}}^{(i)})\big)$ forms a martingale w.r.t. the filtration $\sigma\Big(\big(\mathbf{Z}^{(1)}, y^{(1)}\big), \cdots, \big(\mathbf{Z}^{(k-1)}, y^{(k-1)}\big)\Big) = \sigma^{(k)}$. The martingale difference $\mathcal{R}_i(\widetilde{\mathbf{v}}^{(i)}, \widetilde{\mathbf{W}}^{(i)}) - \mathcal{R}(\widetilde{\mathbf{v}}^{(i)}, \widetilde{\mathbf{W}}^{(i)})$ is bounded by 1 since $-\ell'$ is bounded. And we also have

$$\mathbb{E}\Big[\big(\mathcal{R}_i(\widetilde{\mathbf{v}}^{(i)}, \widetilde{\mathbf{W}}^{(i)}) - \mathcal{R}(\widetilde{\mathbf{v}}^{(i)}, \widetilde{\mathbf{W}}^{(i)})\big)^2\Big|\sigma^{(i)}\Big]$$

$$= \mathbb{E}\big[\mathcal{R}_i(\widetilde{\mathbf{v}}^{(i)}, \widetilde{\mathbf{W}}^{(i)})^2\big|\sigma^{(i)}\big] - 2\mathcal{R}(\widetilde{\mathbf{v}}^{(i)}, \widetilde{\mathbf{W}}^{(i)})\mathbb{E}\big[\mathcal{R}_i(\widetilde{\mathbf{v}}^{(i)}, \widetilde{\mathbf{W}}^{(i)})\big|\sigma^{(i)}\big] + \mathcal{R}(\widetilde{\mathbf{v}}^{(i)}, \widetilde{\mathbf{W}}^{(i)})^2$$

$$= \mathbb{E}\big[\mathcal{R}_i(\widetilde{\mathbf{v}}^{(i)}, \widetilde{\mathbf{W}}^{(i)})^2\big|\sigma^{(i)}\big] - \mathcal{R}(\widetilde{\mathbf{v}}^{(i)}, \widetilde{\mathbf{W}}^{(i)})^2 \leq \mathbb{E}\big[\mathcal{R}_i(\widetilde{\mathbf{v}}^{(i)}, \widetilde{\mathbf{W}}^{(i)})\big|\sigma^{(i)}\big] - \mathcal{R}(\widetilde{\mathbf{v}}^{(i)}, \widetilde{\mathbf{W}}^{(i)})^2 \leq \mathcal{R}(\widetilde{\mathbf{v}}^{(i)}, \widetilde{\mathbf{W}}^{(i)}).$$

Therefore by Theorem 1 in Beygelzimer et al. (2011) (a version of martingale difference concentration inequality), with probability at least $1 - \delta$ for any positive $\delta$, it holds that,

$$\frac{1}{n}\sum_{i=1}^{n}\big(\mathcal{R}(\widetilde{\mathbf{v}}^{(i)}, \widetilde{\mathbf{W}}^{(i)}) - \mathcal{R}_i(\widetilde{\mathbf{v}}^{(i)}, \widetilde{\mathbf{W}}^{(i)})\big) \leq \frac{e-2}{n}\sum_{i=1}^{n}\mathbb{E}\Big[\big(\mathcal{R}_i(\widetilde{\mathbf{v}}^{(i)}, \widetilde{\mathbf{W}}^{(i)}) - \mathcal{R}(\widetilde{\mathbf{v}}^{(i)}, \widetilde{\mathbf{W}}^{(i)})\big)^2\Big|\sigma^{(i)}\Big] + \frac{\log(1/\delta)}{n}$$

$$\leq \frac{e-2}{n}\sum_{i=1}^{n}\mathcal{R}(\widetilde{\mathbf{v}}^{(i)}, \widetilde{\mathbf{W}}^{(i)}) + \frac{\log(1/\delta)}{n}.$$

Rearranging on both sides and applying the results of (D.5), we get

$$\frac{1}{n}\sum_{i=1}^{n}\mathcal{R}(\widetilde{\mathbf{v}}^{(i)}, \widetilde{\mathbf{W}}^{(i)}) \leq \frac{4}{n}\sum_{i=1}^{n}\mathcal{R}_i(\widetilde{\mathbf{v}}^{(i)}, \widetilde{\mathbf{W}}^{(i)}) + \frac{4\log(1/\delta)}{n}$$

$$\leq \frac{16(\sqrt{2}\widetilde{\sigma}_x + 1)^2 \log^2 n(d+D)}{\gamma^2 n} + \frac{4\log(1/\delta)}{n} + \frac{8}{\gamma e^{C_{12}D}}.$$

Since $\mathbb{P}_{(\mathbf{X},y)\sim\widetilde{\mathcal{D}}}\big(y \cdot f(\mathbf{Z}, \widetilde{\mathbf{W}}, \widetilde{\mathbf{v}} \leq 0)\big) \leq 2\mathcal{R}(\widetilde{\mathbf{W}}, \widetilde{\mathbf{v}})$ holds for any $\widetilde{\mathbf{v}}$ and $\widetilde{\mathbf{W}}$, we finally derive that

$$\frac{1}{n}\sum_{i=1}^{n}\mathbb{P}_{(\mathbf{X},y)\sim\widetilde{\mathcal{D}}}\big(y \cdot f(\mathbf{Z}, \widetilde{\mathbf{W}}^{(i)}, \widetilde{\mathbf{v}}^{(i)}) \leq 0\big) \leq \frac{2}{n}\sum_{i=1}^{n}\mathcal{R}(\widetilde{\mathbf{v}}^{(i)}, \widetilde{\mathbf{W}}^{(i)}) \leq O\Big(\frac{\log^2 n(d+D)}{\gamma^2 n}\Big) + O\Big(\frac{\log(1/\delta)}{n}\Big),$$

which completes the proof. $\qquad\square$

# E TECHNICAL LEMMAS

## E.1 INDEPENDENCE AMONG GAUSSIAN RANDOM VECTORS

**Lemma E.1.** Let $z_1, z_2$ be two Gaussian random variables with zero mean, and $y = \text{sign}(z_1)$. Then $y$ is independent with $y \cdot z_1$ and $y \cdot z_2$. Moreover, $y \cdot z_2$ also follows the normal distribution, which has zero mean and the same variance with $z_2$.

*Proof of Lemma E.1.* W.L.O.G, we assume that $z_1, z_2$ are standard Gaussian random variables. We first prove that $y$ is independent with $y \cdot z_1$. It is clear that $y \cdot z_1 = |z_1|$. Then for any $x \geq 0$,

$$\mathbb{P}(y \cdot z_1 \leq x) = \mathbb{P}(|z_1| \leq x) = \mathbb{P}(|z_1| \leq x|y),$$

which indicate that $y \cdot z_1$ has the same distribution with $y \cdot z_1|y$. Therefore, $y$ is independent with $y \cdot z_1$. Next, we prove that $y$ is independent with $y \cdot z_2$ and $y \cdot z_2$ follows the standard normal distribution. We denote $\Phi(\cdot)$ the cumulative density function (c.d.f.) of the standard normal distribution. Then for any $x \in \mathbb{R}$,

$$\mathbb{P}(y \cdot z_2 \leq x) = \mathbb{P}(z_2 \leq x; y = 1) + \mathbb{P}(z_2 \geq -x; y = -1)$$
$$= \mathbb{P}(z_2 \leq x) \cdot \mathbb{P}(y = 1) + \mathbb{P}(z_2 \geq -x) \cdot \mathbb{P}(y = -1)$$
$$= \frac{1}{2}\Phi(x) + \frac{1}{2}\big(1 - \Phi(-x)\big) = \Phi(x),$$

where the second equality holds by the independence between $y$ and $z_2$, and the last equality holds by the symmetry of standard normal distribution that $\Phi(-x) = 1 - \Phi(x)$. This proves that $y \cdot z_2 \sim N(0, 1)$. Moreover, it is obvious that

$$\mathbb{P}(y \cdot z_2 \leq x|y = 1) = \mathbb{P}(z_2 \leq x|y = 1) = \Phi(x),$$

and

$$\mathbb{P}(y \cdot z_2 \leq x|y = -1) = \mathbb{P}(z_2 \geq -x|y = 1) = 1 - \Phi(-x) = \Phi(x),$$

which indicate that $y \cdot z_2$ has the same distribution with $y \cdot z_2|y$. Therefore, we complete the proof of Lemma E.1. $\qquad\square$

**Lemma E.2.** For $y$ and $\mathbf{x}_1, \mathbf{x}_2, \cdots, \mathbf{x}_D$ defined in Definition 2.1, it holds that $y$ is independent with $y \cdot \mathbf{x}_j$ for all $j \in [D]$. Moreover, $y \cdot \mathbf{x}_j \sim N(\mathbf{0}, \sigma_x^2 \mathbf{I}_d)$ for all $j \neq j^*$.

*Proof of Lemma E.2.* For simplicity of expression, we assume $\sigma_x = 1$ here, and the proof for $\sigma_x \neq 1$ is the same. We consider the following two cases: $j \neq j^*$ and $j = j^*$.

When $j \neq j^*$, $y$ is independent with $\mathbf{x}_j$. Then for any $k \in [d]$, we have $y \cdot \mathbf{x}_j[k] \sim N(0,1)$ and $y \cdot \mathbf{x}_j[k]$ is independent with $y$ by Lemma E.1. Therefore, $y \cdot \mathbf{x}_j$ is independent with $y$ since each coordinate of $y \cdot \mathbf{x}_j$ is independent with $y$. Moreover, we have $\mathbb{E}[y \cdot \mathbf{x}_j[k] \cdot y \cdot \mathbf{x}_j[k']] = \mathbb{E}[\mathbf{x}_j[k] \cdot \mathbf{x}_j[k']] = 0$ for any $k \neq k'$, which finally proves that $y \cdot \mathbf{x}_j \sim N(\mathbf{0}, \mathbf{I}_d)$.

When $j = j^*$, let $\mathbf{A}$ be an orthogonal matrix with $\mathbf{v}^*$ being its first column. We denote $\boldsymbol{\xi}_2, \cdots, \boldsymbol{\xi}_d$ the rest columns in $\mathbf{A}$, i.e., $\mathbf{A} = [\mathbf{v}^*, \boldsymbol{\xi}_2, \cdots, \boldsymbol{\xi}_d]$. Then we have

$$
y \cdot \mathbf{x}_{j^*} = y \cdot \mathbf{A}\mathbf{A}^\top \mathbf{x}_{j^*} = y \cdot [\mathbf{v}^*, \boldsymbol{\xi}_2, \cdots, \boldsymbol{\xi}_d] \cdot [\langle \mathbf{v}^*, \mathbf{x}_{j^*}\rangle, \langle \boldsymbol{\xi}_2, \mathbf{x}_{j^*}\rangle, \cdots, \langle \boldsymbol{\xi}_d, \mathbf{x}_{j^*}\rangle]^\top
$$

$$
= y \cdot \langle \mathbf{v}^*, \mathbf{x}_{j^*}\rangle \cdot \mathbf{v}^* + \sum_{i=2}^d y \cdot \langle \boldsymbol{\xi}_i, \mathbf{x}_{j^*}\rangle \cdot \boldsymbol{\xi}_i.
$$

By orthogonality among $\mathbf{v}^*, \boldsymbol{\xi}_2, \cdots, \boldsymbol{\xi}_d$, we obtain that $\langle \mathbf{v}^*, \mathbf{x}_{j^*}\rangle$ and $\langle \boldsymbol{\xi}_i, \mathbf{x}_{j^*}\rangle$ for $i \in \{2, \cdots, d\}$ are all independent random variables with standard normal distribution, which further indicates that $y$ is independent with $\langle \boldsymbol{\xi}_i, \mathbf{x}_{j^*}\rangle$ for all $i \in \{2, \cdots, d\}$. Then by Lemma E.1, we can obtain that $y$ is independent with $\sum_{i=2}^d y \cdot \langle \boldsymbol{\xi}_i, \mathbf{x}_{j^*}\rangle \cdot \mathbf{v}_i$. Besides, note that $y$ is the sign of the standard normal random variable $\langle \mathbf{v}^*, \mathbf{x}_{j^*}\rangle$. From Lemma E.1, we have that $y$ is also independent with $y \cdot \langle \mathbf{v}^*, \mathbf{x}_{j^*}\rangle$. Hence $y$ is independent with $y \cdot \mathbf{x}_{j^*}$. This completes the proof of Lemma E.2. $\qquad\square$

**Lemma E.3.** For $y$ and $\mathbf{x}_1, \mathbf{x}_2, \cdots, \mathbf{x}_D$ defined in Definition 2.1, it holds that $y \cdot \mathbf{x}_1, y \cdot \mathbf{x}_2, \cdots, y \cdot \mathbf{x}_D$ are mutually independent.

*Proof of Lemma E.3.* For any $j_1, j_2 \in [D]$ with $j_1 \neq j_2$, then at least one of $j_1$ and $j_2$ is not $j^*$. W.L.O.G., we assume $j_2 \neq j^*$. Then by Definition 2.1, $\mathbf{x}_{j_2}$ is independent both with $y$ and $\mathbf{x}_{j_1}$, hence it's independent with the product $y \cdot \mathbf{x}_{j_1}$. On the other hand, by applying Lemma E.2, $y \cdot \mathbf{x}_{j_1}$ is independent with $y$. Therefore, we finally prove that $y \cdot \mathbf{x}_{j_1}$ is independent with $y \cdot \mathbf{x}_{j_2}$. $\qquad\square$

### E.2 CALCULATION DETAILS OF EXPECTATIONS

**Lemma E.4.** Let $\mathbf{x} \sim N(0, \sigma_x^2 \mathbf{I}_d)$, for any fixed $\widetilde{\mathbf{v}} \in \mathbb{R}^d$, it holds that $\mathbb{E}\big[\mathbf{x}\, \mathrm{sign}\,(\langle \mathbf{x}, \widetilde{\mathbf{v}}\rangle)\big] = \sigma_x \sqrt{\frac{2}{\pi}} \cdot \frac{\widetilde{\mathbf{v}}}{\|\widetilde{\mathbf{v}}\|_2}$.

*Proof of Lemma E.4.* Let $\widetilde{\mathbf{A}} = [\widetilde{\mathbf{v}}/\|\widetilde{\mathbf{v}}\|_2, \boldsymbol{\xi}_2', \cdots, \boldsymbol{\xi}_d']$ be an orthogonal matrix, which is defined similarly with Lemma E.1. Then by a similar method, we obtain that,

$$
\mathbf{x}\, \mathrm{sign}\,(\langle \mathbf{x}, \widetilde{\mathbf{v}}\rangle) = \mathrm{sign}\,(\langle \mathbf{x}, \widetilde{\mathbf{v}}\rangle) \cdot \Big\langle \frac{\widetilde{\mathbf{v}}}{\|\widetilde{\mathbf{v}}\|_2}, \mathbf{x}\Big\rangle \cdot \frac{\widetilde{\mathbf{v}}}{\|\widetilde{\mathbf{v}}\|_2} + \sum_{i=2}^d \mathrm{sign}\,(\langle \mathbf{x}, \widetilde{\mathbf{v}}\rangle) \cdot \langle \boldsymbol{\xi}_i', \mathbf{x}\rangle \cdot \boldsymbol{\xi}_i'.
$$

Note that $\big\langle \frac{\widetilde{\mathbf{v}}}{\|\widetilde{\mathbf{v}}\|_2}, \mathbf{x}\big\rangle$ and $\langle \boldsymbol{\xi}_i, \mathbf{x}\rangle$ for $i \in \{2, \cdots, d\}$ are i.i.d normal random variables with mean 0 and variance $\sigma_x^2$, therefore we have

$$
\mathbb{E}\Big[\mathbf{x}\, \mathrm{sign}\,(\langle \mathbf{x}, \widetilde{\mathbf{v}}\rangle)\Big] = \mathbb{E}\Big[\mathrm{sign}\,(\langle \mathbf{x}, \widetilde{\mathbf{v}}\rangle) \cdot \Big\langle \frac{\widetilde{\mathbf{v}}}{\|\widetilde{\mathbf{v}}\|_2}, \mathbf{x}\Big\rangle\Big] \cdot \frac{\widetilde{\mathbf{v}}}{\|\widetilde{\mathbf{v}}\|_2} = \sigma_x \sqrt{\frac{2}{\pi}} \cdot \frac{\widetilde{\mathbf{v}}}{\|\widetilde{\mathbf{v}}\|_2}.
$$

This completes the proof. $\qquad\square$

**Lemma E.5.** Let $z_1 \sim N(0, \sigma_1^2)$ and a fixed scalar $a$, then it holds that

1. If $a < 0$, then

$$
\max\left\{\sqrt{\frac{2}{\pi e}} e^{\sigma_1 a}, \sqrt{\frac{2}{\pi}}\Big(\frac{1}{-\sigma_1 a} - \frac{1}{-\sigma_1^3 a^3}\Big)\right\} \leq \mathbb{E}\big[\exp(a|z_1|)\big] \leq \min\left\{2, \sqrt{\frac{2}{\pi}}\frac{1}{-\sigma_1 a}\right\}.
$$

2. If $a \geq 0$, then

$$
e^{\frac{\sigma_1^2 a^2}{2}} \leq \mathbb{E}\big[\exp(a|z_1|)\big] \leq 2e^{\frac{\sigma_1^2 a^2}{2}}.
$$

*Proof of Lemma E.5.* The probability density function for $|z_1|$ is $f_{|z_1|}(x) = \frac{1}{\sigma_1}\sqrt{\frac{2}{\pi}}e^{-x^2/(2\sigma_1^2)}\mathbf{1}_{\{x\geq 0\}}$. By this density function, we can calculate $\mathbb{E}\big[\exp(a|z_1|)\big]$ as,

$$\mathbb{E}\big[\exp(a|z_1|)\big] = \frac{1}{\sigma_1}\sqrt{\frac{2}{\pi}}\int_0^\infty e^{-\frac{x^2}{2\sigma_1^2}+ax}\mathrm{d}x = \frac{1}{\sigma_1}\sqrt{\frac{2}{\pi}}e^{\frac{\sigma_1^2 a^2}{2}}\int_0^\infty e^{-\frac{1}{2}\left(\frac{x}{\sigma_1}-\sigma_1 a\right)^2}\mathrm{d}x = 2e^{\frac{\sigma_1^2 a^2}{2}}\mathbb{P}(z\geq -\sigma_1 a),$$

where $z$ is a standard Gaussian random variable. If $a \geq 0$, it holds that $\frac{1}{2} \leq \mathbb{P}(z \geq -\sigma_1 a) \leq 1$. Then we can obtain that

$$e^{\frac{\sigma_1^2 a^2}{2}} \leq \mathbb{E}\big[\exp(a|z_1|)\big] \leq 2e^{\frac{\sigma_1^2 a^2}{2}}.$$

When $a < 0$, by applying the tail-bound of standard Gaussian random variables and Mills ratio simultaneously, we have

$$\frac{1}{\sqrt{2\pi}}e^{-\frac{\sigma_1^2 a^2}{2}}\left(\frac{1}{-\sigma_1 a}-\frac{1}{-\sigma_1^3 a^3}\right) \leq \mathbb{P}(z\geq -\sigma_1 a) \leq e^{-\frac{\sigma_1^2 a^2}{2}}\min\left\{1,\frac{1}{\sqrt{2\pi}}\frac{1}{-\sigma_1 a}\right\}.$$

Therefore, we derive that,

$$\sqrt{\frac{2}{\pi}}\left(\frac{1}{-\sigma_1 a}-\frac{1}{-\sigma_1^3 a^3}\right) \leq \mathbb{E}\big[\exp(a|z_1|)\big] \leq \min\left\{2,\sqrt{\frac{2}{\pi}}\frac{1}{-\sigma_1 a}\right\}.$$

Besides, we also have

$$\mathbb{P}(z\geq -\sigma_1 a) = \frac{1}{\sqrt{2\pi}}\int_{-\sigma_1 a}^\infty e^{-\frac{x^2}{2}}\mathrm{d}x \geq \frac{1}{\sqrt{2\pi}}e^{-\frac{(-\sigma_1 a+1)^2}{2}},$$

which further implies that $\mathbb{E}\big[\exp(a|z_1|)\big] \geq \sqrt{\frac{2}{\pi e}}e^{\sigma_1 a}$. Combining all preceding results, we finish the proof. $\qquad\square$

**Lemma E.6.** Let $z_1 \sim N(0,\sigma_1^2)$ and two fixed positive scalars $a, b$, then it holds that

$$\mathbb{E}\big[\exp(-a|z_1|)\mathbf{1}_{|z_1|\leq b}\big] \geq \sqrt{\frac{2}{\pi}}\left(\frac{1}{\sigma_1 a}-\frac{1}{\sigma_1^3 a^3}-\frac{1}{\sigma_1 a+b/\sigma_1}\exp\left(-ab-\frac{b^2}{2\sigma_1^2}\right)\right).$$

*Proof of Lemma E.6.* The probability density function for $|z_1|$ is $f_{|z_1|}(x) = \frac{1}{\sigma_1}\sqrt{\frac{2}{\pi}}e^{-x^2/(2\sigma_1^2)}\mathbf{1}_{\{x\geq 0\}}$. By this density function, we can calculate $\mathbb{E}\big[\exp(-a|z_1|)\mathbf{1}_{|z_1|\leq b}\big]$ as,

$$\mathbb{E}\big[\exp(-a|z_1|)\mathbf{1}_{|z_1|\leq b}\big] = \frac{1}{\sigma_1}\sqrt{\frac{2}{\pi}}\int_0^b e^{-\frac{x^2}{2\sigma_1^2}-ax}\mathrm{d}x = \frac{1}{\sigma_1}\sqrt{\frac{2}{\pi}}e^{\frac{\sigma_1^2 a^2}{2}}\int_0^b e^{-\frac{1}{2}\left(\frac{x}{\sigma_1}+\sigma_1 a\right)^2}\mathrm{d}x$$

$$= 2e^{\frac{\sigma_1^2 a^2}{2}}\Big(\mathbb{P}(z\geq \sigma_1 a)-\mathbb{P}(z\geq \sigma_1 a+b/\sigma_1)\Big),$$

where $z$ is a standard Gaussian random variable. By applying the Mills ratio, we have

$$\mathbb{P}(z\geq -\sigma_1 a) \geq \frac{1}{\sqrt{2\pi}}e^{-\frac{\sigma_1^2 a^2}{2}}\left(\frac{1}{\sigma_1 a}-\frac{1}{\sigma_1^3 a^3}\right),$$

and

$$\mathbb{P}(z\geq \sigma_1 a+b/\sigma_1) \leq \frac{1}{\sqrt{2\pi}(\sigma_1 a+b/\sigma_1)}e^{-\frac{(\sigma_1 a+b/\sigma_1)^2}{2}}$$

Therefore, we finally derive that,

$$\mathbb{E}\big[\exp(-a|z_1|)\mathbf{1}_{|z_1|\leq b}\big] \geq \sqrt{\frac{2}{\pi}}\left(\frac{1}{\sigma_1 a}-\frac{1}{\sigma_1^3 a^3}-\frac{1}{\sigma_1 a+b/\sigma_1}\exp\left(-ab-\frac{b^2}{2\sigma_1^2}\right)\right),$$

which completes the proof. $\qquad\square$

**Lemma E.7.** Let $z_1 \sim N(0,\sigma_1^2)$ and a fixed scalar $a$, then it holds that

1. If $a \geq 0$, then

$$lb_1(\sigma_1, a) \leq \mathbb{E}\big[|z_1| \exp(-a|z_1|)\big] \leq \sqrt{\frac{2}{\pi}} \frac{1}{\sigma_1 a^2}, \tag{E.1}$$

where $lb_1(\sigma_1, a) = \max\left\{ \sigma_1 \sqrt{\frac{2}{\pi}} - 2\sigma_1^2 a, \frac{\sigma_1}{2\sqrt{2e\pi}} e^{-\sigma_1 a}, \sqrt{\frac{2}{\pi}}\left(\frac{1}{\sigma_1 a^2} - \frac{3}{\sigma_1^3 a^4}\right) \right\}$.

2. If $a < 0$, then

$$\sigma_1 \sqrt{\frac{2}{\pi}} - \sigma_1^2 a e^{\frac{\sigma_1^2 a^2}{2}} \leq \mathbb{E}\big[|z_1| \exp(-a|z_1|)\big] \leq \sigma_1 \sqrt{\frac{2}{\pi}} - 2\sigma_1^2 a e^{\frac{\sigma_1^2 a^2}{2}}. \tag{E.2}$$

*Proof of Lemma E.7.* Similar to Lemma E.5, we can calculate $\mathbb{E}\big[|z_1| \exp(-a|z_1|)\big]$ as

$$
\begin{aligned}
\mathbb{E}\big[|z_1| \exp(-a|z_1|)\big] &= \frac{1}{\sigma_1} \sqrt{\frac{2}{\pi}} \int_0^\infty x e^{-\frac{x^2}{2\sigma_1^2} - ax} \mathrm{d}x \\
&= \frac{1}{\sigma_1} \sqrt{\frac{2}{\pi}} e^{\frac{\sigma_1^2 a^2}{2}} \int_0^\infty x e^{-\frac{1}{2}\left(\frac{x}{\sigma_1} + \sigma_1 a\right)^2} \mathrm{d}x \\
&= \sigma_1 \sqrt{\frac{2}{\pi}} e^{\frac{\sigma_1^2 a^2}{2}} \int_0^\infty x e^{-\frac{1}{2}(x + \sigma_1 a)^2} \mathrm{d}x \\
&= \sigma_1 \sqrt{\frac{2}{\pi}} e^{\frac{\sigma_1^2 a^2}{2}} \left[ -\sigma_1 a \int_0^\infty e^{-\frac{1}{2}(x+\sigma_1 a)^2} \mathrm{d}x + \int_0^\infty (x + \sigma_1 a) e^{-\frac{1}{2}(x+\sigma_1 a)^2} \mathrm{d}x \right] \\
&= \sigma_1 \left[ \sqrt{\frac{2}{\pi}} - 2\sigma_1 a e^{\frac{\sigma_1^2 a^2}{2}} \mathbb{P}(z \geq \sigma_1 a) \right],
\end{aligned}
\tag{E.3}
$$

where $z$ is a standard Gaussian random variable. When $a \geq 0$, by applying tail-bound of standard Gaussian random variables and Mills ratio simultaneously, we have

$$\frac{1}{\sqrt{2\pi}} e^{-\frac{\sigma_1^2 a^2}{2}} \left(\frac{1}{\sigma_1 a} - \frac{1}{\sigma_1^3 a^3}\right) \leq \mathbb{P}(z \geq \sigma_1 a) \leq e^{-\frac{\sigma_1^2 a^2}{2}} \min\left\{ 1, \frac{1}{\sqrt{2\pi}} \left(\frac{1}{\sigma_1 a} - \frac{1}{\sigma_1^3 a^3} + \frac{3}{\sigma_1^5 a^5}\right) \right\}. \tag{E.4}$$

Applying the result of (E.4) into (E.3), we finally get

$$\max\left\{ \sigma_1 \sqrt{\frac{2}{\pi}} - 2\sigma_1^2 a, \sqrt{\frac{2}{\pi}}\left(\frac{1}{\sigma_1 a^2} - \frac{3}{\sigma_1^3 a^4}\right) \right\} \leq \mathbb{E}\big[|z_1| \exp(-a|z_1|)\big] \leq \sqrt{\frac{2}{\pi}} \frac{1}{\sigma_1 a^2}. \tag{E.5}$$

Besides, we also have

$$
\begin{aligned}
\mathbb{E}\big[|z_1| \exp(-a|z_1|)\big] &= \sigma_1 \sqrt{\frac{2}{\pi}} e^{\frac{\sigma_1^2 a^2}{2}} \int_0^\infty x e^{-\frac{1}{2}(x+\sigma_1 a)^2} \mathrm{d}x \\
&\geq \sigma_1 \sqrt{\frac{2}{\pi}} e^{\frac{\sigma_1^2 a^2}{2}} \int_{1/2}^1 x e^{-\frac{1}{2}(x+\sigma_1 a)^2} \mathrm{d}x \geq \frac{\sigma_1}{2\sqrt{2e\pi}} e^{-\sigma_1 a}.
\end{aligned}
\tag{E.6}
$$

Combining the results of (E.5) and (E.6), we finishes the proof of (E.1). When $a < 0$, it's obvious that $1/2 \leq \mathbb{P}(z \geq \sigma_1 a) \leq 1$. Replacing this results in (E.3), we finish the proof for (E.2)  □

**Lemma E.8.** Let $z_1 \sim N(0, \sigma_1^2)$ and a non-negative fixed scalar $a$, then it holds that

$$\mathbb{E}\big[z_1^2 \exp(a|z_1|)\big] \leq 2\sigma_1^2(a^2 \sigma_1^2 + 1) e^{\frac{\sigma_1^2 a^2}{2}}$$

*Proof of Lemma E.8.* Based on the density function of $|z_1|$, we can calculate that

$$\mathbb{E}\big[z_1^2 \exp(a|z_1|)\big] = \frac{1}{\sigma_1} \sqrt{\frac{2}{\pi}} \int_0^\infty x^2 e^{-\frac{x^2}{2\sigma_1^2} + ax} \mathrm{d}x$$

$$
= \frac{1}{\sigma_1}\sqrt{\frac{2}{\pi}}e^{\frac{\sigma_1^2 a^2}{2}}\int_0^\infty x^2 e^{-\frac{1}{2}\left(\frac{x}{\sigma_1}-\sigma_1 a\right)^2}\mathrm{d}x
$$

$$
= \sigma_1^2\sqrt{\frac{2}{\pi}}e^{\frac{\sigma_1^2 a^2}{2}}\int_{-a\sigma_1}^\infty (x+a\sigma_1)^2 e^{-\frac{1}{2}x^2}\mathrm{d}x
$$

$$
\leq 2\sigma_1^2 e^{\frac{\sigma_1^2 a^2}{2}}\mathbb{E}\big[(z+a\sigma_1)^2\big] = 2\sigma_1^2(a^2\sigma_1^2+1)e^{\frac{\sigma_1^2 a^2}{2}},
$$

where $z$ is a standard Gaussian random variable. This finishes the proof. $\qquad\square$

**Lemma E.9.** For any Gaussian random variable $z$ with mean 0, it holds that $\mathbb{E}[-\ell'(z)] \geq 1/4$.

*Proof of Lemma E.9.* It can be derived that

$$
\mathbb{E}[-\ell'(z)] = \mathbb{E}\big[-\ell'(z)\mathbf{1}_{\{z\leq 0\}}\big] + \mathbb{E}\big[-\ell'(z)\mathbf{1}_{\{z>0\}}\big] \geq \mathbb{E}\big[-\ell'(z)\mathbf{1}_{\{z\leq 0\}}\big] \geq \frac{1}{2}\mathbb{E}\big[\mathbf{1}_{\{z\leq 0\}}\big] = \frac{1}{4},
$$

which finishes the proof. $\qquad\square$

**Lemma E.10.** Let $z_1 \sim N(0,\sigma_1^2)$ and $z_2 \sim N(0,\sigma_2^2)$ be two independent Gaussian random variables, and $a, b$ be two scalars. Then it holds that,

1. If $a \geq 0$, then

$$
\frac{1}{4}lb_1(\sigma_1,a) \leq -\mathbb{E}\big[\ell'(a|z_1|+bz_2)|z_1|\big] \leq ub_1(\sigma_1,\sigma_2,a,b), \tag{E.7}
$$

where $lb_1(\sigma_1,a) = \max\left\{\sigma_1\sqrt{\frac{2}{\pi}} - 2\sigma_1^2 a, \frac{\sigma_1}{2\sqrt{2e\pi}}e^{-\sigma_1 a}, \sqrt{\frac{2}{\pi}}\left(\frac{1}{\sigma_1 a^2} - \frac{3}{\sigma_1^3 a^4}\right)\right\}$ as defined in Lemma E.7, and $ub_1(\sigma_1,\sigma_2,a,b) = \min\left\{\sqrt{\frac{2}{\pi}}\frac{1}{\sigma_1 a^2}e^{\frac{\sigma_2^2 b^2}{2}}, \sigma_1\sqrt{\frac{2}{\pi}}\right\}$.

2. If $a < 0$, then

$$
\frac{\sigma_1}{4}\sqrt{\frac{2}{\pi}} \leq -\mathbb{E}\big[\ell'(a|z_1|+bz_2)|z_1|\big] \leq \sigma_1\sqrt{\frac{2}{\pi}}. \tag{E.8}
$$

*Proof of Lemma E.10.* By the law of total expectation, we have $-\mathbb{E}\big[\ell'(a|z_1|+bz_2)|z_1|\big] = \mathbb{E}\Big[\mathbb{E}\big[-\ell'(a|z_1|+bz_2)|z_1|\big|z_2\big]\Big]$. Since $z_1$, $z_2$ are independent, we still have $z_1|z_2 \sim N(0,\sigma_1)$. We first prove the case for $a \geq 0$. The lower-bound can be calculated as

$$
\mathbb{E}\Big[\mathbb{E}\big[-\ell'(a|z_1|+bz_2)|z_1|\big|z_2\big]\Big] = \mathbb{E}\left[\mathbb{E}\left[\frac{|z_1|}{1+\exp(a|z_1|+bz_2)}\bigg|z_2\right]\right]
$$

$$
\geq \mathbb{E}\left[\mathbb{E}\left[\frac{|z_1|}{\exp(a|z_1|)+\exp(a|z_1|+bz_2)}\bigg|z_2\right]\right]
$$

$$
\geq \mathbb{E}[-\ell'(bz_2)]\mathbb{E}\big[|z_1|\exp(-a|z_1|)\big] \geq \frac{1}{4}lb_1(\sigma_1,a), \tag{E.9}
$$

where the last inequality holds by applying Lemma E.7 and Lemma E.9, and $lb_1(\sigma_1,a)$ is defined in Lemma E.7. On the other hand, we have

$$
-\mathbb{E}\big[\ell'(a|z_1|+bz_2)|z_1|\big] = \mathbb{E}\left[\frac{|z_1|}{1+\exp(a|z_1|+bz_2)}\right] \leq \mathbb{E}\left[\frac{|z_1|}{\exp(a|z_1|+bz_2)}\right]
$$

$$
= \mathbb{E}\big[|z_1|\exp(-a|z_1|)\big]\mathbb{E}\big[\exp(-bz_2)\big] \leq \sqrt{\frac{2}{\pi}}\frac{1}{\sigma_1 a^2}e^{\frac{\sigma_2^2 b^2}{2}}, \tag{E.10}
$$

where the last inequality holds by applying Lemma E.7 and moment-generating function of Gaussian random variables. Besides, we also have

$$
-\mathbb{E}\big[\ell'(a|z_1|+bz_2)|z_1|\big] \leq \mathbb{E}[|z_1|] = \sigma_1\sqrt{\frac{2}{\pi}}. \tag{E.11}
$$

Combining (E.9), (E.10) and (E.11), we finish the proof for (E.7). Next, we prove the case for $a < 0$, and (E.11) still holds for $a < 0$. Besides by Lemma E.9, we can obatin that

$$-\mathbb{E}\big[\ell'(a|z_1| + bz_2)|z_1|\big] \geq -\mathbb{E}\big[\ell'(bz_2)\big]\mathbb{E}\big[|z_1|\big] \geq \frac{\sigma_1}{4}\sqrt{\frac{2}{\pi}}. \qquad (E.12)$$

Combining (E.11) and (E.12), we finish the proof for (E.8). $\qquad\square$

**Lemma E.11.** Let $z_1 \sim N(0, \sigma_1^2)$, $z_2 \sim N(0, \sigma_2^2)$ and $z_3 \sim N(0, \sigma_3^2)$ be three independent Gaussian random variables, and $a, b, c$ be three non-negative scalars. Then it holds that

$$lb_2(\sigma_1, \sigma_2, \sigma_3, a, b, c) \leq -\mathbb{E}[\ell'(a|z_1| + bz_2 + cz_3)z_3] \leq ub_2(\sigma_1, \sigma_2, \sigma_3, a, b, c), \qquad (E.13)$$

where

$$lb_2(\sigma_1, \sigma_2, \sigma_3, a, b, c) = \max\left\{ -\frac{c\sigma_3^2}{4}, -\frac{\sigma_3}{\sqrt{2\pi}}, -2\sqrt{\frac{2}{\pi}}\frac{\sigma_3^2 c}{\sigma_2 b}e^{\frac{\sigma_1^2 a^2 + \sigma_3^2 c^2}{2}}, -\sqrt{\frac{2}{\pi}}\frac{\sigma_3^2 c}{\sigma_1 a}e^{\frac{\sigma_2^2 b^2}{2} + \frac{\sigma_3^2 c^2}{2}} \right\},$$

and

$$ub_2(\sigma_1, \sigma_2, \sigma_3, a, b, c) = -\frac{\sigma_3}{3\pi\sigma_2 b}\max\left\{ \sqrt{\frac{2}{\pi e}}e^{-\sigma_1 a}, \sqrt{\frac{2}{\pi}}\Big(\frac{1}{\sigma_1 a} - \frac{1}{\sigma_1^3 a^3}\Big) \right\}e^{-\frac{1}{2\sigma_2^2 b^2} - \frac{1}{2\sigma_3^2 c^2}}.$$

*Proof of Lemma E.11.* Similarly, by the law of total expectation, we have $-\mathbb{E}\big[\ell'(a|z_1| + bz_2 + cz_3)z_3\big] = \mathbb{E}\Big[\mathbb{E}\big[-\ell'(a|z_1| + bz_2 + cz_3)z_3\big|z_1, z_2\big]\Big]$. Notice that $-\ell'$ is a decreasing function and $z_3$ has a zero-centered symmetric density function. Hence, we obtain that $\mathbb{E}\big[-\ell'(a|z_1| + bz_2 + cz_3)z_3\big|z_1, z_2\big] \leq 0$. Specifically, we have

$$\mathbb{E}\big[-\ell'(a|z_1| + bz_2 + cz_3)z_3\big|z_1, z_2\big]$$
$$=\mathbb{E}\big[-\ell'(a|z_1| + bz_2 + cz_3)z_3\mathbf{1}_{\{z_3 \geq 0\}}\big|z_1, z_2\big] + \mathbb{E}\big[-\ell'(a|z_1| + bz_2 + cz_3)z_3\mathbf{1}_{\{z_3 < 0\}}\big|z_1, z_2\big]$$
$$= -\mathbb{E}\Big[\big(-\ell'(a|z_1| + bz_2 - cz_3) + \ell'(a|z_1| + bz_2 + cz_3)\big)z_3\mathbf{1}_{\{z_3 \geq 0\}}\big|z_1, z_2\Big]. \qquad (E.14)$$

Since $-\ell'$ is Lipschitz continuous with $\frac{1}{4}$ and the value of $-\ell'$ is always in $(0, 1)$, we have

$$-\mathbb{E}\Big[\big(-\ell'(a|z_1| + bz_2 - cz_3) + \ell'(a|z_1| + bz_2 + cz_3)\big)z_3\mathbf{1}_{\{z_3 \geq 0\}}\big|z_1, z_2\Big]$$
$$\geq -\min\Big\{\mathbb{E}\big[cz_3^2\mathbf{1}_{\{z_3 \geq 0\}}/2\big|z_1, z_2\big], \mathbb{E}\big[z_3\mathbf{1}_{\{z_3 \geq 0\}}\big|z_1, z_2\big]\Big\} = \max\Big\{ -\frac{c\sigma_3^2}{4}, -\frac{\sigma_3}{\sqrt{2\pi}}\Big\}. \qquad (E.15)$$

On the other hand, we also have

$$-\ell'(a|z_1| + bz_2 - cz_3) + \ell'(a|z_1| + bz_2 + cz_3)$$
$$=\frac{1}{1 + e^{a|z_1| + bz_2 - cz_3}} - \frac{1}{1 + e^{a|z_1| + bz_2 + cz_3}} = \frac{e^{a|z_1| + bz_2}(e^{cz_3} - e^{-cz_3})}{\big(1 + e^{a|z_1| + bz_2 - cz_3}\big)\big(1 + e^{a|z_1| + bz_2 + cz_3}\big)}$$
$$\geq \frac{e^{a|z_1| + bz_2}(e^{cz_3} - e^{-cz_3})}{\big(e^{a|z_1|} + e^{a|z_1| + bz_2 - cz_3}\big)\big(e^{a|z_1|} + e^{a|z_1| + bz_2 + cz_3}\big)} = e^{-a|z_1|}\frac{e^{cz_3} - e^{-cz_3}}{e^{cz_3} + e^{-cz_3} + e^{bz_2} + e^{-bz_2}}$$
$$\geq e^{-a|z_1|}\frac{e^{cz_3} - e^{-cz_3}}{e^{cz_3} + e^{-cz_3} + e^{bz_2} + e^{-bz_2}}\mathbf{1}_{\{z_3 \geq 1/c\}}\mathbf{1}_{\{|z_2| \leq 1/b\}}$$
$$\geq e^{-a|z_1|}\frac{e - e^{-1}}{2(e + e^{-1})}\mathbf{1}_{\{z_3 \geq 1/c\}}\mathbf{1}_{\{|z_2| \leq 1/b\}} \geq \frac{1}{3}e^{-a|z_1|}\mathbf{1}_{\{z_3 \geq 1/c\}}\mathbf{1}_{\{|z_2| \leq 1/b\}}. \qquad (E.16)$$

By applying result of (E.16) in (E.14) and Lemma E.5, we obtain

$$\mathbb{E}\big[-\ell'(a|z_1| + bz_2 + cz_3)z_3\big] \leq -\frac{1}{3}\mathbb{E}\big[e^{-a|z_1|}\big]\mathbb{E}\big[z_3\mathbf{1}_{\{z_3 \geq 1/c\}}\big]\mathbb{P}(|z_2| \leq 1/b)$$
$$\leq -\frac{\sigma_3}{3\pi\sigma_2 b}\max\left\{ \sqrt{\frac{2}{\pi e}}e^{-\sigma_1 a}, \sqrt{\frac{2}{\pi}}\Big(\frac{1}{\sigma_1 a} - \frac{1}{\sigma_1^3 a^3}\Big) \right\}e^{-\frac{1}{2\sigma_2^2 b^2} - \frac{1}{2\sigma_3^2 c^2}}, \qquad (E.17)$$

where the last inequality holds because $\mathbb{E}\big[z_3\mathbf{1}_{\{z_3\geq 1/c\}}\big] = \frac{1}{\sigma_3\sqrt{2\pi}}\int_{1/c}^{\infty} xe^{-\frac{x^2}{2\sigma_3^2}}\,\mathrm{d}x = \frac{\sigma_3}{\sqrt{2\pi}}e^{-\frac{1}{2c^2\sigma_3^2}}$

and $\mathbb{P}(|z_2|\leq 1/b) = \frac{1}{\sigma_2\sqrt{2\pi}}\int_{-1/b}^{1/b} e^{-\frac{x^2}{2\sigma_2^2}}\,\mathrm{d}x \geq \frac{2}{b\sigma_2\sqrt{2\pi}}e^{-\frac{1}{2\sigma_2^2 b^2}}$ . Similarly, we also have

$$-\ell'(a|z_1|+bz_2-cz_3) + \ell'(a|z_1|+bz_2+cz_3)$$
$$=\frac{1}{1+e^{a|z_1|+bz_2-cz_3}} - \frac{1}{1+e^{a|z_1|+bz_2-cz_3}} = \frac{e^{a|z_1|+bz_2}(e^{cz_3}-e^{-cz_3})}{\big(1+e^{a|z_1|+bz_2-cz_3}\big)\big(1+e^{a|z_1|+bz_2+cz_3}\big)}$$
$$\leq\frac{e^{a|z_1|+bz_2}(e^{cz_3}-e^{-cz_3})}{\big(1+e^{bz_2-cz_3}\big)\big(1+e^{bz_2+cz_3}\big)} = e^{a|z_1|}\frac{e^{cz_3}-e^{-cz_3}}{e^{cz_3}+e^{-cz_3}+e^{bz_2}+e^{-bz_2}}$$
$$\leq e^{a|z_1|}\frac{e^{cz_3}-e^{-cz_3}}{e^{bz_2}+e^{-bz_2}} \leq e^{a|z_1|}e^{-b|z_2|}\big(e^{cz_3}-e^{-cz_3}\big). \tag{E.18}$$

By applying result of (E.18), Lemma E.5 and Lemma E.7 in (E.14), we have

$$\mathbb{E}\big[-\ell'(a|z_1|+bz_2+cz_3)z_3\big] \geq -\mathbb{E}\big[e^{a|z_1|}\big]\mathbb{E}\big[e^{-b|z_2|}\big]\mathbb{E}\big[z_3(e^{cz_3}-e^{-cz_3})\mathbf{1}_{\{z_3\geq 0\}}\big]$$
$$\geq -2\sqrt{\frac{2}{\pi}}\frac{\sigma_3^2 c}{\sigma_2 b}e^{\frac{\sigma_1^2 a^2+\sigma_3^2 c^2}{2}}. \tag{E.19}$$

Moreover, we can also obtain that

$$-\ell'(a|z_1|+bz_2-cz_3) + \ell'(a|z_1|+bz_2+cz_3)$$
$$=\frac{1}{1+e^{a|z_1|+bz_2-cz_3}} - \frac{1}{1+e^{a|z_1|+bz_2-cz_3}} = \frac{e^{a|z_1|+bz_2}(e^{cz_3}-e^{-cz_3})}{\big(1+e^{a|z_1|+bz_2-cz_3}\big)\big(1+e^{a|z_1|+bz_2+cz_3}\big)}$$
$$\leq\frac{e^{a|z_1|+bz_2}(e^{cz_3}-e^{-cz_3})}{e^{a|z_1|+bz_2-cz_3}e^{a|z_1|+bz_2+cz_3}} = e^{-a|z_1|}e^{-bz_2}\big(e^{cz_3}-e^{-cz_3}\big). \tag{E.20}$$

By applying result of (E.20), Lemma E.5 and Lemma E.7 in (E.14), we have

$$\mathbb{E}\big[-\ell'(a|z_1|+bz_2+cz_3)z_3\big] \geq -\mathbb{E}\big[e^{-a|z_1|}\big]\mathbb{E}\big[e^{-bz_2}\big]\mathbb{E}\big[z_3(e^{cz_3}-e^{-cz_3})\mathbf{1}_{\{z_3\geq 0\}}\big]$$
$$\geq -\sqrt{\frac{2}{\pi}}\frac{\sigma_3^2 c}{\sigma_1 a}e^{\frac{\sigma_2^2 b^2}{2}+\frac{\sigma_3^2 c^2}{2}}. \tag{E.21}$$

Combining (E.15), (E.17), (E.19), and (E.21), we finish the proof for (E.13). $\qquad\square$

The proofs for the following Lemmas are very similar to that of Lemma E.11, while we still include them here for completeness.

**Lemma E.12.** Let $z_1 \sim N(0,\sigma_1^2)$, $z_2 \sim N(0,\sigma_2^2)$, $z_3 \sim N(0,\sigma_3^2)$, and $z_4 \sim N(0,\sigma_4^2)$ be four independent Gaussian random variables, and $a_1, a_2, a_3, a_4$ be four non-negative scalars. Then it holds that

$$lb_3(\sigma_1,\sigma_2,\sigma_3,\sigma_4,a_1,a_2,a_3,a_4) \leq -\mathbb{E}[\ell'(a_1|z_1|+a_2z_2+a_3z_3+a_4z_4)z_3^2 z_4] \leq ub_3(\sigma_1,\sigma_2,\sigma_3,\sigma_4,a_1,a_2,a_3,a_4), \tag{E.22}$$

where

$$lb_3(\sigma_1,\sigma_2,\sigma_3,\sigma_4,a_1,a_2,a_3,a_4) = \max\left\{-\frac{a_4\sigma_3^2\sigma_4^2}{4}, -\frac{\sigma_3^2\sigma_4}{\sqrt{2\pi}}, -4\sqrt{\frac{2}{\pi}}\frac{\sigma_3^2\sigma_4^2 a_4(a_3^2\sigma_3^2+1)}{\sigma_2 a_2}e^{\frac{\sigma_1^2 a_1^2+\sigma_3^2 a_3^2+\sigma_4^2 a_4^2}{2}}\right\},$$

and

$$ub_3(\sigma_1,\sigma_2,\sigma_3,\sigma_4,a_1,a_2,a_3,a_4) = -\frac{\sigma_4}{192\pi^3 a_2 a_3^3 \sigma_2\sigma_3}\max\left\{e^{-\sigma_1 a_1-1/2}, \frac{1}{\sigma_1 a_1}-\frac{1}{\sigma_1^3 a_1^3}\right\}e^{-\frac{1}{8\sigma_2^2 a_2^2}-\frac{1}{8\sigma_3^2 a_3^2}-\frac{1}{2\sigma_4^2 a_4^2}}.$$

*Proof of Lemma E.12.* By the law of total expectation, we have $-\mathbb{E}[\ell'(a|z_1|+bz_2+cz_3+dz_4)z_3^2 z_4] = \mathbb{E}\Big[z_3^2\mathbb{E}[-\ell'(a|z_1|+bz_2+cz_3+dz_4)z_4|z_1,z_2,z_3]\Big]$. Notice that $-\ell'$ is a decreasing function and $z_4$ has a zero-centered symmetric density function. Hence, we obtain that $\mathbb{E}[-\ell'(a|z_1|+bz_2+cz_3+dz_4)z_4|z_1,z_2,z_3] \leq 0$. Specifically, we have

$$\mathbb{E}[-\ell'(a_1|z_1|+a_2z_2+a_3z_3+a_4z_4)z_4|z_1,z_2,z_3]$$

$$
\begin{aligned}
=&\mathbb{E}\big[ -\ell'(a_1|z_1| + a_2z_2 + a_3z_3 + a_4z_4)z_4\mathbf{1}_{\{z_4\geq 0\}}\big|z_1, z_2, z_3\big] \\
&+ \mathbb{E}\big[ -\ell'(a_1|z_1| + a_2z_2 + a_3z_3 + a_4z_4)z_4\mathbf{1}_{\{z_4< 0\}}\big|z_1, z_2, z_3\big] \\
=&-\mathbb{E}\Big[\big( -\ell'(a_1|z_1| + a_2z_2 + a_3z_3 - a_4z_4) + \ell'(a_1|z_1| + a_2z_2 + a_3z_3 + a_4z_4)\big)z_4\mathbf{1}_{\{z_4\geq 0\}}\big|z_1, z_2, z_3\Big].
\end{aligned}
$$
(E.23)

Since $-\ell'$ is Lipschitz continuous with $\frac{1}{4}$ and the value of $-\ell'$ is always in $(0, 1)$, we have

$$
-\mathbb{E}\Big[\big( -\ell'(a_1|z_1| + a_2z_2 + a_3z_3 - a_4z_4) + \ell'(a_1|z_1| + a_2z_2 + a_3z_3 + a_4z_4)\big)z_4\mathbf{1}_{\{z_4\geq 0\}}\big|z_1, z_2, z_3\Big]
$$

$$
\geq -\min\Big\{\mathbb{E}\big[a_4z_4^2\mathbf{1}_{\{z_4\geq 0\}}/2\big|z_1, z_2, z_3\big], \mathbb{E}\big[z_4\mathbf{1}_{\{z_4\geq 0\}}\big|z_1, z_2, z_3\big]\Big\} = \max\Big\{ -\frac{a_4\sigma_4^2}{4}, -\frac{\sigma_4}{\sqrt{2\pi}}\Big\}.
$$

Then we have

$$
-\mathbb{E}[\ell'(a_1|z_1| + a_2z_2 + a_3z_3 + a_4z_4)z_3^2z_4] \geq \max\Big\{ -\frac{a_4\sigma_4^2}{4}, -\frac{\sigma_4}{\sqrt{2\pi}}\Big\}\mathbb{E}[z_3^2] = \max\Big\{ -\frac{a_4\sigma_3^2\sigma_4^2}{4}, -\frac{\sigma_3^2\sigma_4}{\sqrt{2\pi}}\Big\}.
$$
(E.24)

On the other hand, we have

$$
\begin{aligned}
&-\ell'(a_1|z_1| + a_2z_2 + a_3z_3 - a_4z_4) + \ell'(a_1|z_1| + a_2z_2 + a_3z_3 + a_4z_4) \\
=&\frac{1}{1 + e^{a_1|z_1|+a_2z_2+a_3z_3-a_4z_4}} - \frac{1}{1 + e^{a_1|z_1|+a_2z_2+a_3z_3+a_4z_4}} \\
=&\frac{e^{a_1|z_1|+a_2z_2+a_3z_3}\big(e^{a_4z_4} - e^{-a_4z_4}\big)}{\big(1 + e^{a_1|z_1|+a_2z_2+a_3z_3-a_4z_4}\big)\big(1 + e^{a_1|z_1|+a_2z_2+a_3z_3+a_4z_4}\big)} \\
\geq&\frac{e^{a_1|z_1|+a_2z_2+a_3z_3}\big(e^{a_4z_4} - e^{-a_4z_4}\big)}{\big(e^{a_1|z_1|} + e^{a_1|z_1|+a_2z_2+a_3z_3-a_4z_4}\big)\big(e^{a_1|z_1|} + e^{a_1|z_1|+a_2z_2+a_3z_3+a_4z_4}\big)} \\
=&e^{-a_1|z_1|}\frac{e^{a_4z_4} - e^{-a_4z_4}}{e^{a_4z_4} + e^{-a_4z_4} + e^{a_2z_2+a_3z_3} + e^{-a_2z_2-a_3z_3}} \\
\geq&e^{-a_1|z_1|}\frac{e^{a_4z_4} - e^{-a_4z_4}}{e^{a_4z_4} + e^{-a_4z_4} + e^{a_2z_2+a_3z_3} + e^{-a_2z_2-a_3z_3}}\mathbf{1}_{\{z_4\geq\frac{1}{a_4}\}}\mathbf{1}_{\{|z_2|\leq\frac{1}{2a_2}\}}\mathbf{1}_{\{|z_3|\leq\frac{1}{2a_3}\}} \\
\geq&e^{-a_1|z_1|}\frac{e - e^{-1}}{2(e + e^{-1})}\mathbf{1}_{\{z_4\geq\frac{1}{a_4}\}}\mathbf{1}_{\{|z_2|\leq\frac{1}{2a_2}\}}\mathbf{1}_{\{|z_3|\leq\frac{1}{2a_3}\}} \\
\geq&\frac{1}{3}e^{-a_1|z_1|}\mathbf{1}_{\{z_4\geq\frac{1}{a_4}\}}\mathbf{1}_{\{|z_2|\leq\frac{1}{2a_2}\}}\mathbf{1}_{\{|z_3|\leq\frac{1}{2a_3}\}}.
\end{aligned}
$$
(E.25)

By applying result of (E.25) in (E.23) and Lemma E.5, we obtain

$$
\mathbb{E}\big[ -\ell'(a_1|z_1| + a_2z_2 + a_3z_3 + a_4z_4)z_4\big] \leq -\frac{1}{3}\mathbb{E}\big[e^{-a_1|z_1|}\big]\mathbb{E}\big[z_4\mathbf{1}_{\{z_4\geq\frac{1}{a_4}\}}\big]\mathbb{E}\big[z_3^2\mathbf{1}_{\{|z_3|\leq\frac{1}{2a_3}\}}\big]\mathbb{P}\Big(|z_2| \leq \frac{1}{2a_2}\Big)
$$

$$
\leq -\frac{\sigma_4}{192\pi^3 a_2 a_3^3\sigma_2\sigma_3}\max\Big\{e^{-\sigma_1a_1-1/2}, \frac{1}{\sigma_1a_1} - \frac{1}{\sigma_1^3a_1^3}\Big\}e^{-\frac{1}{8\sigma_2^2a_2^2} - \frac{1}{8\sigma_3^2a_3^2} - \frac{1}{2\sigma_4^2a_4^2}},
$$
(E.26)

where the last inequality holds because $\mathbb{E}\big[z_4\mathbf{1}_{\{z_4\geq 1/a_4\}}\big] = \frac{1}{\sigma_4\sqrt{2\pi}}\int_{1/a_4}^{\infty} xe^{-\frac{x^2}{2\sigma_4^2}}\,\mathrm{d}x = \frac{\sigma_4}{\sqrt{2\pi}}e^{-\frac{1}{2a_4^2\sigma_4^2}}$,

$\mathbb{E}\big[z_3^2\mathbf{1}_{\{|z_3|\leq\frac{1}{2a_3}\}}\big] = \frac{1}{\sigma_3\sqrt{2\pi}}\int_{-\frac{1}{2a_3}}^{\frac{1}{2a_3}} x^2e^{-\frac{x^2}{2\sigma_3^2}}\,\mathrm{d}x \geq \frac{1}{32a_3^3\sigma_3\sqrt{2\pi}}e^{-\frac{1}{8\sigma_3^2a_3^2}}$, and $\mathbb{P}(|z_2| \leq \frac{1}{2a_2}) = \frac{1}{\sigma_2\sqrt{2\pi}}\int_{-\frac{1}{2a_2}}^{\frac{1}{2a_2}} e^{-\frac{x^2}{2\sigma_2^2}}\,\mathrm{d}x \leq \frac{1}{a_2\sigma_2\sqrt{2\pi}}e^{-\frac{1}{8\sigma_2^2a_2^2}}$. Similarly, we also have

$$
\begin{aligned}
&-\ell'(a_1|z_1| + a_2z_2 + a_3z_3 - a_4z_4) + \ell'(a_1|z_1| + a_2z_2 + a_3z_3 + a_4z_4) \\
=&\frac{1}{1 + e^{a_1|z_1|+a_2z_2+a_3z_3-a_4z_4}} - \frac{1}{1 + e^{a_1|z_1|+a_2z_2+a_3z_3+a_4z_4}} \\
=&\frac{e^{a_1|z_1|+a_2z_2+a_3z_3}\big(e^{a_4z_4} - e^{-a_4z_4}\big)}{\big(1 + e^{a_1|z_1|+a_2z_2+a_3z_3-a_4z_4}\big)\big(1 + e^{a_1|z_1|+a_2z_2+a_3z_3+a_4z_4}\big)}
\end{aligned}
$$

$$\leq \frac{e^{a_1|z_1|+a_2z_2+a_3z_3}(e^{a_4z_4}-e^{-a_4z_4})}{(1+e^{a_2z_2+a_3z_3-a_4z_4})(1+e^{a_2z_2+a_3z_3+a_4z_4})} = e^{a_1|z_1|}\frac{e^{a_4z_4}-e^{-a_4z_4}}{e^{a_4z_4}+e^{-a_4z_4}+e^{a_2z_2+a_3z_3}+e^{-a_2z_2-a_3z_3}}$$

$$\leq e^{a_1|z_1|}\frac{e^{a_4z_4}-e^{-a_4z_4}}{e^{a_2z_2+a_3z_3}+e^{-a_2z_2-a_3z_3}} \leq e^{a_1|z_1|}e^{-|a_2z_2+a_3z_3|}(e^{a_4z_4}-e^{-a_4z_4}) \leq e^{a_1|z_1|}e^{-a_2|z_2|}e^{a_3|z_3|}(e^{a_4z_4}-e^{-a_4z_4})$$

$$(E.27)$$

By applying result of (E.27), Lemma E.5, Lemma E.7 and Lemma E.8 in (E.23), we have

$$\mathbb{E}[-\ell'(a_1|z_1|+a_2z_2+a_3z_3+a_4z_4)z_3^2z_4] \geq -\mathbb{E}[e^{a_1|z_1|}]\mathbb{E}[e^{-a_2|z_2|}]\mathbb{E}[z_3^2e^{a_3|z_3|}]\mathbb{E}[z_4(e^{a_4z_4}-e^{-a_4z_4})\mathbf{1}_{\{z_4\geq 0\}}]$$

$$\geq -4\sqrt{\frac{2}{\pi}}\frac{\sigma_3^2\sigma_4^2a_4(a_3^2\sigma_3^2+1)}{\sigma_2a_2}e^{\frac{\sigma_1^2a_1^2+\sigma_3^2a_3^2+\sigma_4^2a_4^2}{2}}.$$

$$(E.28)$$

Combining (E.24), (E.26), and (E.28), we finish the proof for (E.22). $\square$

**Lemma E.13.** Let $z_1 \sim N(0,\sigma_1^2)$, $z_2 \sim N(0,\sigma_2^2)$, $z_3 \sim N(0,\sigma_3^2)$, $z_4 \sim N(0,\sigma_4^2)$ and $z_5 \sim N(0,\sigma_5^2)$ be five independent Gaussian random variables, and $a_1, a_2, a_3, a_4, a_5$ be five non-negative scalars. Then it holds that

$$-\mathbb{E}[\ell'(a_1|z_1|+a_2z_2+a_3z_3+a_4z_4+a_5z_5)z_3z_4z_5] \geq -8\sqrt{\frac{2}{\pi}}\frac{1}{\sigma_2a_2}e^{\frac{\sigma_1^2a_1^2}{2}}\prod_{i=3}^5\left(\frac{\sigma_i}{\sqrt{2\pi}}+\sigma_i^2a_ie^{\frac{\sigma_i^2a_i^2}{2}}\right);$$

$$-\mathbb{E}[\ell'(a_1|z_1|+a_2z_2+a_3z_3+a_4z_4+a_5z_5)z_3z_4z_5] \leq 8\sqrt{\frac{2}{\pi}}\frac{1}{\sigma_2a_2}e^{\frac{\sigma_1^2a_1^2}{2}}\prod_{i=3}^5\left(\frac{\sigma_i}{\sqrt{2\pi}}+\sigma_i^2a_ie^{\frac{\sigma_i^2a_i^2}{2}}\right).$$

$$(E.29)$$

*Proof of Lemma E.13.* By the law of total expectation, we have

$$-\mathbb{E}[\ell'(a_1|z_1|+a_2z_2+a_3z_3+a_4z_4+a_5z_5)z_3z_4z_5]$$

$$=\mathbb{E}\Big[\mathbb{E}[-\ell'(a_1|z_1|+a_2z_2+a_3z_3+a_4z_4+a_5z_5)z_3z_4z_5\big|z_3,z_4,z_5]\Big]$$

$$=\underbrace{\mathbb{E}\Big[\mathbb{E}[-\ell'(a_1|z_1|+a_2z_2+a_3z_3+a_4z_4+a_5z_5)z_3z_4\mathbf{1}_{\{z_3z_4\geq 0\}}z_5\big|z_3,z_4,z_5]\Big]}_{\textcircled{1}}$$

$$+\underbrace{\mathbb{E}\Big[\mathbb{E}[-\ell'(a_1|z_1|+a_2z_2+a_3z_3+a_4z_4+a_5z_5)z_3z_4\mathbf{1}_{\{z_3z_4< 0\}}z_5\big|z_3,z_4,z_5]\Big]}_{\textcircled{2}}$$

Since $-\ell'$ is a decreasing function and $z_5$ has a zero-centered symmetric density function. We can conclude that $\textcircled{1} \leq 0$ and $\textcircled{2} \geq 0$, and

$$\textcircled{1} \leq -\mathbb{E}[\ell'(a_1|z_1|+a_2z_2+a_3z_3+a_4z_4+a_5z_5)z_3z_4z_5] \leq \textcircled{2}.$$

Next, we provide a lower bound for $\textcircled{1}$ and an upper bound for $\textcircled{2}$. Similar to the proof of Lemma E.12, we have

$$\textcircled{1} \geq -\mathbb{E}[e^{a_1|z_1|}]\mathbb{E}[e^{-a_2|z_2|}]\mathbb{E}[z_3z_4e^{a_3|z_3|+a_4|z_4|}\mathbf{1}_{\{z_3z_4\geq 0\}}]\mathbb{E}[z_5e^{a_5z_5}\mathbf{1}_{\{z_5\geq 0\}}]$$

$$= -2\mathbb{E}[e^{a_1|z_1|}]\mathbb{E}[e^{-a_2|z_2|}]\mathbb{E}[z_3e^{a_3z_3}\mathbf{1}_{\{z_3\geq 0\}}]\mathbb{E}[z_4e^{a_4z_4}\mathbf{1}_{\{z_4\geq 0\}}]\mathbb{E}[z_5e^{a_5z_5}\mathbf{1}_{\{z_5\geq 0\}}]$$

$$\geq -8\sqrt{\frac{2}{\pi}}\frac{1}{\sigma_2a_2}e^{\frac{\sigma_1^2a_1^2}{2}}\prod_{i=3}^5\left(\frac{\sigma_i}{\sqrt{2\pi}}+\sigma_i^2a_ie^{\frac{\sigma_i^2a_i^2}{2}}\right).$$

The last inequality is derived by applying Lemma E.5, Lemma E.7 and Lemma E.8. On the other hand, we can obtain that

$$\textcircled{2} \leq -\mathbb{E}[e^{a_1|z_1|}]\mathbb{E}[e^{-a_2|z_2|}]\mathbb{E}[z_3z_4e^{a_3|z_3|+a_4|z_4|}\mathbf{1}_{\{z_3z_4< 0\}}]\mathbb{E}[z_5e^{a_5z_5}\mathbf{1}_{\{z_5\geq 0\}}]$$

$$= 2\mathbb{E}[e^{a_1|z_1|}]\mathbb{E}[e^{-a_2|z_2|}]\mathbb{E}[z_3e^{a_3z_3}\mathbf{1}_{\{z_3\geq 0\}}]\mathbb{E}[z_4e^{a_4z_4}\mathbf{1}_{\{z_4\geq 0\}}]\mathbb{E}[z_5e^{a_5z_5}\mathbf{1}_{\{z_5\geq 0\}}]$$

$$\leq 8\sqrt{\frac{2}{\pi}}\frac{1}{\sigma_2a_2}e^{\frac{\sigma_1^2a_1^2}{2}}\prod_{i=3}^5\left(\frac{\sigma_i}{\sqrt{2\pi}}+\sigma_i^2a_ie^{\frac{\sigma_i^2a_i^2}{2}}\right).$$

This finishes the proof. $\square$

**Lemma E.14.** Let $z_1, z_2, z_3$ be three standard Gaussian random variables, and $z_i$, $z_j$ are either independent or $z_i = z_j$ for $i \neq j$. Then it holds that $\mathbb{E}\big[|z_1 z_2 z_3|\big] \leq 2\sqrt{\frac{2}{\pi}}$.

*Proof of Lemma E.13.* There are three possible cases:

1. $z_1, z_2, z_3$ are all independent. Then $\mathbb{E}\big[|z_1 z_2 z_3|\big] = \mathbb{E}\big[|z_1|\big]\mathbb{E}\big[|z_2|\big]\mathbb{E}\big[|z_3|\big] = \frac{2}{\pi}\sqrt{\frac{2}{\pi}}$

2. Two of $z_1, z_2, z_3$ are equal while another one is independent with these two random variables. W.L.O.G, we assume $z_1 = z_2$ and $z_3$ is independent with both $z_1, z_2$. Then $\mathbb{E}\big[|z_1 z_2 z_3|\big] = \mathbb{E}\big[z_1^2\big]\mathbb{E}\big[|z_3|\big] = \sqrt{\frac{2}{\pi}}$

3. $z_1, z_2, z_3$ are all equal. Then $\mathbb{E}\big[|z_1 z_2 z_3|\big] = \mathbb{E}\big[|z_1|^3\big] = 2\sqrt{\frac{2}{\pi}}$

This finishes the proof. $\square$

### E.3 CONCENTRATION RESULTS

**Lemma E.15.** Let $z_1, z_2, \cdots, z_D \overset{i.i.d.}{\sim} N(0, \sigma_x^2)$. Then with probability at least $1 - \sqrt{\frac{2}{\pi}}\frac{D^{3/4}}{c_1}e^{-c_1^2\sqrt{D}/2}$, it holds that $|z_i| \leq c_1 \sigma_x D^{1/4}$ for all $i \in [D]$ and any constant $c_1$.

*Proof of Lemma E.15.* By Mills ratio, we can obtain that

$$\mathbb{P}\Big(|z_i| > c_1 \sigma_x D^{1/4}\Big) \leq \sqrt{\frac{2}{\pi}}\frac{1}{c_1 D^{1/4}}\exp\Big(-\frac{c_1^2\sqrt{D}}{2}\Big).$$

By applying a union bound over all $i \in [D]$, we complete the proof. $\square$

**Lemma E.16.** For $\mathbf{x}_1, \mathbf{x}_2, \cdots, \mathbf{x}_D$ defined in Definition 2.1, it holds that

$$\|\mathbf{x}_j\|_2 \leq \sigma_x \sqrt{d} + \sigma_x \sqrt{D}$$

with probability at least $1 - De^{-\frac{D}{2}}$ for all $j \in [D]$.

*Proof of Lemma E.16.* Since $\|\cdot\|_2$ is 1-Lipschitz continuous, then by Theorem 2.26 in Wainwright (2019), we can obtain that

$$\mathbb{P}\Big(\|\mathbf{x}_j\|_2 - \mathbb{E}\big[\|\mathbf{x}_j\|_2\big] \geq \sigma_x \sqrt{D}\Big) \leq e^{-\frac{D}{2}}.$$

Besides, by Jensen's inequality, we also have

$$\mathbb{E}\big[\|\mathbf{x}_j\|_2\big] \leq \sqrt{\mathbb{E}\big[\|\mathbf{x}_j\|_2^2\big]} = \sigma_x \sqrt{d}.$$

Applying a union-bound over all $j \in [D]$ completes the proof. $\square$

**Lemma E.17.** For $\mathbf{x}_j^{(i)}$ defined in Definition 3.1, it holds that

$$\big|\langle \mathbf{v}^*, \mathbf{x}_j^{(i)}\rangle\big| \leq c\widetilde{\sigma}_x D^{1/4}$$
$$\big\|\mathbf{x}_j^{(i)}\big\|_2 \leq \sqrt{2}\widetilde{\sigma}_x\Big(\sqrt{d} + \sqrt{D}\Big)$$

with probability at least $1 - ne^{-c'\sqrt{D}}$ for all $i \in [n]$ and $j \in [D]$, where $c$ is any positive absolute constant, and $c'$ is positive constant solely depending on $c$.

*Proof of Lemma E.17.* By definition of $\|\cdot\|_{\psi_2}$, we have $\left|\mathbb{E}[\mathbf{x}_{j,k}^{(i)}]\right| \leq \sqrt{2}\widetilde{\sigma}_x$ and $\mathbb{E}[\mathbf{x}_{j,k}^{(i)}]^2 \leq 2\widetilde{\sigma}_x^2$ for all $i \in [n]$, $j \in [D]$ and $k \in [d]$. By Hoeffding inequality, we have

$$\mathbb{P}\left(\left|\left|\langle \mathbf{v}^*, \mathbf{x}_j^{(i)} \rangle\right| - \mathbb{E}\left[\left|\langle \mathbf{v}^*, \mathbf{x}_j^{(i)} \rangle\right|\right]\right| \geq c\widetilde{\sigma}_x D^{1/4}\right) \leq 2\exp\left(-\frac{c^2\sqrt{D}}{2}\right).$$

Besides, by Bernstein's inequality, we also have

$$\mathbb{P}\left(\left|\|\mathbf{x}_j^{(i)}\|_2^2 - \mathbb{E}\left[\|\mathbf{x}_j^{(i)}\|_2^2\right]\right| \geq \widetilde{2}\sigma_x^2 D\right) \leq 2\exp(-c''D), \tag{E.30}$$

where $c''$ is an absolute constant. Applying a union-bound over all $i \in [n]$ and $j \in [D]$ completes the proof. $\qquad\square$

## E.4 ORTHOGONALITY AND NORM OF DISCRETE SINE TRANSFORM

**Lemma E.18.** For positional encodings $\mathbf{p}_1, \mathbf{p}_2, \cdots, \mathbf{p}_D$ defined in (2.1), it holds that $\|\mathbf{p}_j\| = \sqrt{\frac{D+1}{2}}$ for all $j \in [D]$ and $\langle \mathbf{p}_j, \mathbf{p}_{j'} \rangle = 0$ for all $j \neq j' \in [D]$.

*Proof of Lemma E.18.* This lemma is equal to prove that

$$\sum_{j=1}^{D} \sin\left(\frac{\pi k j}{D+1}\right) \sin\left(\frac{\pi k' j}{D+1}\right) = \frac{D+1}{2}\delta_{kk'}, \quad \forall\, k, k' \in [D],$$

where $\delta_{kk'} = 1$ when $k = k'$ and $\delta_{kk'} = 0$ when $k \neq k'$. Applying the form $\sin(x) = \{\exp(ix) - \exp(-ix)\}/(2i)$, we can rewrite the left side above by

$$\sum_{j=1}^{D} \sin\left(\frac{\pi k j}{D+1}\right) \sin\left(\frac{\pi k' j}{D+1}\right) = -\frac{1}{4}\sum_{j=0}^{D}\left\{\exp\left(\frac{i\pi(k+k')}{D+1}j\right) - \exp\left(\frac{i\pi(k-k')}{D+1}j\right)\right.$$
$$\left. - \exp\left(-\frac{i\pi(k-k')}{D+1}j\right) + \exp\left(-\frac{i\pi(k+k')}{D+1}j\right)\right\}.$$

When $k + sk' \neq 0$, the geometric partial sum shows that

$$\sum_{j=0}^{D} \exp\left(\pm\frac{i\pi(k+sk')}{D+1}j\right) = \frac{\exp\{\pm i\pi(k+sk')\} - 1}{\exp\{\pm i\pi(k+sk')/(D+1)\} - 1},$$

where $s \in \{+1, -1\}$. In this case, it is required that $k + sk' \neq 0$. We have that when $k + sk' \neq 0$, if $k + sk'$ is even, the terms will vanish and then the equation equals 0. When $k + sk'$ is odd, it must hold that one of $k$ and $k'$ is odd and the other is even, under this case $\exp(\pm i\pi(k+sk')) = -1$, the term compensates as

$$\frac{1}{\exp\{ix\} - 1} + \frac{1}{\exp\{-ix\} - 1} = -1.$$

This indicates that the equation above equals 0. We conclude that when $k + sk' \neq 0$,

$$\sum_{j=1}^{D} \sin\left(\frac{\pi k j}{D+1}\right) \sin\left(\frac{\pi k' j}{D+1}\right) = 0.$$

We consider the case $k + sk' = 0$. By the definition, it only holds when $s = -1$ and $k = k'$. In such case, simple algebra shows that

$$\sum_{j=0}^{D} \sin\left(\frac{\pi k j}{D+1}\right) \sin\left(\frac{\pi k' j}{D+1}\right) = \frac{D+1}{2},$$

which completes the proof of Lemma E.18. $\qquad\square$

### E.5 SEQUENCE ITERATION BOUND

The following lemma is inspired by Cao et al. (2023); Meng et al. (2024); Zhang et al..

**Lemma E.19.** Suppose that a positive sequence $x_t$, $t \geq 0$ follows the iterative formula

$$x_{t+1} = x_t + cx_t^{-a}$$

for some positive constant $a, c > 0$. Then it holds that

$$\left((a+1)ct + x_0^{a+1}\right)^{\frac{1}{a+1}} \leq x_t \leq cx_0^{-a} + \left((a+1)ct + x_0^{a+1}\right)^{\frac{1}{a+1}}$$

for all $t \geq 0$.

*Proof of Lemma E.19.* We first show the lower bound of $x_t$. Consider a continuous-time sequence $\underline{x}_t$, $t \geq 0$ defined by the integral equation with the same initialization.

$$\underline{x}_t = \underline{x}_0 + c \cdot \int_0^t \underline{x}_\tau^{-a} \mathrm{d}\tau, \quad \underline{x}_0 = x_0. \tag{E.31}$$

Note that $\underline{x}_t$ is obviously an increasing function of $t$. Therefore we have

$$\underline{x}_{t+1} = \underline{x}_t + c \cdot \int_t^{t+1} \underline{x}_\tau^{-a} \mathrm{d}\tau$$

$$\leq \underline{x}_t + c \cdot \int_t^{t+1} \underline{x}_t^{-a} \mathrm{d}\tau$$

$$= \underline{x}_t + c\underline{x}_t^{-a}$$

for all $t \in \mathbb{N}$. Comparing the above inequality with the iterative formula of $\{x_t\}$, we conclude by the comparison theorem that $x_t \geq \underline{x}_t$ for all $t \in \mathbb{N}$. Note that (E.31) has an exact solution

$$\underline{x}_t = \left((a+1)ct + x_0^{a+1}\right)^{\frac{1}{a+1}}$$

Therefore we have

$$x_t \geq \left((a+1)ct + x_0^{a+1}\right)^{\frac{1}{a+1}}$$

for all $t \in \mathbb{N}$, which completes the first part of the proof. Now for the upper bound of $x_t$, we have

$$x_t = x_0 + c \cdot \sum_{\tau=0}^{t-1} x_\tau^{-a}$$

$$\leq x_0 + c \cdot \sum_{\tau=0}^{t} \left((a+1)c\tau + x_0^{a+1}\right)^{-\frac{a}{a+1}}$$

$$= x_0 + \frac{c}{x_0^a} + c \cdot \sum_{\tau=1}^{t} \left((a+1)c\tau + x_0^{a+1}\right)^{-\frac{a}{a+1}}$$

$$\leq x_0 + \frac{c}{x_0^a} + c \cdot \int_0^t \left((a+1)c\tau + x_0^{a+1}\right)^{-\frac{a}{a+1}} \mathrm{d}\tau,$$

where the second inequality follows by the lower bound of $x_t$ as the first part of the result of this lemma. Therefore we have

$$x_t \leq x_0 + cx_0^{-a} + \left((a+1)ct + x_0^{a+1}\right)^{\frac{1}{a+1}} - x_0$$

$$= cx_0^{-a} + \left((a+1)ct + x_0^{a+1}\right)^{\frac{1}{a+1}},$$

which completes the proof of Lemma E.19. $\qquad \square$

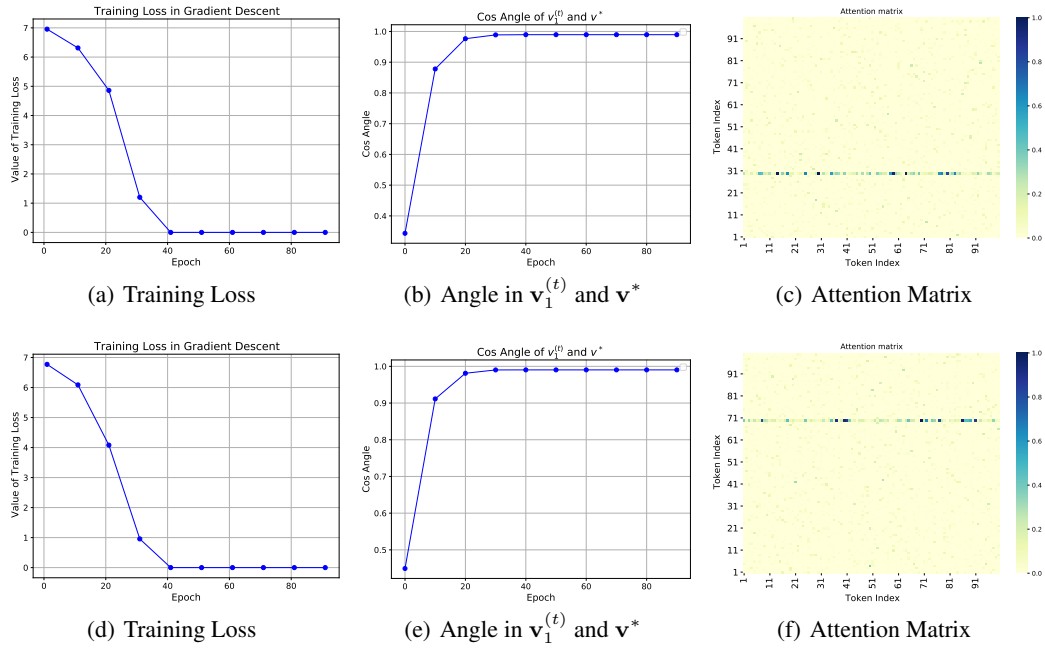

Figure 4: Figures on training loss, cosine similarity and attention matrix. The first line presents the training results with $j^* = 30$. The second line shows the training results for $j^* = 70$.

## F    ADDITIONAL SIMULATION RESULTS.

In this section, we present additional simulation results with larger numbers of variable groups $D$ and higher variable dimensions $d$. Specifically, we generate data according to Definition 2.1, setting $\sigma_x = 0.25$ and $(n, d, D) = (10000, 100, 100)$. We consider two scenarios with $j^* = 30$ and $j^* = 70$, respectively, and investigate whether the attention matrix effectively concentrates on the specified $j^*$, even under high-dimensional settings designed to mimic the scale of image data. The vector $\mathbf{v}^*$ is randomly generated and kept fixed throughout the simulations.

Due to the large amount of sample size $n$, we consider the SGD training with batch size $64$. We set the learning rate $\eta = 0.01$ and train the model for $100$ epochs. During the training process, we plot training loss and the cosine similarity $\frac{\langle \mathbf{v}_1^{(t)}, \mathbf{v}^* \rangle}{\|\mathbf{v}_1^{(t)}\| \|\mathbf{v}^*\|}$. After the training loss converges at the final epoch, we calculate the attention score matrix for each sample and display the heatmap of the average attention score matrix across all samples.

As shown in Figure 4, the training loss steadily decreases, eventually approaching zero after sufficient training iterations, and $\mathbf{v}_1^{(t)}$ rapidly aligns with the direction of $\mathbf{v}^*$ even when $d$ and $D$ have a higher dimension. More interestingly, when the value of $j^*$ varies, the attention mechanism consistently adapts to the target index. For instance, when $j^* = 30$, the attention matrix sharply focuses on the 30th row and effectively isolates the label-relevant group. The focus shifts with the same precision to $j^* = 70$. This behavior highlights the model's ability to redirect its attention to the specified index, even in high-dimensional settings designed to mimic the complexity of image data.

## G    REAL DATA EXPERIMENTS

In this section, we conduct experiments using the CIFAR-10 dataset, where each image has a shape of $3 \times 32 \times 32$, representing three color channels (RGB). For this experiment, we select two labels, "Frog" and "Airplane," and use 500 images from each label. To prepare the input for our framework, each CIFAR-10 image is embedded as either the first patch (positioned at (1,1)) or the 25th patch (positioned at (4,4)) in a grid, while the remaining 48 patches are filled with noise. These noise

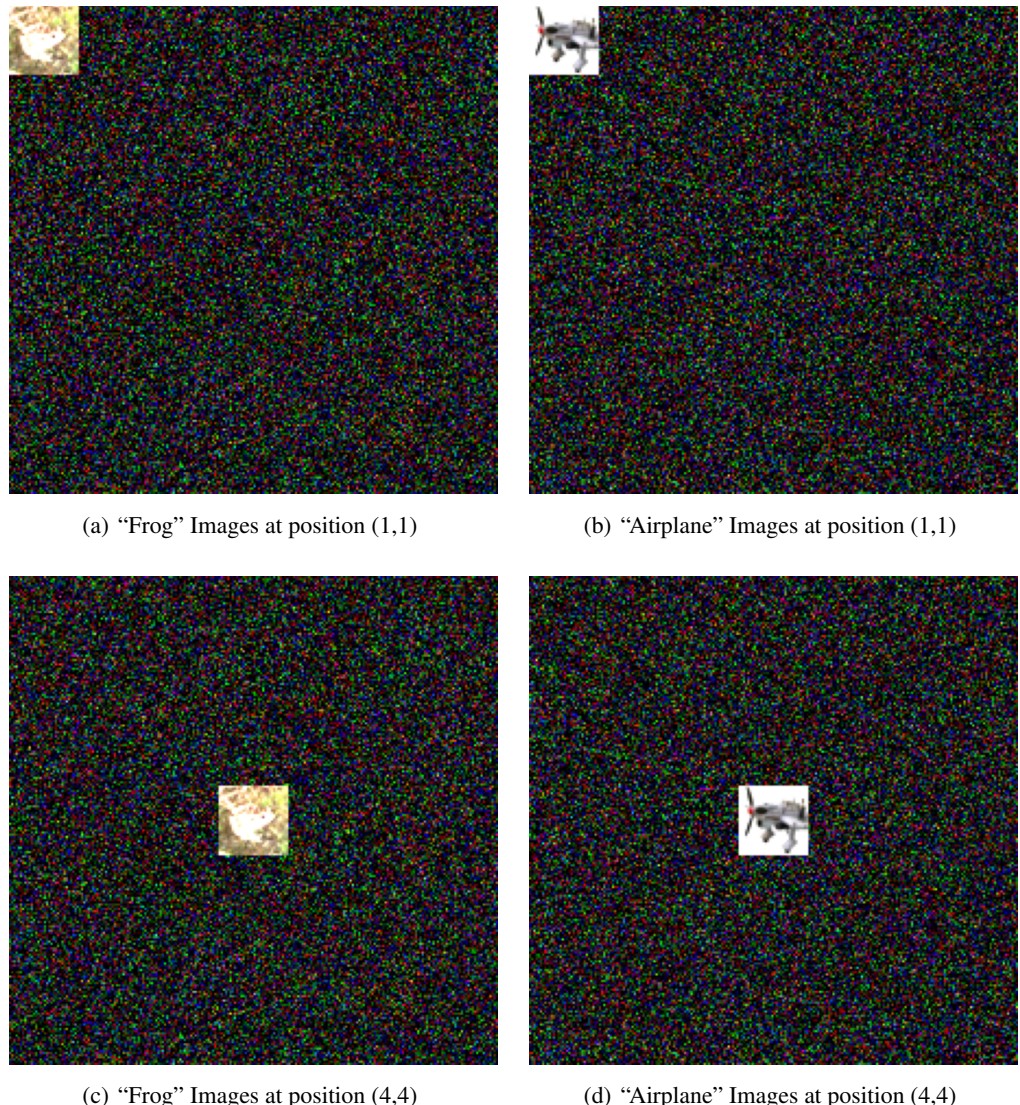

(a) "Frog" Images at position (1,1)

(b) "Airplane" Images at position (1,1)

(c) "Frog" Images at position (4,4)

(d) "Airplane" Images at position (4,4)

Figure 5: Examples of embedded images. Figure 5(a) and Figure 5(b) show images labeled as "Frog" and "Airplane," respectively, embedded at position (1,1) with a token index of 1. Figure 5(c) and Figure 5(d) show images labeled as "Frog" and "Airplane," respectively, embedded at position (4,4) with a token index of 25.

patches, each of size $3 \times 32 \times 32$, are generated using random values sampled from a Gaussian distribution with a mean of $0$ and a standard deviation of $1/3$. This arrangement forms a $7 \times 7$ grid (49 patches in total), resulting in a final input with dimensions of $3 \times 224 \times 224$. Examples of the processed images are shown in Figure 5.

We apply a one-layer transformer to learn from the processed images. Each patch, including the original CIFAR-10 image and the 48 noise patches, is flattened into a vector of size $3 \times 32 \times 32 = 3072$, resulting in a total of 49 vectors corresponding to the $7 \times 7$ grid of patches. These vectors, each with a dimension of 3072, form the input sequence for the transformer model. As shown in Figure 5(a) and Figure 5(b), images positioned at (1,1) correspond to $j^* = 1$, while Figure 5(c) and Figure 5(d) show images positioned at (4,4), corresponding to $j^* = 25$. In this setup, $d = 3072$ represents the dimensionality of the data, and $D = 49$ denotes the number of variable groups. The transformer model is initialized to $0$, and we train it using a batch size of $64$ and a learning rate of $10^{-3}$. For comparison, we also present results from directly applying logistic regression to the clean

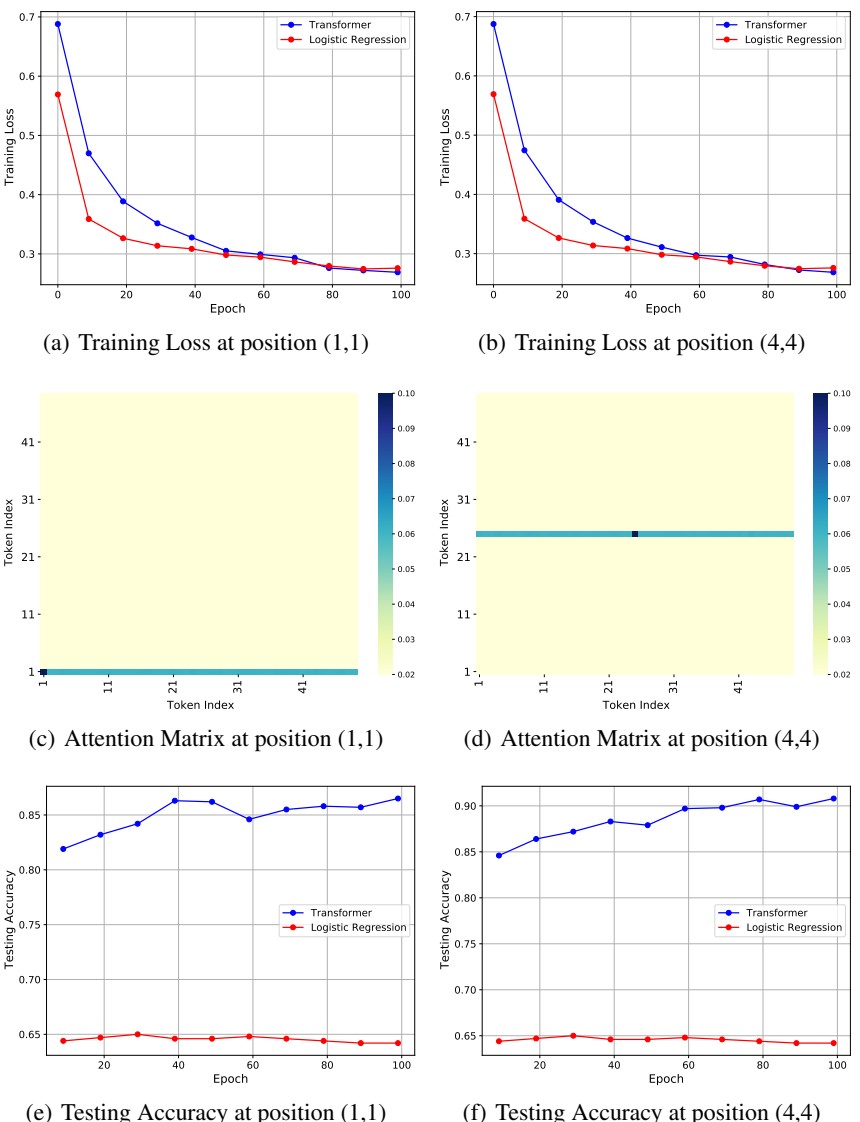

(a) Training Loss at position (1,1)

(b) Training Loss at position (4,4)

(c) Attention Matrix at position (1,1)

(d) Attention Matrix at position (4,4)

(e) Testing Accuracy at position (1,1)

(f) Testing Accuracy at position (4,4)

Figure 6: Experiment results on training loss, attention matrix and testing accuracy. The first column shows the results when images have position at (1,1). The second column shows the results when images have position at (4,4).

data points, where each CIFAR-10 image is flattened into a single vector of size 3072, and logistic regression is performed on these single vectors.

The results of our experiment are presented in Figures 6. Figures 6(a) and 6(b) show the training loss over 100 epochs using gradient descent. The plot demonstrates a steady decrease in the loss during training, which closely aligns with the decreasing trend observed in the clean logistic regression model. This indicates the effectiveness of the optimization process and the transformer model's ability to focus on the true images. Figures 6(c) and 6(d) display the attention matrices for the trained images, further confirming the model's ability to focus on relevant features. For images positioned at (1,1) with $j^* = 1$, the attention matrix shows that the model focuses predominantly on the first row, corresponding to the original CIFAR-10 image. Similarly, for images positioned at (4,4) with $j^* = 25$, the attention matrix highlights that the model concentrates on the 25th row, again corresponding to the true image. Figures 6(e) and 6(f) demonstrate the generalization performance

of the transformer on an unseen test dataset. The results show that the transformer maintains strong performance, further verifying its ability to effectively generalize to new data.

