# OpenReview forum: "Transformer Learns Optimal Variable Selection in Group-Sparse Classification"
_ICLR.cc/2025/Conference — ICLR 2025 Poster_

### Official Review · Reviewer_tEUD · 2024-10-31

**Soundness:** 3
**Presentation:** 2
**Contribution:** 3
**Rating:** 6
**Confidence:** 3

**Summary:**

This paper investigates the theoretical understanding of transformers, which have shown success in various applications but lack a clear theoretical foundation. The study focuses on how transformers can learn a statistical model with group sparsity, where only a subset of input variables (groups) influences the label. The research shows that a one-layer transformer can use attention mechanisms to select relevant variables and ignore irrelevant ones for classification. Additionally, it demonstrates that a well-pretrained one-layer transformer can adapt to new tasks with limited samples and achieve good accuracy. This study provides insights into how transformers learn structured data effectively.

**Strengths:**

1. This paper investigates the theoretical understanding of transformers, an important and interesting problem that may greatly benefit the deep learning community.
2. This paper is well-written and has a good motivation.
3. The conclusion that a one-layer transformer can use attention mechanisms to select relevant variables and ignore irrelevant ones for classification makes sense to me.

**Weaknesses:**

1. Though the problem is interesting, the conclusion, a one-layer transformer can use attention mechanisms to select relevant variables and ignore irrelevant ones for classification, seems to be well-known in the deep learning community and has already been verified by the original paper (Attention Is All You Need).
2. The theoretical understanding of transformers given by this paper mainly focuses on the well-pretrained one-layer transformer. However, it will be much more interesting to study the multi-layer transformer since the scaling law shows that the loss of transformer scales as a power-law with model size, dataset size, and the amount of compute used for training, with some trends spanning more than seven orders of magnitude.
3. The study focuses on how transformers can learn a statistical model with group sparsity, where only a subset of input variables (groups) influences the label. However, the concept of group sparse is not properly defined, making me confused to fully understand it. Further, what is a similar “group-sparse” structure as mentioned in the Introduction section? What kind of “group-sparse” inputs are similar?

**Questions:**

Please refer to the weaknesses section.

---

> ### Author Response · Authors · 2024-11-21
>
> Thank you for your constructive feedback! We address your comments as follows:
>
> >**Q1**:
> Though the problem is interesting, the conclusion, a one-layer transformer can use attention mechanisms to select relevant variables and ignore irrelevant ones for classification, seems to be well-known in the deep learning community and has already been verified by the original paper (Attention Is All You Need).
>
> **A1**:
> We would like to clarify that the main focus of this paper is to theoretically study the capacity of transformers to solve a classic statistical problem, which is linear classification with group sparsity (we have added explanations of the problem from line 105 to line 120 in our updated manuscript). Although the capability of transformers is intuitive, there still lacks a formal mathematical demonstration that a transformer model can indeed be trained to solve such problems. In this paper, we establish a precise theoretical analysis rigorously confirming that a one-layer transformer can effectively solve linear classification with group sparsity by efficiently attending to relevant features. To the best of our knowledge, [1] does not provide any theoretical analysis, and our results are the first to theoretically demonstrate this result. Establishing rigorous theoretical guarantees to backup people’s empirical beliefs is one of the major contributions of our work.
>
> >**Q2**:
> The theoretical understanding of transformers given by this paper mainly focuses on the well-pretrained one-layer transformer. However, it will be much more interesting to study the multi-layer transformer since the scaling law shows that the loss of transformer scales as a power-law with model size, dataset size, and the amount of compute used for training, with some trends spanning more than seven orders of magnitude.
>
> **A2**:
> There might be some misunderstanding regarding our paper. We would like to clarify that Theorem 3.2 (Theorem 2.2 in our updated manuscript) provides a rigorous and thorough theoretical analysis of the pre-training process of transformers from zero initialization. Moreover, Theorem 4.2 (Theorem 3.2 in our updated manuscript) studies the capacity of transformers when transferring to downstream tasks by providing the sample complexity. Therefore, our theoretical analysis covers both the pre-training stage and fine-tuning stage of one-layer transformers.
>
> Additionally, we acknowledge that studying the properties of multi-layer transformers is both important and interesting. Existing theoretical analyses on the training dynamics of transformers are mainly limited to one-layer models [2, 3, 4, 5, 6], and even one-layer transformers have not yet been thoroughly understood from a theoretical perspective.  Compared to existing results, our work is already one step towards more practical settings from the following two aspects:
> 1. In our settings, the parameters $\boldsymbol v, \boldsymbol W$ are trained simultaneously, while in [2, 3] only $\boldsymbol W$ is trained and $\boldsymbol v$ remains fixed throughout the training process. Additionally, [4] implemented a two-stage training strategy. In the first stage, only $\boldsymbol v$ get trained with $\boldsymbol W$ fixed, while in the second stage, only $\boldsymbol W$ get trained with $\boldsymbol v$ fixed.
>
> 2. In our settings, the parameters $\boldsymbol v, \boldsymbol W$ are initialized from zero, without any specific warm start which implies prior knowledge regarding data structure. However, [5] requires that the initialization of the value vector $\boldsymbol v$ strictly aligns with the ground truth, and [6] requires that the initialization satisfies a series of technical assumptions that can not be satisfied by random or zero initialization.
>
> Based on the discussions above, our work studies a more challenging (and slightly more practical) scenario of training both the $\boldsymbol v$ and $\boldsymbol W$ parameters in one-layer transformers from zero initialization compared with existing works. We believe our work is a cornerstone of this line of research, and can serve as a foundation towards more practical results.

---

> ### Author Response · Authors · 2024-11-21
>
> >**Q3**:
> The study focuses on how transformers can learn a statistical model with group sparsity, where only a subset of input variables (groups) influences the label. However, the concept of group sparse is not properly defined, making me confused to fully understand it. Further, what is a similar “group-sparse” structure as mentioned in the Introduction section? What kind of “group-sparse” inputs are similar?
>
> **A3**:
> Thank you for pointing out the absence of a definition and illustration for the concept of group sparsity. We have included a formal definition for "group sparse" learning problem in our updated manuscripts, from line 105 to line 120 for your reference. In brief, for a standard linear classification problem, we denote $\boldsymbol{\hat x} \in \mathbb{R}^p$ the feature vector, with the label $y$ determined by $y = \mathrm{sign}(\boldsymbol x^\top \boldsymbol\beta^*) $for some pre-defined linear ground truth $\boldsymbol{\beta^*}$. Besides, there exist predefined disjoint partitions for $[p]$ as $[p] = \cup_{j=1}^D G_j$.  Then we refer to this learning problem as "group sparse'' linear classification if the ground truth linear vector $\boldsymbol{\beta^*}$ satisfies that
> $$\mathrm{support}(\boldsymbol\beta^*):= \\{k:\boldsymbol\beta^*_k \neq 0; k\in [p]\\} \subset G_j$$
> for some $j\in [D]$.
>
> Besides, we provide further clarification to help understand the connections and different notations between the preceding definition of group sparsity and Definition 3.1 (Definition 2.1 in our updated manuscript) of the data model. We consider the setting where all $G_j$’s are of the same size $d$, implying $p=dD$. Besides, we convert the feature vector $\boldsymbol{\hat x} \in \mathbb{R}^{dD}$ into a matrix $\boldsymbol X\in \mathbb{R}^{d\times D}$, where each column $\boldsymbol x_j$ is the collection of the variables from $G_j$. Let $j^*$ be the index of the label-relevant group, i.e. $\mathrm{supp}(\boldsymbol\beta^*) \subset G_{j^*}$. Then we can denote $\boldsymbol v^*$ the $d$-dimensional vector obtained by restricting $\boldsymbol\beta^*$ on $G_{j^*}$, so that $\langle \boldsymbol\beta^*, \boldsymbol{\hat x}\rangle = \langle \boldsymbol v^*, \boldsymbol x_{j^*}\rangle$. Based on the preceding illustrations, it is clear that Definition 3.1 (Definition 2.1 in our updated manuscript) defines a linear classification problem with group sparsity.
>
> In addition, we also thank you for indicating the unclear presentation of the phrase “similar group sparse structure”, and we have replaced it with “the same group sparsity pattern”. Furthermore, the phrase “the same group sparsity pattern” indicates that the index of the label-relevant group of the downstream distribution $\tilde{\mathcal{D}}$ is the same as that of the original distribution $\mathcal{D}$, and it has been formally defined in our Definition 4.1 (Definition 3.1 in our updated manuscript).
>
> [1] Ashish Vaswani, Noam Shazeer, Niki Parmar, Jakob Uszkoreit, Llion Jones, Aidan N Gomez,
> Łukasz Kaiser, and Illia Polosukhin. Attention is all you need. NeurIPS, 2017.
>
> [2] Zihao Li, Yuan Cao, Cheng Gao, Yihan He, Han Liu, Jason Matthew Klusowski, Jianqing Fan, and Mengdi Wang. One-Layer Transformer Provably Learns One-Nearest Neighbor In Context. NeurIPS, 2024.
>
> [3] Davoud Ataee Tarzanagh, Yingcong Li, Christos Thrampoulidis, and Samet Oymak. Transformers as support vector machines. In NeurIPS 2023 Workshop on Mathematics of Modern Machine Learning, 2023.
>
> [4] Yuchen Li, Yuanzhi Li, and Andrej Risteski. How do transformers learn topic structure: Towards a mechanistic understanding. ICML, 2023.
>
> [5] Samy Jelassi, Michael Sander, and Yuanzhi Li. Vision transformers provably learn spatial structure. NeurIPS, 2022.
>
> [6] Hongkang Li, Meng Wang, Sijia Liu, and Pin-Yu Chen. A theoretical understanding of shallow vision transformers: Learning, generalization, and sample complexity. ICLR, 2023.

---

> > ### Comment · Reviewer_tEUD · 2024-12-03
> > **Response to the authors**
> >
> > The authors addressed most of my concerns so that I will raise my score to "6: marginally above the acceptance threshold"

---

> > > ### Author Response · Authors · 2024-12-03
> > >
> > > Dear Reviewer,
> > >
> > > We appreciate your decision to raise your score and we are delighted that our explanation has resolved your questions and concerns.  Thanks once again for your dedicated efforts in reviewing our paper!
> > >
> > > Best regards,
> > >
> > > Authors

---

> ### Author Response · Authors · 2024-11-26
>
> Dear Reviewer,
>
> Thank you for taking the time to review our paper. As the deadline for updating the manuscript is approaching, we would like to follow up with you regarding your comments. We believe that our response has addressed all your concerns. In our revision, following your suggestions, we have also added a concrete definition of group sparsity (lines 105 - 120 in the revised manuscript). We sincerely hope that you could check our response and revision, and reevaluate our work taking the following points into consideration:
>
> - Our work establishes rigorous theoretical guarantees on a classic statistical problem (linear classification with group sparsity) to back up people’s empirical belief (that transformers can use attention mechanisms to select relevant variables).
>
> - Our work develops novel analytical tools to theoretically study transformer models, and is an important step towards theoretical investigations of deeper and more complex transformer architectures.
>
> If you have any further questions, please let us know, and we will try our best to address them. Thank you.
>
> Best regards,
>
> Authors

---

> ### Author Response · Authors · 2024-12-02
>
> Dear Reviewer,
>
> We believe we have thoroughly addressed all your comments and concerns. However, we have not yet received any feedback following our response and revision. With the deadline for feedback only one day away, we sincerely hope you can review our response and revision at your earliest convenience. Thank you.
>
> Best regards,
>
> Authors

---

### Official Review · Reviewer_BJ8K · 2024-11-03

**Soundness:** 3
**Presentation:** 3
**Contribution:** 3
**Rating:** 6
**Confidence:** 3

**Summary:**

This paper analyzes the variable selection of the Transformer theoretically.

**Strengths:**

The author presented a novel theoretical analysis on how the transformer can learn to select variables for the group sparse classification, which is interesting and novel.

**Weaknesses:**

1. Given your previous definition of the notations, pls define $\Theta$ in advance （otherwise people would get confused about this common notation.What is the 'variable' in the paper? Are they some features or some attributes of the data that define the label? What is the $v_1$ in the Thereom 3.2-2？ The weight corresponding to the label 1 or the value vector (output from value matrix in Transformer)？\\
2. The proof stretch should be placed in the Supplementary instead of the main paper. \\
3. The experiments are conducted on synthetic data, however, in the higher dimensional images (e.g. 3*224*224), things would change. I have concerns on these too simple experiments. \\
4. The Lemma 5.1, I do not see why the $W^{(T^*)}_{1,2}$ holds and $W$ is diagonal. In NLP, it seems that the feature is correlated to its position and would correspond to the output, is that a too strong assumption?
5. Can you show a more detailed demonstration on why the inequality of $\alpha^(T^*)$ holds (or is just a definition?) and why Lemma5.2 can be incorporated into Lemma 5.3? Also, I do not think Lemma 5.2. holds as the underlying assumption is the Transformer is naturally with low error. Lemma 5.2 is more like a strong assumption and definition to me. \\
6. How you define the group sparse? What is this different from the standard classification?


I would like to raise my scores if more details are provided.

**Questions:**

See weakness

---

> ### Author Response · Authors · 2024-11-21
>
> Thank you for your detailed feedback!
>
> Before we address your questions in detail, we want to clarify that this paper's main focus is to provide a precise theoretical characterization when applying transformers to a classic statistical problem, which is linear classification with group sparsity. All the problem settings and assumptions for the theoretical derivation are outlined in Definitions 3.1, 4.1 (Definitions 2.1, 3.1 in our updated manuscript), and Theorems 3.2, 4.2  (Theorems 2.2, 3.2 in our updated manuscript). Our theoretical conclusions, including the results of Lemmas 5.1, 5.2 (Lemmas 4.1, 4.2 in our updated manuscript), are rigorously derived based on Definition 3.1 (Definition 2.1 in our updated manuscript) and the assumptions in Theorem 3.2 (Theorems 2.2 in our updated manuscript), and do not rely on any other assumptions or definitions. We will now address your comments as follows:
>
> >**Q1**:
> Given your previous definition of the notations, pls define $\Theta()$ in advance（otherwise people would get confused about this common notation.)
>
> **A1**:
> We provide all the definitions of mathematical notations at the end of the introduction section, including the definition of $\Theta()$. However, we still thank you for pointing this out. After double-checking the definitions for all notations, we found that we had omitted the definition of $\omega(\cdot)$, and we have updated from line 92 to line 94 in our manuscript for your reference.
>
> >**Q2**:
> What is the 'variable' in the paper? Are they some features or some attributes of the data that define the label?
>
> **A2**:
> Yes, your understanding is correct. The word 'variable' indicates the features of each data point. Specifically, each entry in the matrix $\boldsymbol X$ is a single variable and each column $\boldsymbol x_j$ in $\boldsymbol X$ for $j \in [D]$ is defined as a group of variables. And informally, “group sparsity” means that among all the groups of variables, only one group of variables determines the label of this data point, which is defined as the label-relevant group.
>
> Please note that the main goal of this paper is to study the capability of transformers in solving a classic statistical problem. Therefore, we use the terminology “variable selection”, which is a popular terminology in statistics.
>
> >**Q3**:
> What is the  $\boldsymbol v_1$ in the Thereom 3.2-2？ The weight corresponding to the label 1 or the value vector (output from value matrix in Transformer)？
>
> **A3**:
> Thank you for pointing out this unclear notation. $\boldsymbol v_1$ indicates the first block of the value vector $\boldsymbol v$, composed by the first $d$ entries of the value vector $\boldsymbol v$. We have clarified the definition of $\boldsymbol v_1$ and specified its dimension in Theorem 3.2 (Theorem 2.2 in our updated manuscript).
>
> Furthermore, we would like to further illustrate why we separate the value vector into two blocks. For each group of variables, i.e. each column $\boldsymbol x_j \in \mathbb{R}^d$,  we concatenate it with a position encoding $\boldsymbol p_j \in \mathbb{R}^D$.  Consequently, the new concatenated column is defined as $\boldsymbol z_j = [\boldsymbol x_j^\top, \boldsymbol p_j^\top]^\top \in \mathbb{R}^{d+D}$, while only the first $d$ entries is correlated with the label $y$ as $y$ is determined by $y =\mathrm{sign}(\boldsymbol x_{j^*}^\top v^*)$. As the value vector $\boldsymbol v$ is also of dimension $d+D$, we can observe that only the first $d$ entries interact with $\boldsymbol x_j$. Therefore, we separate the value vector into two blocks with the first block $\boldsymbol v_1$, having the dimension $d$. Then the directional alignment between $\boldsymbol v_1$ and $\boldsymbol v^*$ plays a key role in achieving correct classification.

---

> ### Author Response · Authors · 2024-11-21
>
> >**Q4**:
> The experiments are conducted on synthetic data, however, in the higher dimensional images (e.g. 3224224), things would change. I have concerns on these too simple experiments.
>
> **A4**:
> To address your concerns, we have conducted additional experiments on high-dimensional Gaussian data and CIFAR-10 image data,  as detailed in Appendix F and Appendix G of our updated manuscript respectively.
>
> In Appendix F, we present the experimental results on high-dimensional Gaussian data with $d=D=100$. We can observe that, despite the number of groups and the number of variables in each group being relatively large, the one-layer transformer can still attend to the label-relevant group. In particular, the $j^*$-th row of the attention score matrix takes the largest value among all the rows. Besides, the trained vector $v_1$ aligns in direction with the ground truth vector $v^*$ well, which is indicated by the plot of the cosine similarity curve. These observations demonstrate our theoretical findings still hold under the high-dimensional scenario.
>
> In Appendix G, we consider a binary classification task on the images from the label “Frog” and label “Airplane” in the CIFAR-10 dataset. For each label, we randomly sample 500 images. Besides, we extend the original images from dimension $3 \times 32 \times 32$ to $3 \times 224 \times 224$, with extended pixels filled by Gaussian noise. In other words, each extended image is composed of 49 patches, while the original true image appears on one patch and the other 48 patches are Gaussian noise. We provide some examples demonstrating the extended images in our updated manuscript. This design for the extended images aligns with our focus on “variable selection”, as one-layer transformers can only correctly classify the images by attending to the true image patch. Moreover, we also conduct logistic regression on the vectorized original image (with no noise patches). Our results show that by efficiently attending to the true image patch, one-layer transformers can achieve better test accuracy than logistic regression. These experiment results further validate our theory findings.
>
> >**Q5**:
> The Lemma 5.1, I do not see why the $\boldsymbol W_{1,2}^{(T^*)}$ holds and $\boldsymbol W$ is diagonal. In NLP, it seems that the feature is correlated to its position and would correspond to the output, is that a too strong assumption?
>
> **A5**:
> First, we would like to clarify again that, all conclusions in Lemma 5.1 (Lemma 4.1 in our updated manuscript) are rigorously derived solely based on our assumptions regarding the data distribution in Definition 3.1 (Definition 2.1 in our updated manuscript) and assumptions regarding the scale of parameters in Theorem 3.2 (Theorem 2.2 in our updated manuscript).
>
> Since we assume that our features $\boldsymbol X$ are generated from Gaussian distribution in Definition 3.1 (Definition 2.1 in our updated manuscript), then by the independence between the Gaussian features and fixed positional encoding and symmetry of Gaussian features (Specifically, we leverage a key property that the absolute value of a zero-mean Gaussian random variable is independent with its sign. This property is rigorously proved in Lemma E.1 on page 37 and is utilized in the proofs of Lemmas E.2, E.3, C.6, C.7, and C.8 in our updated manuscript.), we can rigorously derive that $\nabla_{\boldsymbol W_{1, 2}} \mathcal L(\boldsymbol v^{(t)}, \boldsymbol W^{(t)})$ and $\nabla_{\boldsymbol W_{2, 1}} \mathcal L(\boldsymbol v^{(t)}, \boldsymbol W^{(t)})$ is always $\boldsymbol 0$ throughout training. That’s why $\boldsymbol W_{1,2}^{(T^*)} = \boldsymbol 0$ holds and $\boldsymbol W$ is diagonal. The detailed proof is demonstrated in the proof for Proposition C.1 on page 17 of our updated manuscript (Proposition A.1 on page 16 of our original manuscript) for your reference.
>
> We acknowledge that the derivation of the diagonal structure of $ \boldsymbol W^{(T^*)}$ indeed relies on the independence between the features and positional encodings, which is induced by our assumption of Gaussian features. However, similar assumptions that features are generated from Gaussian distribution or Gaussian mixture distribution are widely considered in recent theoretical works [1, 2, 3, 4]. We would also like to clarify that our proof technique for Theorem 3.2 (Theorem 2.2 in our updated manuscript) doesn’t rely on $\boldsymbol W_{1,2}^{(T^*)}, \boldsymbol W_{2,1}^{(T^*)}$ to be exactly $ \boldsymbol 0$. Intuitively, as long as $\boldsymbol W_{2,2}^{(T^*)}$ can dominate the other blocks $\boldsymbol W_{1,1}^{(T^*)}, \boldsymbol W_{1,2}^{(T^*)}$ and $\boldsymbol W_{2,1}^{(T^*)}$, we can still conclude that $\mathcal{S}\_{j^*, j}^{(T^*)} \approx 1$, which is our desired result.

---

> ### Author Response · Authors · 2024-11-21
>
> >**Q6**:
> Can you show a more detailed demonstration on why the inequality of $\alpha^{(T^*)}$ holds (or is just a definition?) and why Lemma 5.2 can be incorporated into Lemma 5.3?
>
> **A6**:
> As we have clarified previously, the conclusion regarding the inequalities of $\alpha^{(T^*)}$ in Lemma 5.2 (Lemma 4.2 in our updated manuscript)  is rigorously derived solely based on our assumptions regarding the data distribution in Definition 3.1 (Definition 2.1 in our updated manuscript) and assumptions regarding the scale of parameters in Theorem 3.2  (Theorem 2.2 in our updated manuscript). The complete proof for conclusions in Lemma 5.2 is demonstrated from page 24 to page 28 and specifically the inequalities for $\alpha^{(T^*)}$ are from line 1424 to line 1445 for your reference.
>
> Here, we also briefly introduce the procedure for developing the inequalities for $\alpha^{(T^*)}$. Firstly, by the assumptions regarding the data distribution, we can derive both upper bound and lower bound of the iterative rule of $\alpha^{(t)}$ as follows:
> $$\alpha^{(t+1)} \geq \alpha^{(t)} + \sqrt{\frac{2}{\pi}}\frac{\eta}{2\sigma_x D}\frac{1}{\big(\alpha^{(t)}\big)^2}; \quad (1)$$
> $$\alpha^{(t+1)} \leq  \alpha^{(t)} + \sqrt{\frac{2}{\pi}}\frac{2\eta}{\sigma_x D}\frac{1}{\big(\alpha^{(t)}\big)^2}. \quad (2)$$
> We can further observe that both the upper-bound and the lower-bounds above are Euler discretizations of the ODE $\frac{\text{d} \alpha^{(t)}}{\text{d} t} = c \big(\alpha^{(t)}\big)^{-2}$ with different coefficient $c$. The fact that the solution to this ODE is $\alpha^{(t)} = \big(\alpha^{(0)} + 3ct \big)^{1/3}$ can well explain why both the upper and lower bounds for $\alpha^{(T^*)}$ are at the scale of $(T^*)^{1/3}$.  The rigorous proof regarding the bounds for discretized iterative inequalities (1) and (2) is demonstrated in Lemma E.19 (Lemma C.19 in the original manuscript). Since we derive matching upper and lower bounds for $\alpha^{(T^*)}$, we can rigorously plug these results into both the lower and upper bounds for $y\cdot f(Z, W^{(T^*)}, v^{(T^*)})$ in Lemma 5.3 (Lemma 4.3 in our updated manuscript). Then we can also get matching lower and upper bounds for $y\cdot f(Z, W^{(T^*)}, v^{(T^*)})$.
>
> >**Q7**:
> How you define the group sparse? What is this different from the standard classification?
>
> **A7**:
> Thank you for pointing out the absence of a definition and illustration for the concept of group sparsity. We have included a formal definition for "group sparse" learning problem in our updated manuscripts, from line 105 to line 120 for your reference. In brief, for a standard linear classification problem, we denote $\boldsymbol{\hat x} \in \mathbb{R}^p$ the feature vector, with the label $y$ determined by $y = \mathrm{sign}(\boldsymbol x^\top \boldsymbol\beta^*) $ for some pre-defined linear ground truth $\boldsymbol{\beta^*}$. Besides, there exist predefined disjoint partitions for $[p]$ as $[p] = \cup_{j=1}^D G_j$.  Then we refer to this learning problem as "group sparse'' linear classification if the ground truth linear vector $\boldsymbol{\beta^*}$ satisfies that
> $$\mathrm{support}(\boldsymbol\beta^*):= \\{k:\boldsymbol\beta^*_k \neq 0; k\in [p]\\} \subset G_j$$
> for some $j\in [D]$. This requirement for the ground-truth vector
> $\boldsymbol \beta^*$ clarifies the distinction between the standard classification and group sparse classification.
>
> Besides, we provide further clarification to help understand the connections and different notations between the definition above of group sparsity and Definition 3.1 (Definition 2.1 in our updated manuscript) of the data model. We consider the setting where all $G_j$’s are of the same size $d$, implying $p=dD$. Besides, we convert the feature vector $\boldsymbol{\hat x} \in \mathbb{R}^{dD}$ into a matrix $\boldsymbol X\in \mathbb{R}^{d\times D}$, where each column $\boldsymbol x_j$ is the collection of the variables from $G_j$. Let $j^*$ be the index of the label-relevant group, i.e. $\mathrm{supp}(\boldsymbol\beta^*) \subset G_{j^*}$. Then we can denote $\boldsymbol v^*$ the $d$-dimensional vector obtained by restricting $\boldsymbol\beta^*$ on $G_{j^*}$, so that $\langle \boldsymbol\beta^*, \boldsymbol{\hat x}\rangle = \langle \boldsymbol v^*, \boldsymbol x_{j^*}\rangle$. Based on the preceding illustrations, it is clear that Definition 3.1 defines a linear classification problem with group sparsity.
>
>
> [1] Ruiqi Zhang, Spencer Frei, and Peter L Bartlett. Trained transformers learn linear models in-context. JMLR, 2024.
>
> [2] Zixuan Wang, Stanley Wei, Daniel Hsu, and Jason D Lee. Transformers provably learn sparse token selection while fully-connected nets cannot. ICML, 2024.
>
> [3] Samy Jelassi, Michael Sander, and Yuanzhi Li. Vision transformers provably learn spatial structure. NeurIPS, 2022.
>
> [4] Spencer Frei, and Gal Vardi. Trained transformer classifiers generalize and exhibit benign overfitting in-context. arXiv preprint arXiv:2410.01774, 2024.

---

> ### Author Response · Authors · 2024-11-26
>
> Dear Reviewer,
>
> Thank you for your detailed and constructive comments. We have carefully addressed your questions in our response. In particular, we would like to emphasize again that we are confident that all the lemmas and theorems are established based on rigorous proofs. Please refer to our response above for our detailed explanations about Lemmas 5.1, 5.2 and 5.3 (Lemmas 4.1, 4.2 and 4.3 in our revised manuscript). Moreover, following your suggestion, we have also added experiments on high-dimensional (3x224x224) data, and the results demonstrate that our theoretical conclusions still hold in such high-dimensional cases.
>
> We are confident that our response and revision have addressed your concerns in detail. We would greatly appreciate it if you could review our response and the revised manuscript and reconsider your evaluation. We are happy to answer any further questions you may have. Thank you.
>
> Best regards,
>
> Authors

---

> ### Author Response · Authors · 2024-12-02
>
> Dear Reviewer,
>
> Thank you for your efforts in reviewing our paper. We are confident that we have addressed all your questions and concerns. However, we have not received any feedback from you since we submitted our response to your initial review. With the deadline for the discussion period approaching, we sincerely hope you can take a moment to review our response and revision. Thank you.
>
> Best regards,
>
> Authors

---

> > ### Comment · Reviewer_BJ8K · 2024-12-03
> >
> > I appreciate the efforts made by the authors, and the detailed explanation has addressed my concerns, I will raise my score to 6.

---

> > > ### Author Response · Authors · 2024-12-03
> > >
> > > Dear Reviewer,
> > >
> > > We are pleased to hear that our explanation addressed your questions and concerns. Thank you for raising your score and for your efforts in reviewing our paper!
> > >
> > > Best regards,
> > >
> > > Authors

---

### Official Review · Reviewer_CzbC · 2024-11-04

**Soundness:** 3
**Presentation:** 3
**Contribution:** 2
**Rating:** 6
**Confidence:** 4

**Summary:**

This paper investigates one-layer transformers trained on specific dataset, where the input variables are generated from multiple groups, while the true label of this input is determined by variables from a single group. Based on these simplifications and assumptions, it theoretically demonstrates that the one-layer transformers can almost attend to the variables from the label-relevant group. Moreover, it provides a tight lower and upper bound for the population cross-entropy loss of a one-layer transformer trained by gradient descent. It also shows that the well pre-trained one-layer transformers can be efficiently transferred to a downstream task sharing a similar “group-sparse” structure and further provides an improved generalization error bound for one-layer transformers fine-tuned by SGD, which surpasses that of linear logistic regression applied to vectorized features. The numerical experiment observations support the theoretical findings.

**Strengths:**

The paper is well written and easy to follow.

The theoretical analysis is thorough and the results align with the group sparsity assumption, although I did not check all the math and proof in detail.

In a sense, it provides new insights into sparsity analysis on the workings of attention based models.

**Weaknesses:**

(1) too much data assumption and model simplification. According to Definition 3.1, each patch x_j is i.i.d from Gaussian, and its label is determined from a given v (which can be learned from samples later). These data assumptions make group attention trivial.

(2) And the model is one-layer transformers, which may cannot capture attention-based models to a sufficiently satisfactory extent. With these simplifications, the contribution of this paper is limited, especially considering the pre-work from Jelassi et al. (2022).

**Questions:**

the input x_j is concatenated with position encoding, does Theorem 3.2 still hold if using addition instead of concatenation?

---

> ### Author Response · Authors · 2024-11-21
>
> Thank you for your constructive feedback! We address your comments as follows:
>
> >**Q1**:
> Too many data assumptions and model simplification. According to Definition 3.1, each patch $\boldsymbol x_j$ is i.i.d from Gaussian, and its label is determined from a given v (which can be learned from samples later). These data assumptions make group attention trivial.
>
> **A1**:
> We would like to emphasize the focus of this paper is to theoretically examine the performance of transformers when applied to a classic statistical problem, which is linear classification with “group sparsity”. As a case study on this classic problem, we believe it is reasonable to consider Gaussian data. Similar assumptions have been widely adopted in recent theoretical works on transformers [1, 2, 3, 4].
>
> Please note that the assumptions do not make the learning task trivial. The classic statistical problem we consider has been one of the hot topics in high-dimensional statistics for years. Our work gives the first study on how a transformer can be trained to solve such a task, which we believe is highly nontrivial. Moreover, we would like to point out that the Gaussian data assumption does not trivialize the problem either: we assume that all features are i.i.d. Gaussian random variables, which means that the data input alone provides no information about the label-relevant group of variables the transformer should select. Therefore, we believe that the problem we study is highly nontrivial.
>
> >**Q2**:
> The model is one-layer transformers, which may cannot capture attention-based models to a sufficiently satisfactory extent. With these simplifications, the contribution of this paper is limited, especially considering the pre-work from [3].
>
> **A2**:
> As we clarified in A1, the main focus of this paper is to study the capacity of transformers in addressing a classical statistical problem. Our contribution is to establish a rigorous theoretical guarantee that a one-layer transformer can effectively solve linear classification with group sparsity by efficiently attending to relevant features. While we acknowledge that studying the properties of multi-layer transformers is both important and interesting, even one-layer transformers have not yet been thoroughly understood from a theoretical perspective. Therefore, we believe our work provides valuable insights into the theoretical understanding of transformers.
>
>
> Besides, we thank you for pointing out the potential confusion regarding the comparisons between [3] and this paper. We agree that such a discussion is essential and we hope that the following discussion can clarify your concerns:
>
> - [3] considers a learning task where tokens (more precisely, image patches) form groups, and [3] relies on an assumption that there are a large number of patches in each group to make sure the classification task is solvable. In comparison, our work studies a more classic statistical problem, and we do not require the size of each group to be large.
>
> - [3] considers a one-layer vision transformer defined as
> $$f(\boldsymbol X) = \sigma(\boldsymbol v^\top \boldsymbol X \mathrm{softmax}(\boldsymbol A))\mathbf{1}_D,$$
> where $\boldsymbol X$ is the sequences of tokens with each column $\boldsymbol x_j$ denoting a token, $\sigma(\cdot)$ represents the activation function, $\boldsymbol v$ indicates the value vector, and $\boldsymbol A$ is input matrix of the softmax function. Unlike the common design of transformers, they directly treat the entries of the matrix $\boldsymbol A$ as the trainable parameters and consider training $\boldsymbol A$ with gradient descent. In contrast, we consider softmax attention with the formulation $\mathrm{softmax}(\boldsymbol Z^\top \boldsymbol W \boldsymbol Z)$, and treat the coefficient matrix $\boldsymbol W$ as the trainable parameters, which aligns with the general design of transformers. Besides, the initialization of the value vector $\boldsymbol v$ in [3] is assumed to strictly align with the direction of ground truth vector $\boldsymbol v^*$, which is a strong and impractical assumption. In comparison, we consider general zero initializations.
>
> - [3] provides an upper bound on the number of iterations needed to achieve a population loss of $1/\mathrm{poly}(d)$, we establish matching upper and lower bounds on the number of iterations required to reach arbitrarily small population loss. Furthermore, we present a sample complexity analysis for transfer learning, which surpasses the conclusion of linear logistics regression on vectorized inputs from the PAC learning theory. In contrast, [3] does not include such sample complexity analyses.
>
> We have also included the discussion above in Appendix B (page 14) of our updated manuscript for your reference.

---

> > ### Comment · Reviewer_CzbC · 2024-11-27
> >
> > Thank you for the through response. My point is that from Definition 2.1 (updated version), if any two patches x_i and x_j are generated from the same label, then x_i and x_j should be highly related, indirectly leading to group sparsity. These assumption will make your proof much easier. Compared to [3], your assumption is much simpler, but you can make more tight bound as you claimed. Corrected me if I am wrong here.

---

> > > ### Author Response · Authors · 2024-11-28
> > >
> > > Dear Reviewer,
> > >
> > > Thanks for your follow-up comments. We would like to clarify that while your comment
> > >
> > > *“if any two patches $\boldsymbol x_i$ and $\boldsymbol x_j$ are generated from the same label, then $\boldsymbol x_i$ and $\boldsymbol x_j$ should be highly related, indirectly leading to group sparsity”*
> > >
> > > accurately describes the intuition of [3], it does not apply to our problem setting. Please note that in our setting, only one of the patches is related to the label $y$, and the other patches are all independent Gaussians. Therefore, the model cannot rely on the correlations among patches (patches are not correlated) to learn variable selection. This highlights the key difference between the settings studied in [3] and our work. You are right that our setting is ‘cleaner’ (motivated by a classic statistical problem), but we believe that the results are not necessarily simpler to prove. Establishing tighter bounds is indeed one of the contributions of our work.
> > >
> > > Thank you.
> > >
> > > Best regards,
> > >
> > > Authors
> > >
> > >
> > > [3] Samy Jelassi, Michael Sander, and Yuanzhi Li. Vision transformers provably learn spatial structure. NeurIPS, 2022.

---

> ### Author Response · Authors · 2024-11-21
>
> >**Q3**:
> The input $x_j$ is concatenated with position encoding, does Theorem 3.2 still hold if using addition instead of concatenation?
>
> **A3**:
> Theoretical studies usually consider concatenation with position encoding for simplicity, like [2, 5].  Intuitively, we suspect that similar results still hold if additions were adopted instead. However, different settings can introduce entirely distinct dynamical systems, making it difficult to provide a rigorous theoretical guarantee at this stage. We believe that exploring various settings of positional encoding is a promising and interesting future work direction.
>
> [1] Ruiqi Zhang, Spencer Frei, and Peter L Bartlett. Trained transformers learn linear models in-context. JMLR, 2024.
>
> [2] Zixuan Wang, Stanley Wei, Daniel Hsu, and Jason D Lee. Transformers provably learn sparse token selection while fully-connected nets cannot. ICML, 2024.
>
> [3] Samy Jelassi, Michael Sander, and Yuanzhi Li. Vision transformers provably learn spatial structure. NeurIPS, 2022.
>
> [4] Spencer Frei, and Gal Vardi. Trained transformer classifiers generalize and exhibit benign overfitting in-context. arXiv preprint arXiv:2410.01774, 2024.
>
> [5] Eshaa Nichani, Alex Damian, and Jason D. Lee. How Transformers Learn Causal Structure with Gradient Descent. ICML, 2024.

---

> ### Author Response · Authors · 2024-11-26
>
> Dear Reviewer,
>
> Thank you very much for your supportive comments. We believe that your concerns and questions have been addressed in our response above. We have also revised the paper and clarified the classic statistical problem of linear classification with group sparsity, which we believe better explains the motivation behind our work. Please let us know if you have any further comments or questions. Thank you.
>
> Best regards,
>
> Authors

---

> ### Comment · Reviewer_CzbC · 2024-11-29
>
> If that is case, I need to low the my rating below average.
>
> (1) the definition S in Eq. 2.2 should not include Z for general transformer architecture in NLP since W is already the attention matrix. And W should be R^{D \times D} if you have D patches for each input instance Z.
>
> (2) If there is only one of the patches is related to the label $y$, it means one parch highly related to $y$.
> In Theorem 2.2, you claim S_{j^*, j } holds for all j \in [D]. If S is normalized softmax, then there exists j \in [D] holds for that. Any typo here?
>
> Open to discuss if there is any misunderstanding here.

---

> > ### Author Response · Authors · 2024-11-30
> >
> > Dear Reviewer,
> >
> >
> > Thank you for your follow-up comments. We believe that these two points are both misunderstandings. We will address each of your questions in detail below.
> >
> > ---
> >
> > Regarding your first question, we would like to clarify that your statement
> >
> > ”the definition $\boldsymbol S$ in Eq. 2.2 should not include $\boldsymbol Z$ since $\boldsymbol W$ is already the attention matrix”
> >
> > is a huge misunderstanding.
> >
> > We first clarify that $\boldsymbol W$ is not the 'attention matrix'. **$\boldsymbol W$ is a parameter matrix reparameterizing the query parameter matrix and the key parameter matrix.** Please note that in practice, a self-attention layer is commonly defined as
> >
> > $\boldsymbol V \boldsymbol Z \mathrm{softmax}[ (\boldsymbol W\^{K}\boldsymbol Z)^\top \boldsymbol W\^{Q} \boldsymbol Z] = \boldsymbol V \boldsymbol Z \mathrm{softmax}(\boldsymbol Z^\top \boldsymbol W\^{K\top} \boldsymbol W\^{Q} \boldsymbol Z)$
> >
> > where $\boldsymbol Z$ is the sequence of patches/tokens with each column $\boldsymbol z_j$ a patch/token, $\boldsymbol W\^{Q}$ is the query parameter matrix, and $\boldsymbol W\^{K}$ is the key parameter matrix.
> >
> > Our paper considers the slightly simplified version with the reparameterization $\boldsymbol W = \boldsymbol W\^{K\top} \boldsymbol W\^{Q} $. This ensures that in our model, the softmax scores indeed directly depend on the tokens, which is the case in practice. Such reparameterization has been widely considered in a series of recent works on theoretical studies of transformers [1, 2, 4], and we are sure that our setting is correct. Please note that the matrices $\boldsymbol Z^\top \boldsymbol W \boldsymbol Z$ and $\mathrm{softmax}(\boldsymbol Z^\top \boldsymbol W \boldsymbol Z)$ are $D\times D$ matrices, and the self-attention layer we consider is
> >
> > $\boldsymbol V \boldsymbol Z \mathrm{softmax}(\boldsymbol Z^\top \boldsymbol W \boldsymbol Z) \quad\quad (*)$
> >
> > Please also note that [3] considers a more simplified model where the softmax scores are directly calculated as $\text{softmax}(\boldsymbol A)$, and the matrix $\boldsymbol A$ is directly updated by gradient descent. Compared to this setting in [3] where the softmax scores no longer depend on tokens, our setting is much closer to the classic definition of self-attention, and this is a strength of our paper.
> >
> > ---
> >
> > Your comment
> >
> > ”In Theorem 2.2, you claim $\boldsymbol{S}\_{j^*, j }$ holds for all $j \in [D]$. If $\boldsymbol{S}$ is normalized softmax, then there exists $j \in [D]$ holds for that. Any typo here?”
> >
> > is also a misunderstanding.
> >
> > We are sure that Theorem 2.2 is stated correctly, and the result for $\boldsymbol S\_{j^*, j }$ holds for all $j \in [D]$. Please note that in our definition, the correct formulation is to normalize each column of $\boldsymbol{S}$ to have a sum of $1$. Our results show that "the $j^*$-th entry at the $j$-th column of $\boldsymbol{S}$ is close to 1, for all $j \in [D]$". This does not contradict the normalization. Again, we are sure that this formulation is correct. Normalizing each column of $\boldsymbol{S}$ is correct according to our definition in equation (*) above, and similar notations have been considered in [1,2].
> >
> > It is true that in our setting, one patch is correlated with $y$. However, this does not mean that the problem we study is easy. As we have clarified, compared with [3], one of the strengths of our work is that we consider the case where softmax scores $\mathrm{softmax}(\boldsymbol Z^\top \boldsymbol W \boldsymbol Z)$ directly depend on the tokens. Even though our intuition is to utilize the correlation between $y$ and one of the patches, the fact that $\boldsymbol Z$ appears inside the softmax essentially “makes everything in the gradients correlated with $y$”. Establishing concrete guarantees under this more complicated setting is one of the contributions of our paper. Moreover, as you acknowledged, our analysis establishes much tighter bounds, which is also an important technical contribution of our work.
> >
> > We hope that our response above addresses your concerns. If you have any further questions, please let us know.
> >
> > ---
> >
> > [1] Zixuan Wang, Stanley Wei, Daniel Hsu, and Jason D Lee. Transformers provably learn sparse token selection while fully-connected nets cannot. ICML, 2024.
> >
> > [2] Yu Huang, Yuan Cheng, and Yingbin Liang. In-context convergence of transformers. ICML, 2024.
> >
> > [3] Samy Jelassi, Michael Sander, and Yuanzhi Li. Vision transformers provably learn spatial structure. NeurIPS, 2022.
> >
> > [4] Ruiqi Zhang, Spencer Frei, and Peter L Bartlett. Trained transformers learn linear models in-context. JMLR, 2024.

---

> ### Comment · Reviewer_CzbC · 2024-12-01
>
> For the self-attention transformer, If you have linear mappings for K and Q, then in general there exists a linear mapping for V from input X. This is how the one layer transformer maps input X into K, Q and V.
>
> (1) From your Eq. 2.2, I do not see linear mapping to transform Z, so I assume you do not need linear mapping for K and Q from the input Z inside S.
>
> (2) I misunderstood, you normalized in column.
>
> Good. I keep my original rating above the average.

---

> > ### Author Response · Authors · 2024-12-02
> >
> > Dear Reviewer,
> >
> > Thank you for your response and for maintaining your original score. We are pleased to see that the misunderstandings have been resolved, and we greatly appreciate your support for our paper.
> >
> > Best regards,
> >
> > Authors

---

> > > ### Comment · Reviewer_CzbC · 2024-12-03
> > >
> > > Is it possible to add another experiment?
> > >
> > > set j*=1 and generate synthetic data, then learn the model and draw S as Fig. 2.
> > > I need to see the attention map when j*=1 for all the synthetic data.
> > >
> > > Thanks,

---

> > > > ### Author Response · Authors · 2024-12-03
> > > >
> > > > Dear Reviewer,
> > > >
> > > > Thanks for your further question. Since the revision period has ended, we are unable to update the manuscript to include more experiment results. However, we can assure that conducting such additional experiments are very straightforward, and we have obtained the experiment results.
> > > >
> > > > Below, we directly give the attention score matrices we have obtained from your suggested experiments. These results are the counterpart of the results of Figure 2 – all experiment setups are the same except that now we set $j^* = 1$.
> > > >
> > > >
> > > > - Heatmap of attention score matrix for $(d, D)=(4, 6)$
> > > > ||||||||
> > > > | ------ | ------ | ------ | ------- | ------ | ------ | ------ |
> > > > ||0.964|0.962|0.957|0.968|0.952|0.966|
> > > > ||0.004|0.004|0.004|0.003|0.004|0.004|
> > > > ||0.010|0.011|0.014|0.009|0.009|0.011|
> > > > ||0.004|0.004|0.004|0.004|0.005|0.004|
> > > > ||0.010|0.011|0.013|0.010|0.020|0.008|
> > > > ||0.008|0.008|0.008|0.006|0.009|0.008|
> > > >
> > > > - Heatmap of attention score matrix for $(d, D)=(2, 4)$
> > > > ||||||
> > > > | ------ | ------ | ------ | ------- | ------ |
> > > > ||0.987|0.987|0.987|0.987|
> > > > ||0.005|0.005|0.005|0.005|
> > > > ||0.003|0.003|0.003|0.003|
> > > > ||0.005|0.005|0.005|0.005|
> > > >
> > > > It is evident that self-attention can effectively attend to the position $j^*=1$ in both cases, supporting our theoretical findings. We will add these results, together with more detailed discussions in the camera-ready version of the paper.
> > > >
> > > > Thank you once again for your efforts in reviewing our paper!
> > > >
> > > > Best regards,
> > > >
> > > > Authors

---

### Author Response · Authors · 2024-11-24
**Revision Overview**

Dear Reviewers,

We appreciate your detailed and constructive comments on our paper. We are glad to hear that all reviewers agree our paper provides interesting and novel insights for the theoretical understanding of transformers. We have addressed all your questions in detail in our individual responses and have updated the manuscript accordingly. We would greatly appreciate it if you could review our responses and the revised manuscript to see whether you are satisfied, and let us know if you have any further comments or suggestions. Below, we give an overview of the major changes we have made in the revised manuscript.

In particular, we appreciate the valuable comments from Reviewer BJ8K and Reviewer tEUD regarding the absence of a formal definition of “group sparsity” in our original manuscript. We have included a formal definition of "group sparsity" from lines 105 to 120 in the revised manuscript, and we would like to clarify that "group sparsity" is a popular concept in statistics and the goal of our work is to study the capability of simple one-layer transformers in solving the classic statistical problem of “linear classification with group sparsity”. We have also provided further explanations to clarify the connections between “linear classification with group sparsity” and our data model in Definition 2.1 (Definition 3.1 in the original manuscript). We believe that the formal definition and the additional explanations can address your concerns and clarify the motivation and significance of our study.

Additionally, we have conducted further experiments on synthetic data as well as real data, to demonstrate our theoretical conclusions on high-dimensional data,  as suggested by Reviewer BJ8K. The results on synthetic and real data are given in Appendices F and G respectively. We would like to remark that all the experiment results align with our theoretical conclusion, offering more robust empirical evidence to support our findings.

We look forward to your further feedback and comments, and we are happy to address any remaining concerns.

$\ $

Best regards,

Authors

---

### Meta-Review · Area_Chair_5pTB · 2024-12-20

**Metareview:**

This work provides a theoretical analysis of transformers, demonstrating their ability to learn structured data with group sparsity. It shows that a one-layer transformer trained via gradient descent can use attention mechanisms to focus on relevant variables while ignoring irrelevant ones. Additionally, a well-pretrained transformer can adapt efficiently to downstream tasks with limited samples, offering insights into how transformers excel in structured data learning.

This paper is borderline at this stage. Some of the reviewers have highlighted a few issues with the paper, and the authors seem to have addressed many of them in the rebuttal. I would encourage the authors to go through the reviews carefully and improve their submission for the next version of this paper.

**Additional Comments On Reviewer Discussion:**

This paper is borderline at this stage. Some of the reviewers have highlighted a few issues with the paper, and the authors seem to have addressed many of them in the rebuttal. I would encourage the authors to go through the reviews carefully and improve their submission for the next version of this paper.

---

### Decision · Program_Chairs · 2025-01-22

Accept (Poster)